# QCircuitBench: A Large-Scale Dataset for Benchmarking Quantum Algorithm Design

**Rui Yang**[1,2]  **Ziruo Wang**[1,2]  **Yuntian Gu**[3]  **Yitao Liang**[4*]  **Tongyang Li**[1,2*]

[1]School of Computer Science, Peking University
[2]Center on Frontiers of Computing Studies, Peking University
[3]School of Intelligence Science and Technology, Peking University
[4]Institute for Artificial Intelligence, Peking University

`ypyangrui@pku.edu.cn`  `zrwang25@stu.pku.edu.cn`  `guyuntian@stu.pku.edu.cn`
`yitaol@pku.edu.cn`  `tongyangli@pku.edu.cn`
`https://github.com/EstelYang/QCircuitBench`

## Abstract

Quantum computing is an emerging field recognized for the significant speedup it offers over classical computing through quantum algorithms. However, designing and implementing quantum algorithms pose challenges due to the complex nature of quantum mechanics and the necessity for precise control over quantum states. Despite the significant advancements in AI, there has been a lack of datasets specifically tailored for this purpose. In this work, we introduce QCircuitBench, the first benchmark dataset designed to evaluate AI's capability in designing and implementing quantum algorithms using quantum programming languages. Unlike using AI for writing traditional codes, this task is fundamentally more complicated due to highly flexible design space. Our key contributions include:

1. A general framework which formulates the key features of quantum algorithm design for Large Language Models.
2. Implementations for quantum algorithms from basic primitives to advanced applications, spanning 3 task suites, 25 algorithms, and 120,290 data points.
3. Automatic validation and verification functions, allowing for iterative evaluation and interactive reasoning without human inspection.
4. Promising potential as a training dataset through preliminary fine-tuning results.

We observed several interesting experimental phenomena: LLMs tend to exhibit consistent error patterns, and fine-tuning does not always outperform few-shot learning. In all, QCircuitBench is a comprehensive benchmark for LLM-driven quantum algorithm design, and it reveals limitations of LLMs in this domain.

## 1 Introduction

Quantum computing is an emerging field in recent decades because algorithms on quantum computers may solve problems significantly faster than their classical counterparts. From the perspective of theoretical computer science, the design of quantum algorithms have been investigated in various research directions - see the survey [Dalzell et al., 2023] and the quantum algorithm zoo [Jordan, 2025]. However, the design of quantum algorithms on quantum computers has been completed manually by researchers. This process is notably challenging due to highly flexible design space and extreme demands for a comprehensive understanding of mathematical tools and quantum properties.

---

*Corresponding authors: `yitaol@pku.edu.cn`, `tongyangli@pku.edu.cn`

For these reasons, quantum computing is often considered to have high professional barriers. As the discipline evolves, we aim to explore more possibilities for algorithm design and implementation in the quantum setting. This is aligned with recent advances in AI for Science, including AlphaFold [Jumper et al., 2021], AlphaGeometry [Trinh et al., 2024], etc. Specifically, large language models (LLMs) have emerged as powerful tools in this domain [Yang et al., 2024b, Zhang et al., 2024, Yu et al., 2024, Ren et al., 2025, Lin et al., 2025]. LLMs represent the best practice of sequential modeling methods at current stage. They have an edge over other models in possessing abundant pre-training knowledge and providing human-friendly interfaces which support human-machine collaboration. Therefore, we employ LLMs as a core component for facilitating quantum algorithm design.

To the best of our knowledge, there is no existing dataset specifically designed for AI-driven quantum algorithm design. Existing work combining quantum computing and AI mostly targets at exploiting quantum computing for AI; there are some papers applying AI for quantum computing, but they either consider niche problems [Nakayama et al., 2023, Schatzki et al., 2021] or limited functions [Tang et al., 2023, Fürrutter et al., 2024], not quantum algorithm datasets of general interest (see Section 2). Unlike classical code generation where abundant data exist, the most challenging aspect for quantum algorithm design is the lack of sufficient data, and hence the difficulty of generalization in training AI models. Therefore, datasets for quantum algorithm design are solicited.

Choosing an appropriate representation is critical for enabling both precise reasoning and practical usability in quantum algorithm design. Descriptions of quantum algorithms in natural language can be verbose and ambiguous. Mathematical formulas, while precise and compact, are difficult to parse and verify automatically. To accommodate with LLMs, we make a change of perspective by formulating quantum algorithm design as quantum code generation. This allows for precise representation of a quantum algorithm, enables automatic verification procedure, and bridges the gap between theoretical design and circuit implementations. Furthermore, meaningful quantum algorithms which can be efficiently implemented have no more than polynomially many gates [Poulin et al., 2011], and thus such formulations have the theoretical benefits of allowing for scalable representations.

**Key Contributions.** We propose QCircuitBench, the first comprehensive, structured dataset for quantum algorithm design. Technically, it has the following key contributions:

- It formulates the task for LLMs with a carefully designed framework encompassing the key features of quantum algorithm design, including problem description, quantum circuit codes, classical post-processing, and verification functions. It maintains the black-box nature of oracles and characterizes query complexity properly.
- It implements a wide range of quantum algorithms, covering 3 task suites, 25 algorithms, and 120,290 data points. The dataset spans from basic primitives and textbook-level algorithms to advanced applications such as Generalized Simon's Problem, demonstrating compatibility with complex algorithms and easy extensibility.
- It has automatic validation and verification functions, enabling iterative, human-free evaluation and supporting interactive reasoning to enhance performance.
- It showcases the potential as a training dataset through preliminary fine-tuning results. As we expand the dataset to include more algorithms and explore novel fine-tuning methods, it will hopefully contribute to interactive quantum algorithm design and implementation significantly.

## 2 Related Work

**Quantum Machine Learning.** To the best of our knowledge, QCircuitBench is the first dataset tailored specifically for quantum algorithm design. Previous efforts combining quantum computing with AI primarily fall under the category of Quantum Machine Learning (QML), which aims at leveraging the unique properties of quantum systems to enhance machine learning algorithms and achieve improvements over their classical counterparts [Schuld et al., 2015, Biamonte et al., 2017, Ciliberto et al., 2018]. Corresponding datasets often focus on encoding classical data into quantum states. For instance, MNISQ [Placidi et al., 2023] is a dataset of quantum circuits representing the original MNIST dataset [LeCun et al., 1998] generated by the AQCE algorithm [Shirakawa et al., 2021]. Another category of datasets focuses on collecting quantum data to demonstrate quantum advantages since classical machine learning methods can fail to characterize particular patterns of quantum data. Nakayama et al. [2023] created a VQE-generated quantum circuit dataset for

classification of variational ansatzes. NTangled [Schatzki et al., 2021] further investigated different types of entanglement and composed quantum states with various multipartite entanglement for classification. While these datasets successfully demonstrate quantum supremacy, the practical applications of the problem addressed are unclear.

**AI for Quantum Computing.** This research direction explores the possibility of leveraging AI to facilitate the advancement of quantum computing. QDataSet [Perrier et al., 2022] collects data from simulations of one- and two-qubit systems and targets training classical machine learning algorithms for quantum control, quantum tomography, and noise mitigation. LLM4QPE [Tang et al., 2023] is a large language model style paradigm for predicting quantum system properties with pre-training and fine-tuning workflows. While the paradigm is interesting, the empirical experiments are limited to two downstream tasks: quantum phase classification and correlation prediction. Fürrutter et al. [2024] studied the application of diffusion models [Sohl-Dickstein et al., 2015, Rombach et al., 2022] to quantum circuit synthesis [Saeedi and Markov, 2013, J. et al., 2022]. Scalability issues must be addressed to achieve practical and meaningful unitary compilation through this methodology.

**Quantum Circuit Benchmarks.** The aforementioned works represent meaningful explorations at the intersection of AI and quantum computing. However, none of them considers the task which interests the quantum computing community (from the theoretical side) the most: quantum algorithm design. Our work aims to take the first step in bridging this gap. It is worth noting that several quantum algorithm circuit benchmarks already exist, such as QASMBench [Li et al., 2023], MQTBench [Quetschlich et al., 2023], and VeriQBench [Chen et al., 2022]. However, these benchmarks are designed to evaluate the performance of NISQ (Noisy Intermediate-Scale Quantum) [Preskill, 2018] machines or quantum software tools, rather than for training and evaluating AI models. For instance, QASMBench includes a diverse variety of quantum circuits based on OpenQASM representation [Cross et al., 2022], covering quantum circuits with qubit sizes ranging from 2 to 127. However, it fails as a dataset for AI in that it includes only a few entries for each algorithm and ignores the post-processing procedure and construction of different oracles, which are crucial to quantum algorithm design. Similar limitations apply to MQTBench and VeriQBench.

## 3 QCircuitBench Dataset

### 3.1 Task Suite

For the general purpose of quantum algorithm design, we consider three categories of tasks: oracle construction, quantum algorithm design, and random circuit synthesis. These tasks are crucial for devising and implementing quantum algorithms, with oracle construction serving as the premise for algorithm design, and random circuits serving as a main demonstration for quantum supremacy. These task suites encompass 25 algorithms and a total of 120,290 data points with the following distribution:

#### 3.1.1 Task I: Oracle Construction

This task suite contains 35,872 data points in total, focused on two types of oracle constructions.

To study a Boolean function $f\colon \{0,1\}^n \to \{0,1\}^m$, we need to gain its access. In quantum computing, the function $f$ is encoded as an oracle $U_f$ such that for any $x \in \{0,1\}^n$, $z \in \{0,1\}^m$, $U_f|x\rangle|z\rangle = |x\rangle|z \oplus f(x)\rangle$, where $\oplus$ is the plus modulo 2. The construction of $U_f$ using quantum gates is deeply rooted in reversible quantum logic synthesis, which remains a challenge for complex Boolean functions. In this dataset, we mainly focus on the construction of textbook-level oracles: Bernstein-Vazirani Problem [Bernstein and Vazirani, 1993], Deutsch-Jozsa Problem [Deutsch and Jozsa, 1992], Simon's Problem [Simon, 1997], and Grover's algorithm for unstructured search [Grover, 1996] (including constructions of both the oracle and the diffusion operator).

There is another category of more flexible oracle construction tasks which we refer to as "Problem Encoding". For example, one can apply Grover's oracle to solving constraint problems such as SAT and triangle finding [Ambainis, 2004]. Formulating problem encoding tasks for LLMs slightly differs from quantum logic synthesis, and we refer the readers to Appendix A.4 for more detailed discussion.

### 3.1.2 Task II: Quantum Algorithm Design

In this category, we cover a wide range of quantum algorithms with varying complexity, from fundamental primitives to *advanced applications*, covering 6,534 data points:

- Textbook-level algorithms: These range from the Bernstein-Vazirani problem [Bernstein and Vazirani, 1993], Deutsch-Jozsa problem [Deutsch and Jozsa, 1992], Simon's problem [Simon, 1997], Grover's algorithm [Grover, 1996], phase estimation [Kitaev, 1995], quantum Fourier transform [Coppersmith, 2002], GHZ state preparation [Greenberger et al., 2007], W state preparation [Dür et al., 2000], random number generator [Herrero-Collantes and Garcia-Escartin, 2017], swap test [Barenco et al., 1997, Buhrman et al., 2001] to Shor's algorithm [Shor, 1999] for factorization, one of the most famous quantum algorithms with superpolynomial speedup.

- Generalized Simon's Problem [Ye et al., 2021]: This is a more advanced version of the standard Simon's problem and an active area of research in recent years [Ye et al., 2021, Wu et al., 2022]. The setting is formally stated as follows: given an unknown function $f \colon \mathbb{Z}_p^n \to X$, where $X$ is a finite set and a $k$ is a positive integer satisfying $k < n$, it is guaranteed that there exists a subgroup $S \le \mathbb{Z}_p^n$ of rank $k$ such that for any $x, y \in \mathbb{Z}_p^n$, $f(x) = f(y)$ iff $x - y \in S$. The goal is to find $S$. Intuitively, the generalized Simon's problem extends the standard Simon's problem from binary to $p$-ary bases and from a single secret string to a subgroup of rank $k$. We conduct experiments for the case $p = 3$ (ternary).

- Variational quantum algorithms (VQAs): Beyond universal quantum algorithms, VQAs including VQE [Peruzzo et al., 2014] for ground-state energy estimation, QAOA [Farhi et al., 2014] for combinatorial optimization, QAE [Romero et al., 2017] for quantum state compression, and ENC [LaRose and Coyle, 2020] for encoding classical data into quantum representations, are potentially implementable on near-term quantum computers. Unlike fixed-circuit quantum algorithms, VQAs require iterative optimization over parameterized circuits, challenging models to generate both circuit structures and effective initialization and optimization procedures.

- Quantum information protocols: Additionally, we also include quantum information protocols such as quantum teleportation [Bennett et al., 1993] and quantum key distribution [Bennett and Brassard, 2014], which have wide applications in quantum communications, quantum cryptography, etc. See Appendix B for further details.

### 3.1.3 Task III: Random Circuit Synthesis

The third task we consider is random circuit synthesis, containing 77,884 data points. On the one hand, random circuit sampling is the first algorithm for showing quantum supremacy by Google [Arute et al., 2019], and is still widely applied to demonstrate the power of quantum algorithms in recent research [Wu et al., 2021, Bluvstein et al., 2024, DeCross et al., 2024]. In this suite, circuits are randomly sampled from a Clifford gate set {H, S, CNOT} and a universal set {H, S, T, CNOT}, and the task is to generate circuits reproducing the specified quantum state.

### 3.2 Dataset Structure

The overall structure of QCircuitBench is illustrated in Figure 1 (more details given in Appendix A).

**Programming Languages and Platforms.** QCircuitBench is designed to generalize across platforms. We provide two functionally equivalent versions, Qiskit [Javadi-Abhari et al., 2024]+OpenQASM [Cross et al., 2022] and Cirq [Cirq Developers, 2025], so the same tasks, oracles, and verification flow are available in either environment. For concreteness, this subsection describes the Qiskit+OpenQASM version. Similar dataset specification applies to Cirq.

**Design Principles.** Different tasks encounter different challenges. Here we highlight the following construction principles, which are specially designed to adapt to these tasks:

- **Paradox of Oracle Implementation:** Quantum algorithms often treat the oracle $U_f$ as a black-box, aiming to deduce properties of function $f(x)$ without directly accessing its implementation. However, quantum circuits for real-world platforms need an explicit gate definition to compile and run successfully. To address this challenge, which is often overlooked in tutorials and benchmarks, we provide the oracle as a black-box gate with its explicit definition in a separate "oracle.inc"

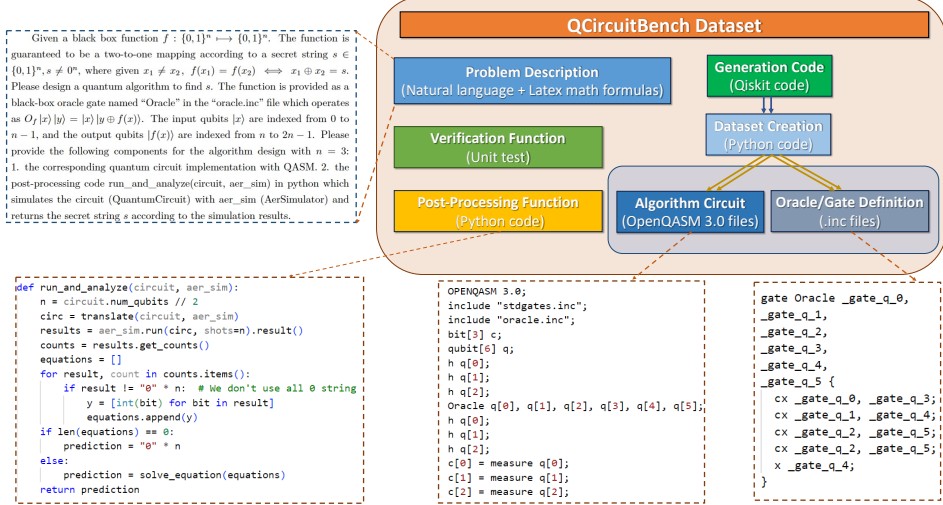

Figure 1: Structure of QCircuitBench. The components of QCircuitBench are presented in the frame on the top-right. As a showcase, this figure presents the components for Simon's problem [Simon, 1997], including its problem description in natural language, post-processing function in python code, circuit in a .qasm file, and oracle definition in a .inc file.

library. This complies with OpenQASM syntax while ensures the oracle's functionality is accessible to the model without exposing its internal structure.

- **Classical Processing Specification:** A quantum algorithm involves not only quantum circuits but also the classical processing steps to interpret measurement results. For example, in Simon's algorithm, the model must solve linear equations $s \cdot y_i = 0$ from measured $y_i$. In addition to quantum circuits, we require the model to specify the classical processing function and define the shot count to characterize query complexity, crucial for the theoretical analysis of the algorithm.

- **Custom Quantum Gates:** Some composite gates, not part of the standard QASM library, are essential for advanced algorithms. To avoid model distractions, we provide these custom gates, such as multi-controlled X gates (45,060 lines for 14 qubits), in a "customgates.inc" file. These gates are defined hierarchically, allowing the model to use them without the burden of generating complex gate structures.

- **Automated Verification Function:** To ensure that model outputs are syntactically correct and functionally valid, we implement automatic verification tools that check OpenQASM syntax and circuit functionality. Instead of performing exhaustive formal verification or equivalence checking, we use a suite of end-to-end test cases to validate the functional correctness of the generated circuits, enabling efficient model evaluation.

Based on theses principles, we proposed the framework of QCircuitBench. Below is a more detailed explanation for the 7 components of the dataset:

1. **Problem Description:** carefully hand-crafted prompts stating the oracle to be constructed or the target problem to be solved in natural language and latex math formulas. If the problem involves the usage of a quantum oracle or composite gates beyond the standard gate library, the interfaces of the oracle / gate will also be included (input qubits, output qubits, function mechanism).

2. **Generation Code:** one general Qiskit code to create quantum circuits for oracles or algorithms of different settings, such as distinct secret strings or various qubit numbers. We choose Qiskit as the main experiment platform because it is a general quantum programming software widely used for the complete workflow from creating quantum circuits to transpiling, simulation, and execution on real hardware.

3. **Algorithm Circuit:** a .qasm file storing the quantum circuit for each specific setting. We choose OpenQASM 3.0 as the format to store the quantum circuits, because Qiskit, as a python library, can only create quantum circuits at runtime instead of explicitly saving the circuits at gate level.[2]

4. **Post-Processing Function:** this is for Quantum Algorithm Design task only, see Section 3.1.2. The function takes a complete quantum circuit as input, uses the Qiskit AerSimulator to execute the circuit, and returns the final answer to the original problem according to the simulation results. For state preparation problems such as creating a GHZ state of $n$ qubits, this function returns the qubit indices of the generated state.

5. **Oracle/Gate Definition:** a .inc file to provide definitions of oracles or composite gates. For oracle construction tasks, this only includes the definition of composite gates required to build the oracle. For algorithm design tasks, we also provide the gate definition of the oracle in this file, which successfully delivers the oracle in a black-box way.

6. **Verification Function:** a function to evaluate whether the implemented oracle or algorithm achieves the desired purpose with grammar validation and semantic verification. The function returns two scores: syntax score and semantic score, with details explained in Section 4.1, Evaluation Metrics. If the program fails to run successfully, a detailed error message is provided as the feedback for LLMs to improve through interactive reasoning.[3] All the verification functions were executed by classical simulations in our experiments, but the APIs we implemented are compatible with IBM hardware and can be easily adapted to real quantum computers.

7. **Dataset Creation Script:** the script to create the dataset from scratch in the format suitable for benchmarking or fine-tuning LLMs. It contains the following functions: 1. generate primitive QASM circuits. 2. extract gate definitions and add include instructions to create an algorithm circuit as the direct output. 3. validate and verify the correctness of the data points in the dataset. 4. concatenate the circuit with problem description as a json file for the benchmark pipeline.

This structure of QCircuitBench provides a general framework to formulate quantum algorithm design for large language models, with an easy extension to more advanced quantum algorithms.

## 4 Experiments

### 4.1 Benchmarking LLMs on QCircuitBench

We benchmark the quantum algorithm design capabilities of leading closed-source and open-source large language models using QCircuitBench. The workflow of our benchmark is illustrated in Figure 2. The total compute footprint per model includes ≈48 A100-GPU hours for inference and an additional 24 single-core CPU hours for verification and simulation.

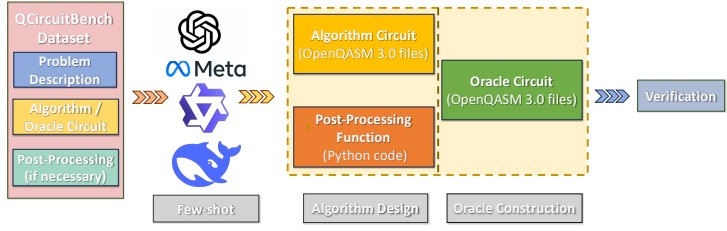

Figure 2: Flowchart of benchmarking QCircuitBench.

**Models.** We evaluate a range of state-of-the-art LLMs, including both proprietary and open-source representatives. From OpenAI, we include GPT-4o [OpenAI et al., 2024], which serve as widely adopted baseline due to its strong general-purpose performance. For open-source models, we

---

[2]Although currently the Qiskit APIs for importing and dumping OpenQASM 3.0 files are still in experimental stage, we choose to adopt version 3.0 over 2.0 in that it supports saving parameterized circuits, which allows for extending the framework to variational quantum algorithms [Cerezo et al., 2021].

[3]The verification function explicitly integrates the oracle/gate definition library with output algorithm circuit since Qiskit importer for OpenQASM 3.0 does not support non-standard gate libraries currently.

benchmark LLAMA-3-8B [Meta, 2024], a prominent member of the LLAMA family [Touvron et al., 2023a,b, Meta, 2024] known for its efficiency and competitive performance. We further include Qwen 2.5 [Yang et al., 2024a] and DeepSeek-R1 [Guo et al., 2025], two recently released models that have shown promising results across a variety of reasoning and code generation tasks. To contextualize model performance, we conduct a human study and report human baseline scores at the bottom line of each table as reference. The instruction for human study is presented in Appendix A.3.

All models are evaluated using a few-shot prompting strategy, a widely used method in generative language model evaluation [Brown et al., 2020, Xie et al., 2021, Dong et al., 2024]. We provide 1 or 5 in-context examples, followed by the task description. To prevent data leakage, we use $k$-fold cross validation, holding out each quantum algorithm from the prompt during evaluation.

**Evaluation Metrics.** We use four evaluation metrics:

1. **BLEU Score**: This metric measures how closely the generated code matches the reference code, with a higher BLEU score indicating more similarity.

2. **Byte Perplexity**: Evaluates the model's ability to predict the next byte in a sequence. Lower PPL suggests stronger autoregressive modeling and more coherent outputs.

3. **Verification Function**: This function performs syntax checking and semantic evaluation of the code produced by language models.

   To be specific, we evaluate the result using three criteria:

   (a) QASM Syntax: We first check the syntax of the OpenQASM code provided by the model. The syntax verification function $V_{\text{QASM}}(q)$ is set to be 1 if the QASM syntax is correct, and 0 otherwise.

   (b) Python Syntax: Similarly, the syntax of the post-processing Python code, denoted $V_{\text{code}}(c)$, is set to be 1 if the Python syntax is correct, and 0 otherwise.

   (c) Semantic: If both syntax checks pass, we proceed to evaluate the functional correctness. For quantum algorithm design and oracle construction tasks, for each test case $t$, we run the quantum circuit simulation for a number of shots $M$, and compare the result with the ground truth. The score reflects the success rate over test cases, which is calculated as:

   $$E_{\text{sem}} = \frac{\sum_{t=1}^{T} \sum_{m=1}^{M} \mathbb{I}[\text{result} = \text{ground-truth}]}{T \times M}.$$

   For state preparation and random circuit synthesis tasks, let $|\psi^\star\rangle$ be the target state and $|\psi\rangle$ be the state produced by the model's circuit, the semantic score is defined as the fidelity between the generated and target quantum states:

   $$E_{\text{sem}} = |\langle\psi^\star|\psi\rangle|^2,$$

   Overall, this task-specific semantic score $E_{\text{sem}}$ provides a principled and quantitative measure of quantum correctness tailored to each task.

   The final verification score is a triplet $(V_{\text{QASM}}(q), V_{\text{code}}(c), E_{\text{sem}})$.

4. **Efficiency Evaluation**: We incorporate another three metrics to evaluate the efficiency of generated algorithms.

   (a) Gate Count Ratio: the number of quantum gates used by the model, divided by that of the reference implementation. A lower ratio indicates higher gate-level efficiency.

   (b) Shot Count Ratio: the query complexity required by the model, divided by the reference. Again, a smaller value indicates better efficiency.

   (c) Time Count Ratio: the total execution time of the model, divided by the reference. A lower ratio reflects better runtime performance.

The BLEU scores of all three tasks are shown in Figure 3. The qasm syntax and semantic verification scores of quantum algorithm design task are shown in Table 1 and Table 2. We include the results of python syntax verification, efficiency metrics, Byte Perplexity, and open-book setting of quantum algorithm design task in Appendix C.1.1. Other results for oracle construction and random circuit synthesis tasks are included in Appendix C.1.2. Benchmarking results for the Cirq implementation are presented in Appendix C.1.3.

We observe the following phenomena from the results:

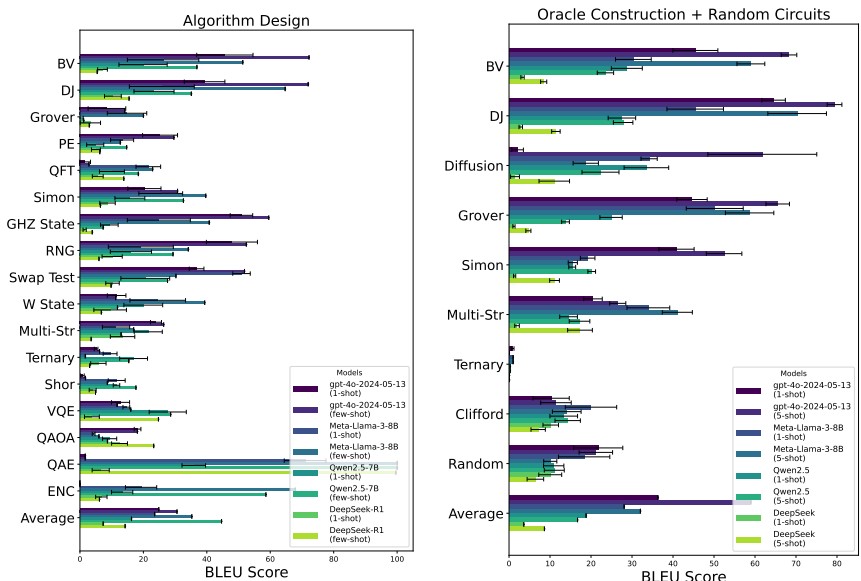

Figure 3: Benchmarking algorithm design and oracle construction tasks in BLEU scores.

Table 1: QASM syntax score for benchmarking quantum algorithm design.

| Model | Shot | Bernstein Vazirani | Deutsch Jozsa | Grover | Phase Estimation | QFT | Simon | GHZ | Random Number Generator | Swap Test | W State | Generalized Simon (multi-str) | Generalized Simon (ternary) | Shor | VQE | QAOA | QAE | ENC | Avg |
|---|---|---|---|---|---|---|---|---|---|---|---|---|---|---|---|---|---|---|---|
| GPT-4o | 1 | 0.0000 (±0.0000) | 0.0000 (±0.0000) | 0.0000 (±0.0000) | 0.0000 (±0.0000) | 0.0000 (±0.0000) | 0.0000 (±0.0000) | 1.0000 (±0.0000) | 1.0000 (±0.0000) | 0.0000 (±0.0000) | 0.0000 (±0.0000) | 0.0000 (±0.0000) | 0.0000 (±0.0000) | 0.0000 (±0.0000) | 0.2308 (±0.0843) | 1.0000 (±0.0000) | 0.8333 (±0.0904) | 0.5833 (±0.1486) | 0.2734 |
| GPT-4o | 5 | 1.0000 (±0.0000) | 1.0000 (±0.0000) | 0.0000 (±0.0000) | 0.6154 (±0.1404) | 0.5385 (±0.1439) | 0.9231 (±0.0769) | 0.5714 (±0.2020) | 1.0000 (±0.0000) | 1.0000 (±0.0000) | 0.4444 (±0.1757) | 0.0769 (±0.0769) | 0.1111 (±0.1111) | 0.0000 (±0.0000) | 0.2308 (±0.0843) | 0.7222 (±0.1086) | 1.0000 (±0.0000) | 0.5833 (±0.1486) | 0.5775 |
| Llama3 | 1 | 0.1538 (±0.1042) | 0.2308 (±0.1216) | 0.3077 (±0.1332) | 0.4615 (±0.1439) | 0.0000 (±0.0000) | 0.1538 (±0.1042) | 0.1429 (±0.1429) | 0.4615 (±0.1439) | 0.1429 (±0.0971) | 0.3333 (±0.1667) | 0.5385 (±0.1439) | 0.4444 (±0.1757) | 0.0000 (±0.0000) | 0.2574 (±0.0285) | 0.1667 (±0.0544) | 0.0000 (±0.0000) | 0.3438 (±0.0853) | 0.2435 |
| Llama3 | 5 | 0.5385 (±0.1439) | 0.3846 (±0.1404) | 0.6154 (±0.1404) | 0.5385 (±0.1439) | 0.3846 (±0.1404) | 0.1538 (±0.1042) | 0.2857 (±0.1844) | 0.9231 (±0.0769) | 0.5000 (±0.1387) | 0.3333 (±0.1667) | 0.8462 (±0.1042) | 0.3333 (±0.1667) | 0.0000 (±0.0000) | 0.2363 (±0.0277) | 0.9375 (±0.0353) | 0.0000 (±0.0000) | 0.8125 (±0.0701) | 0.4602 |
| Qwen 2.5 | 1 | 0.0769 (±0.0769) | 0.1538 (±0.1042) | 0.0000 (±0.0000) | 0.0769 (±0.0769) | 0.0769 (±0.0769) | 0.3077 (±0.1216) | 0.4286 (±0.2020) | 0.2308 (±0.1429) | 0.2857 (±0.1253) | 0.2222 (±0.1470) | 0.5385 (±0.1439) | 0.2222 (±0.1470) | 0.0000 (±0.0000) | 0.3544 (±0.0324) | 0.0583 (±0.0482) | 1.0000 (±0.0000) | 1.0000 (±0.0000) | 0.2844 |
| Qwen 2.5 | 5 | 0.3077 (±0.1332) | 0.6154 (±0.1404) | 0.1538 (±0.1042) | 0.3077 (±0.1332) | 0.2308 (±0.1216) | 0.1538 (±0.1042) | 0.4286 (±0.2020) | 0.6154 (±0.1404) | 0.5714 (±0.1373) | 0.2222 (±0.1470) | 0.4615 (±0.1439) | 0.2222 (±0.1470) | 0.0000 (±0.0000) | 0.3544 (±0.0311) | 0.0583 (±0.0291) | 1.0000 (±0.0000) | 0.7188 (±0.0808) | 0.4307 |
| DeepSeek-R1 | 1 | 0.0000 (±0.0000) | 0.0769 (±0.0769) | 0.0000 (±0.0000) | 0.0000 (±0.0000) | 0.0000 (±0.0000) | 0.0000 (±0.0000) | 0.1429 (±0.1429) | 0.0769 (±0.0769) | 0.0714 (±0.0714) | 0.0000 (±0.0000) | 0.1538 (±0.1042) | 0.0000 (±0.0000) | 0.0000 (±0.0000) | 0.07173 (±0.0168) | 0.2292 (±0.0613) | 0.0000 (±0.0000) | 0.1563 (±0.0652) | 0.0576 |
| DeepSeek-R1 | 5 | 0.3846 (±0.1404) | 0.0769 (±0.0769) | 0.0000 (±0.0000) | 0.0769 (±0.0769) | 0.0769 (±0.0769) | 0.0000 (±0.0000) | 0.0000 (±0.0000) | 0.1538 (±0.1042) | 0.1429 (±0.0971) | 0.0000 (±0.0000) | 0.2308 (±0.1216) | 0.0000 (±0.0000) | 0.0000 (±0.0000) | 0.0084 (±0.0060) | 0.4167 (±0.0719) | 0.0000 (±0.0000) | 0.4375 (±0.0891) | 0.1768 |
| Human | - | 0.5000 | 1.0000 | 0.0000 | 1.0000 | 1.0000 | 1.0000 | 1.0000 | 0.0000 | 0.0000 | 1.0000 | 1.0000 | 0.0000 | 0.5000 | 1.0000 | 1.0000 | 1.0000 | 0.6667 | 0.6862 |

- We observe notable discrepancies among BLEU, syntax, and semantic scores. While BLEU loosely correlates with verification outcomes, counter-examples exist. For instance, the GHZ state attains relatively high BLEU but low semantic accuracy. Although models achieve relatively high QASM syntax scores on several algorithms, their overall semantic performance remains poor. This observation highlights the necessity of our multi-faceted verification function.

- Few-shot learning improves model performance on simpler tasks, demonstrating the models' ability to learn effectively from contextual examples. Notably, in the Bernstein-Vazirani Algorithm, the QASM syntax score of GPT-4o improves from 0.0000 to 1.0000, and in the Deutsch-Jozsa Algorithm, Qwen 2.5 achieves an increase in semantic score from 0.0000 to 0.2854. However, even with few-shot prompting, all models struggle on more complex algorithms such as Quantum Fourier Transform, Generalized Simon's Problem, and Shor's Algorithm, revealing clear differences in task difficulty.

- Near-term quantum algorithms pose persistent challenges. In particular, for variational quantum algorithms, models frequently fail to construct correct parameterized circuits or apply optimization procedures appropriately. For example, DeepSeek-R1 achieves all zero scores on semantic score of VQE and QAOA, reflecting the current limitations of LLMs in modeling hybrid quantum-classical workflows.

- GPT-4o consistently excels at long-context comprehension, outperforming other models across tasks and demonstrating superior in-context learning. In contrast, DeepSeek-R1 often underperforms due to its long-chain reasoning style, which often exceeds the context window before producing a complete and verifiable solution.

Table 2: Semantic score for benchmarking quantum algorithm design.

| Model | Shot | Bernstein Vazirani | Deutsch Jozsa | Grover | Phase Estimation | QFT | Simon | GHZ | Random Number Generator | Swap Test | W State | Generalized Simon (multi-str) | Generalized Simon (ternary) | Shor | VQE | QAOA | QAE | ENC | Avg |
|---|---|---|---|---|---|---|---|---|---|---|---|---|---|---|---|---|---|---|---|
| GPT-4o | 1 | 0.0000 (±0.0000) | 0.0000 (±0.0000) | 0.0000 (±0.0000) | 0.0000 (±0.0000) | 0.0000 (±0.0000) | 0.0000 (±0.0000) | 0.0000 (±0.0000) | 0.0000 (±0.0000) | 0.0000 (±0.0000) | 0.0000 (±0.0000) | 0.0000 (±0.0000) | 0.0000 (±0.0000) | 0.0000 (±0.0000) | 0.0000 (±0.0000) | 0.0000 (±0.0000) | 0.0000 (±0.0000) | 0.4167 (±0.1205) | 0.0245 |
| GPT-4o | 5 | 1.0000 (±0.0000) | 1.0000 (±0.0000) | 0.0000 (±0.0000) | 0.0846 (±0.0576) | 0.0000 (±0.0000) | 0.0923 (±0.0769) | 0.0000 (±0.0000) | 0.0000 (±0.0000) | 0.7852 (±0.0203) | 0.1156 (±0.0585) | 0.0000 (±0.0000) | 0.0000 (±0.0000) | 0.0000 (±0.0000) | 0.0000 (±0.0000) | 0.0000 (±0.0000) | 0.0000 (±0.0000) | 0.3333 (±0.1421) | 0.2006 |
| Llama3 | 1 | 0.0000 (±0.0000) | 0.0154 (±0.0154) | 0.0019 (±0.0019) | 0.0000 (±0.0000) | 0.0000 (±0.0000) | 0.0000 (±0.0000) | 0.0000 (±0.0000) | 0.0000 (±0.0000) | 0.0000 (±0.0000) | 0.0591 (±0.0591) | 0.0000 (±0.0000) | 0.0000 (±0.0000) | 0.0000 (±0.0000) | 0.0000 (±0.0000) | 0.0000 (±0.0000) | 0.0000 (±0.0000) | 0.1101 (±0.0526) | 0.0110 |
| Llama3 | 5 | 0.0442 (±0.0442) | 0.2269 (±0.1017) | 0.0000 (±0.0000) | 0.0000 (±0.0000) | 0.0000 (±0.0000) | 0.0000 (±0.0000) | 0.0000 (±0.0000) | 0.0000 (±0.0000) | 0.0986 (±0.0673) | 0.0000 (±0.0000) | 0.0000 (±0.0000) | 0.0000 (±0.0000) | 0.0000 (±0.0000) | 0.0000 (±0.0000) | 0.0000 (±0.0000) | 0.0000 (±0.0000) | 0.1799 (±0.0482) | 0.0323 |
| Qwen 2.5 | 1 | 0.0000 (±0.0000) | 0.0000 (±0.0000) | 0.0000 (±0.0000) | 0.0000 (±0.0000) | 0.0000 (±0.0000) | 0.0000 (±0.0000) | 0.0000 (±0.0000) | 0.0000 (±0.0000) | 0.0000 (±0.0000) | 0.0000 (±0.0000) | 0.0000 (±0.0000) | 0.0000 (±0.0000) | 0.0000 (±0.0000) | 0.0000 (±0.0000) | 0.0000 (±0.0000) | 0.0000 (±0.0000) | 0.0000 (±0.0000) | 0.0000 |
| Qwen 2.5 | 5 | 0.0154 (±0.0154) | 0.2854 (±0.0947) | 0.0000 (±0.0000) | 0.0000 (±0.0000) | 0.0000 (±0.0000) | 0.0000 (±0.0000) | 0.0000 (±0.0000) | 0.0000 (±0.0000) | 0.0000 (±0.0000) | 0.0000 (±0.0000) | 0.0000 (±0.0000) | 0.0000 (±0.0000) | 0.0000 (±0.0000) | 0.0042 (±0.0042) | 0.1042 (±0.0446) | 1.0000 (±0.0000) | 0.2881 (±0.0801) | 0.0998 |
| DeepSeek-R1 | 1 | 0.0000 (±0.0000) | 0.0600 (±0.0600) | 0.0000 (±0.0000) | 0.0000 (±0.0000) | 0.0000 (±0.0000) | 0.0000 (±0.0000) | 0.0000 (±0.0000) | 0.0000 (±0.0000) | 0.0605 (±0.0605) | 0.0000 (±0.0000) | 0.0000 (±0.0000) | 0.0000 (±0.0000) | 0.0000 (±0.0000) | 0.0000 (±0.0000) | 0.0000 (±0.0000) | 0.0000 (±0.0000) | 0.0938 (±0.0524) | 0.0126 |
| DeepSeek-R1 | 5 | 0.0788 (±0.0768) | 0.0000 (±0.0000) | 0.0000 (±0.0000) | 0.0000 (±0.0000) | 0.0000 (±0.0000) | 0.0000 (±0.0000) | 0.0000 (±0.0000) | 0.0000 (±0.0000) | 0.0000 (±0.0000) | 0.0000 (±0.0000) | 0.0000 (±0.0000) | 0.0000 (±0.0000) | 0.0000 (±0.0000) | 0.0000 (±0.0000) | 0.0000 (±0.0000) | 0.0000 (±0.0000) | 0.0948 (±0.0387) | 0.0102 |
| Human | - | 0.2500 | 0.0000 | 0.0000 | 0.8500 | 0.0000 | 0.0000 | 0.0000 | 0.0000 | 0.0000 | 0.5000 | 0.5000 | 0.0000 | 0.0000 | 0.0000 | 0.0000 | 0.2357 | 0.0000 | 0.1374 |

**Types of Errors Made by LLMs.** In Appendix C.3, we include several case studies to illustrate and analyze various types of errors made by LLMs. In particular, they can be summarized as follows:

- Improvisation error: GPT-4o tends to use advanced OpenQASM 3.0 features unsupported by Qiskit yet and novel namespace which might result in global conflicts in one-shot setting. This tendency to improvise by drawing on pre-training knowledge rather than closely following the syntax of the example leads to avoidable "errors" and low verification scores. This issue is significantly alleviated in the 5-shot setting, highlighting GPT-4o's strong in-context learning ability. A detailed case study is given in Appendix C.3.1.

- Counting error: LLMs often fail to correctly identify the positions of '1's in a binary string when constructing oracles for problems such as Bernstein-Vazirani. For instance, given the secret string $s = 000101$, GPT-4o misplaces the control qubits for CX gates, despite being explicitly reminded of the correct rule and asked to list the indices with value 1. This misidentification highlights a fundamental limitation of LLMs in performing basic indexing tasks. A detailed case study is given in Appendix C.3.2.

- Data contamination: We observe a performance separation between writing general Qiskit codes and explicit gate-level circuits in QASM. Since Qiskit provides detailed tutorial with general codes for several algorithms, LLMs may rely on memorization and retrieval rather than genuine algorithm design. Our dataset, based on QASM files created from scratch, may help circumvent this issue and serve as a stable and fair method for benchmarking AI syntax learning. A detailed case study is given in Appendix C.3.3.

## 4.2 Fine-tuning on QCircuitBench

Although QCircuitBench is targeted as a benchmark dataset at current stage, we consider fine-tuning or training from scratch based on our dataset as an interesting and important research direction. The unique nature of quantum data requires novel fine-tuning methods and model architecture designs, which could serve as a standalone topic. As a preliminary demonstration, we present the results of fine-tuning the Llama3-8B model on the oracle construction and random circuit synthesis tasks here.

Following the QLoRA approach by Dettmers et al. [2024], we first quantize the model to 8-bit precision and then fine-tune it using LoRA [Hu et al., 2022]. Our experiments use the fp16 data type and apply LoRA with rank $r = 16$, scaling factor $\alpha = 32$, and modules inserted into all query and value projection layers. We adopt the AdamW optimizer [Loshchilov and Hutter, 2019] and set the LoRA dropout rate to $0.05$. We define an *aggregated verification score* to represent the overall syntax and semantic performance of the model. If grammar errors exist, the function returns -1; if the program can execute successfully, the function returns a score between [0, 1] indicating the success rate on test cases. The results are shown as follows:

We observe that the Llama3 model demonstrates the most notable improvement on Grover's algorithm after fine-tuning, with the verification score increased by 0.1900. Case studies on the Bernstein-Vazirani oracle reveal that, before fine-tuning, the model would indiscriminately apply CX gates to all qubits. After fine-tuning, it begins to selectively apply CX gates to qubits corresponding to '1's in the secret string. While some counting errors persist, the model occasionally identifies all correct positions, demonstrating a marked improvement. This suggests that fine-tuning enables the model to internalize structural patterns in oracle construction, leading to improved performance across tasks.

Regarding the interesting performance decrease on Clifford and universal random circuits, we conducted additional experiments on temperature and entropy. We refer the readers to Appendix C.2.1

Table 3: Fine-tuning oracle construction scores.

| Score | Model | Setting | Bernstein-Vazirani | Deutsch-Jozsa | Grover | Simon | Clifford | Universal | Avg |
|---|---|---|---|---|---|---|---|---|---|
| BLEU | gpt4o | few-shot(5) | 95.6388 (±0.3062) | 91.0564 (±0.6650) | 92.0620 (±0.6288) | 80.3390 (±2.0900) | 39.5469 (±3.6983) | 33.3673 (±3.1007) | 72.0017 |
| | Llama3 | few-shot(5) | 53.5574 (±5.2499) | 69.8996 (±5.7812) | 61.3102 (±5.4671) | 26.3083 (±2.0048) | 13.0729 (±0.9907) | 13.4185 (±1.2299) | 39.5945 |
| | Llama3 | finetune | 76.0480 (±7.9255) | 71.8378 (±2.4179) | 67.7892 (±7.8900) | 43.8469 (±3.2998) | 10.8978 (±0.6169) | 7.1854 (±0.5009) | 46.2675 |
| Verification (Aggr.) | gpt4o | few-shot(5) | 0.0000 (±0.0246) | 0.4300 (±0.0590) | 0.0000 (±0.1005) | -0.0200 (±0.0141) | -0.0333 (±0.0401) | -0.1023 (±0.0443) | 0.0457 |
| | Llama3 | few-shot(5) | -0.2700 (±0.0468) | 0.0900 (±0.0668) | -0.5200 (±0.0858) | -0.6600 (±0.0476) | -0.7303 (±0.0473) | -0.5056 (±0.0549) | -0.4327 |
| | Llama3 | finetune | -0.1300 (±0.0485) | -0.2000 (±0.0402) | -0.3300 (±0.0900) | -0.7400 (±0.0441) | -0.8741 (±0.0343) | -0.9342 (±0.0262) | -0.5347 |
| PPL | Llama3 | few-shot(5) | 1.1967 (±0.0028) | 1.1174 (±0.0015) | 1.1527 (±0.0021) | 1.1119 (±0.0017) | 1.4486 (±0.0054) | 1.4975 (±0.0051) | 1.2541 |
| | Llama3 | finetune | 1.0004 (±0.0002) | 1.1090 (±0.0014) | 1.0010 (±0.0006) | 1.1072 (±0.0011) | 1.2944 (±0.0053) | 1.3299 (±0.0055) | 1.1403 |

for more details. We also performed fine-tuning on the quantum algorithm design task, which is included in Appendix C.2.2.

## 5 Conclusions and Future Work

In this paper, we propose **QCircuitBench**, the first comprehensive, structured quantum algorithm dataset and quantum code generation benchmark for AI models. We highlight the main takeaways as follows:

- **Novelty and Impact**: QCircuitBench introduces the first comprehensive benchmark specifically tailored for LLM-driven quantum algorithm design, addressing a critical research gap at the intersection of quantum computing and AI, with implications for both communities.

- **Dataset Design**: The dataset defines a general framework which formulates the key components of quantum algorithm design in a manner suitable for LLMs, with careful considerations for various subtle challenges. Its code generation perspective, modular and extensible structure, automatic verification functions, and broad coverage of diverse quantum algorithms together establish a rigorous benchmark for evaluating the capabilities of LLMs in quantum algorithm design. The framework is scalable to larger qubit numbers and can incorporate more advanced algorithms that are polynomially implementable.

- **Extensive experiments**: Our experiments show that QCircuitBench poses significant challenges to state-of-the-art LLMs, making it a valuable benchmark for AI models. We identify three predominant error types: (1) *improvisation errors*, where models hallucinate or deviate from task constraints such as syntax or required structure; (2) *counting errors*, especially in qubit or gate indices; and (3) *data contamination*, likely stemming from prior exposure to known implementations. Despite these challenges, fine-tuning experiments demonstrate early promise in helping models internalize structural patterns. These insights position QCircuitBench not only as a benchmark, but also as a path for future model development.

- **Clarity and Reproducibility**: We release the dataset in a fully reproducible format, accompanied by detailed documentation, executable code and transparent benchmarking protocols. This ensures that researchers can effortlessly use, validate, and extend the benchmark, with the aim of contributing to open scientific efforts.

Our work leaves several open questions for future investigation:

- While QCircuitBench is designed as a benchmark, its structure also supports training objectives. It is of general interest to extend benchmarking to training, which will help LLMs better maneuver quantum algorithm design. We have implemented advanced algorithms such as the Generalized Simon's Problem, but further coverage of more advanced algorithms will make it more impactful.

- As quantum algorithms differ fundamentally from classical ones, novel fine-tuning methods and model architectures may be required. These could focus on quantum-specific inductive biases, hybrid neural-symbolic pipelines, or even autonomous quantum algorithm discovery via LLMs, potentially enabling generative models to contribute novel quantum methods.

## Acknowledgements

We would like to thank Tianyi Chen for great help with experiments. We would like to thank Zihang Li, Kecheng Liu, Ruihua Liu, Shaozhen Liu, Zhixin Song, Kai Wang, Mengfan Yuan, Lu Zhang, Shuo Zhang, Xingyu Zhao for participating in the human study. Special thanks to Hengli Li and Zihao Wang for their valuable discussions.

This work was supported by the National Natural Science Foundation of China (Grant Number 92365117).

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

# A  Details of QCircuitBench

The QCircuitBench Dataset, along with detailed documentation, benchmarking scripts, and Croissant metadata, is available on GitHub (`https://github.com/EstelYang/QCircuitBench`) and Harvard Dataverse (`https://doi.org/10.7910/DVN/ZC4PNI`). We recommend visiting the GitHub repository for the latest updates.

The following structure represents the default Qiskit-based implementation of QCircuitBench. The separate directory named *Cirq_Version* mirrors this structure to provide equivalent implementations using Cirq.

```
QCircuitBench
├── Algorithm Design ........... All data and code for the quantum algorithm design task
│   ├── Quantum Computing ................... Universal quantum computing algorithms
│   ├── Quantum Information ......................... Quantum information protocols
│   └── Variational Quantum Algorithms .............. Variational quantum algorithms
├── Oracle Construction ...................... All data for the oracle construction task
│   ├── Quantum Logic Synthesis ................. Textbook-level and advanced oracles
│   └── Problem Encoding ....................... Oracles encoding application scenarios
├── Random Circuits ..................... All data for the random circuit synthesis task
│   ├── Clifford .............................. Random circuits with the Clifford gate set
│   └── Universal ........................... Random circuits with the universal gate set
└── Cirq Version ...................................... Implementation using Cirq
```

In each subdirectory, there is a folder for each specific algorithm. For instance, the folder structure for Simon's algorithm is as follows:

```
Algorithm Design
└── Quantum Computing
    └── simon ........................................ All data for the Simon's Problem
        ├── simon-dataset.py ................................... Dataset creation script
        ├── simon-generation.py ............................... Qiskit generation code
        ├── simon-post-processing.py ........................ Post-processing function
        ├── simon-utils.py ............................ Utility functions for verification
        ├── simon-verification.py .............................. Verification function
        ├── simon-description.txt .............................. Problem description
        ├── simon-verification.txt ............... Verification results of the data points
        ├── full circuit .............................. Raw data of quantum circuits
        │   ├── simon-n2
        │   │   └── simon-n2-s11-k11.qasm .............. Full circuit for a concrete setting
        │   ├── simon-n3
        │   │   ├── simon-n3-s011-k001.qasm
        │   │   └── simon-n3-s011-k101.qasm
        │   └── ...
        ├── test oracle ................................... Extracted oracle definitions
        │   ├── n2
        │   │   └── trial1
        │   │       ├── oracle.inc ......................... Oracle definition as a .inc file
        │   │       └── oracle-info.txt ........... Oracle information (such as key strings)
        │   ├── n3
        │   │   ├── trial1
        │   │   │   ├── oracle.inc
        │   │   │   └── oracle-info.txt
        │   │   └── trial2
        │   │       ├── oracle.inc
        │   │       └── oracle-info.txt
        │   └── ...
        ├── simon-n2.qasm ......................... Algorithm circuit for model output
        ├── simon-n3.qasm
        └── ...
```

We expect to extend QCircuitBench under this general structure.

## A.1 Format

In this subsection, we provide concrete examples to illustrate the different components of QCircuitBench. We use the case of Simon's Problem throughout the demonstration to achieve better consistency. For further details, please check the code repository.

1. **Problem Description:** this is the carefully hand-crafted description of the task in natural language and latex math formulas. The description is provided as one template for each algorithm, and the concrete settings (such as the qubit number) are replaced when creating the data points in json. The file is named as "{algorithm_name}_description.txt".

> **Problem Description Template for Simon's Problem**
>
> Given a black box function $f : \{0,1\}^n \longmapsto \{0,1\}^n$. The function is guaranteed to be a two-to-one mapping according to a secret string $s \in \{0,1\}^n, s \neq 0^n$, where given $x_1 \neq x_2, f(x_1) = f(x_2) \iff x_1 \oplus x_2 = s$. Please design a quantum algorithm to find $s$. The function is provided as a black-box oracle gate named "Oracle" in the "oracle.inc" file which operates as $O_f |x\rangle |y\rangle = |x\rangle |y \oplus f(x)\rangle$. The input qubits $|x\rangle$ are indexed from 0 to $n-1$, and the output qubits $|f(x)\rangle$ are indexed from $n$ to $2n-1$. Please provide the following components for the algorithm design with $n =$ {qubit number}:
> 1. the corresponding quantum circuit implementation with {QASM / Qiskit}. 2. the post-processing code run_and_analyze(circuit, aer_sim) in python which simulates the circuit (QuantumCircuit) with aer_sim (AerSimulator) and returns the secret string $s$ according to the simulation results.

2. **Generation Code:** one general Qiskit code to create quantum circuits of different settings. Note that the oracle for the problem is provided as a black-box gate "oracle" here. This code is used to generate the raw data, but can also be used as a testing benchmark for writing Qiskit codes. The file is named as "{algorithm_name}_generation.py".

```python
from qiskit import QuantumCircuit

def simon_algorithm(n, oracle):
    """Generates a Simon algorithm circuit.

    Parameters:
    - n (int): number of qubits
    - s (str): the secret string of length n

    Returns:
    - QuantumCircuit: the Simon algorithm circuit
    """
    # Create a quantum circuit on 2n qubits
    simon_circuit = QuantumCircuit(2 * n, n)

    # Initialize the first register to the |+> state
    simon_circuit.h(range(n))

    # Append the Simon's oracle
    simon_circuit.append(oracle, range(2 * n))

    # Apply a H-gate to the first register
    simon_circuit.h(range(n))

    # Measure the first register
    simon_circuit.measure(range(n), range(n))

    return simon_circuit
```

Listing 1: Qiskit generation code for Simon's algorithm.

3. **Algorithm Circuit:** the OpenQASM 3.0 format file storing the quantum circuit in gate level for each specific setting. Note that the explicit construction of "Oracle" is provided separately in "oracle.inc" file, which guarantees the usage of oracle in a black-box way. This filed is named as "{algorithm_name}_n{qubit_number}.qasm".

```
OPENQASM 3.0;
include "stdgates.inc";
include "oracle.inc";
bit[3] c;
qubit[6] q;
h q[0];
h q[1];
h q[2];
Oracle q[0], q[1], q[2], q[3], q[4], q[5];
h q[0];
h q[1];
h q[2];
c[0] = measure q[0];
c[1] = measure q[1];
c[2] = measure q[2];
```

Listing 2: OpenQASM 3.0 Code for Simon's algorithm with $n = 3$.

4. **Post-Processing Function:** this function simulates the quantum circuit and derives the final answer to the problem. The file is named as "{algorithm_name}_post_processing.py".

```python
from sympy import Matrix
import numpy as np
from qiskit import transpile

def mod2(x):
    return x.as_numer_denom()[0] % 2

def solve_equation(string_list):
    """
    A^T | I
    after the row echelon reduction, we can get the basis of the
        ↪ nullspace of A in I
    since we just need the string in binary form, so we can just
        ↪ use the basis
    if row == n-1 --> only one
    if row < n-1 --> get the first one (maybe correct or wrong)
    """
    M = Matrix(string_list).T

    # Augmented   : M | I
    M_I = Matrix(np.hstack([M, np.eye(M.shape[0], dtype=int)]))

    # RREF row echelon form , indices of the pivot columns
    # If x % 2 = 0, it will not be chosen as pivot (modulo 2)
    M_I_rref = M_I.rref(iszerofunc=lambda x: x % 2 == 0)

    # Modulo 2
    M_I_final = M_I_rref[0].applyfunc(mod2)

    # Non-Trivial solution
    if all(value == 0 for value in M_I_final[-1, : M.shape[1]]):
        result_s = "".join(str(c) for c in M_I_final[-1, M.shape[1]
            ↪ :])

    # Trivial solution
    else:
        result_s = "0" * M.shape[0]
```

```
        return result_s

def run_and_analyze(circuit, aer_sim):
    n = circuit.num_qubits // 2
    circ = transpile(circuit, aer_sim)
    results = aer_sim.run(circ, shots=n).result()
    counts = results.get_counts()
    equations = []
    for result, count in counts.items():
        if result != "0" * n:  # We don't use all 0 string
            y = [int(bit) for bit in result]
            equations.append(y)
    if len(equations) == 0:
        prediction = "0" * n
    else:
        prediction = solve_equation(equations)
    return prediction
```

Listing 3: Post-processing code for Simon's algorithm.

5. **Oracle / Gate Definition:** this .inc file provides the definitions of composite gates or oracles. The file is named "customgates.inc" for oracle construction tasks and "oracle.inc" for algorithm design tasks.

```
gate Oracle _gate_q_0, _gate_q_1, _gate_q_2, _gate_q_3, _gate_q_4,
    ↪ _gate_q_5 {
  cx _gate_q_0, _gate_q_3;
  cx _gate_q_1, _gate_q_4;
  cx _gate_q_2, _gate_q_5;
  cx _gate_q_2, _gate_q_5;
  x _gate_q_3;
}
```

Listing 4: One test case oracle for Simon's algorithm with $n = 3$.

For algorithm design tasks, this .inc file is accompanied with an "oracle_info.txt" file to describe the encoded information of the oracle. This helps the verification function to check the correctness of the derived answer by the model. The above test case is equipped with the following information text:

> oracle_info.txt for Simon's Problem with qubit number 3 and test case 2.
>
> Secret string: 100
> Key string: 001

6. **Verification Function:** the function to evaluate the output with grammar validation and test cases verification. The file is named as "{algorithm_name}_verification.py".

```
from simon_utils import *

def check_model(qasm_string, code_string, n, t=1):
    """Check the Bernstein-Vazirani model."""
    # Verify the syntax of the QASM code with the first test case
        ↪ oracle
    qasm_syntax = -1
    code_syntax = -1
    result_score = 0.
    gate_count_ratio = float('nan')
    shot_ratio = float('nan')
    time_ratio = float('nan')
```

```python
    # QASM syntax verification
    ......
    full_qasm = plug_in_oracle(qasm_string, oracle_def)
    circuit = verify_qasm_syntax(full_qasm)
    if circuit is None:
        print("QASM syntax error detected, using ground truth.")
        qasm_syntax = 0
        dire_gt = "lm_eval/tasks/QuantumAlgorithm/Circuit/simon"
        with open(f"{dire_gt}/simon_n{n}.qasm", "r") as file:   #
            ↪  model generated is wrong, use ground truth
            qasm_string = file.read()
    else:
        qasm_syntax = 1

    # Post-Processing code verification
    try:
        local_vars = {}
        code = '\n'.join([line for line in code_string.splitlines()
            ↪   if not line.strip().startswith('from qiskit') and '
            ↪  import qiskit' not in line])
        code_string = code.replace("def run_and_analyze(circuit,
            ↪  aer_sim):\n", "def run_and_analyze(circuit, aer_sim)
            ↪  :\n    circuit = transpile(circuit, aer_sim)\n", 1)
        exec(code_string, globals(), local_vars)
        run_and_analyze_func = local_vars['run_and_analyze']
        code_syntax = 1
        print("Post-processing code loaded successfully.")
    except Exception as e:
        print(f"Post-processing syntax error: {e}\nusing ground
            ↪  truth.")
        run_and_analyze_func = ground_truth_run_and_analyze
        code_syntax = 0

    if qasm_syntax == 1 and code_syntax == 1:  # Only check the
        ↪  case : model has at least one correct
        gate_count_ratio, shot_ratio, time_ratio = efficiency_check
            ↪  (qasm_string, dire_gt, code_string,
            ↪  run_and_analyze_func, ground_truth_run_and_analyze,
            ↪  dire, n)
        try:
            result_score = execute_test_cases(qasm_string,
                ↪  run_and_analyze_func, n)
        except Exception as e:
            print(f"Post-processing running-time error: {e}")
            code_syntax = 0

    return qasm_syntax, code_syntax, result_score, gate_count_ratio
        ↪ , shot_ratio, time_ratio
```

Listing 5: Verification function for Simon's algorithm.

This verification function is accompanied with an "{algorithm_name}_utils.py" file to provide necessary utility functions.

```python
from qiskit.qasm3 import loads
from qiskit_aer import AerSimulator
import re

def print_and_save(message, text):
    print(message)
    text.append(message)

def plug_in_oracle(qasm_code, oracle_def):
```

```
    """Plug-in the oracle definition into the QASM code."""
    oracle_pos = qasm_code.find('include "oracle.inc";')
    if oracle_pos == -1:
        raise ValueError("Oracle include statement not found in the
            ↪   file")
    full_qasm = (
        qasm_code[:oracle_pos]
        + oracle_def
        + qasm_code[oracle_pos + len('include "oracle.inc";') :]
    )
    return full_qasm

def verify_qasm_syntax(output):
    """Verify the syntax of the output and return the corresponding
        ↪   QuantumCircuit (if it is valid)."""
    assert isinstance(output, str)
    try:
        # Parse the OpenQASM 3.0 code
        circuit = loads(output)
        print(
            "    The OpenQASM 3.0 code is valid and has been
                ↪ successfully loaded as a QuantumCircuit."
        )
        return circuit
    except Exception as e:
        print(f"   Error: The OpenQASM 3.0 code is not valid.
            ↪ Details: {e}")
        return None
```

Listing 6: Utility functions for verification of Simon's algorithm.

7. **Dataset Creation Script:** this script involves all the code necessary to create the data points from scratch. The file is named as "{algorithm_name}_dataset.py". The main function looks like this:

```
def main():
    parser = argparse.ArgumentParser()
    parser.add_argument(
        "-f",
        "--func",
        choices=["qasm", "json", "gate", "check"],
        help="The function to call: generate qasm circuit, json
            ↪ dataset or extract gate definition.",
    )
    args = parser.parse_args()
    if args.func == "qasm":
        generate_circuit_qasm()
    elif args.func == "json":
        generate_dataset_json()
    elif args.func == "gate":
        extract_gate_definition()
    elif args.func == "check":
        check_dataset()
```

Listing 7: Main function of the dataset script for Simon's algorithm.

Here the "generate_circuit_qasm()" function generates the raw data of quantum circuits in Open-QASM 3.0 format where the algorithm circuit and the oracle definition are blended, then "extract_gate_definition()" function extracts the definition of oracles and formulates the algorithm circuits into the format suitable for model output. The "check_dataset()" function is used to check the correctness of the created data points and "generate_dataset_json()" function to combine the data into json format for easy integration with the benchmarking pipeline.

## A.2 Visualization of reference circuits

In this subsection, we present visualizations of reference quantum circuits for each of the 25 algorithms across all three task suites.

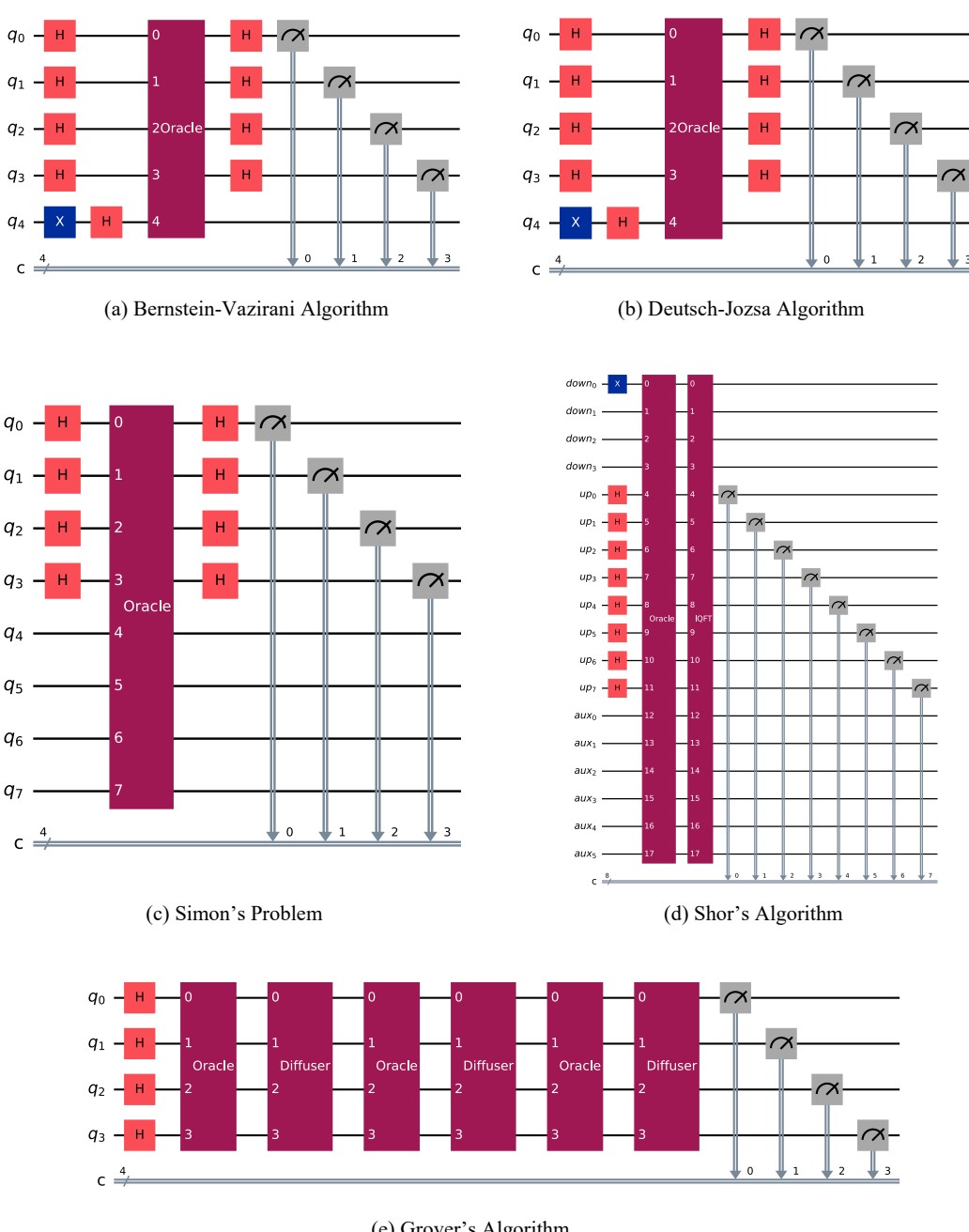

(a) Bernstein-Vazirani Algorithm

(b) Deutsch-Jozsa Algorithm

(c) Simon's Problem

(d) Shor's Algorithm

(e) Grover's Algorithm

Figure 4: Visualization of Reference Quantum Circuits for Quantum Algorithm Design Task (I).

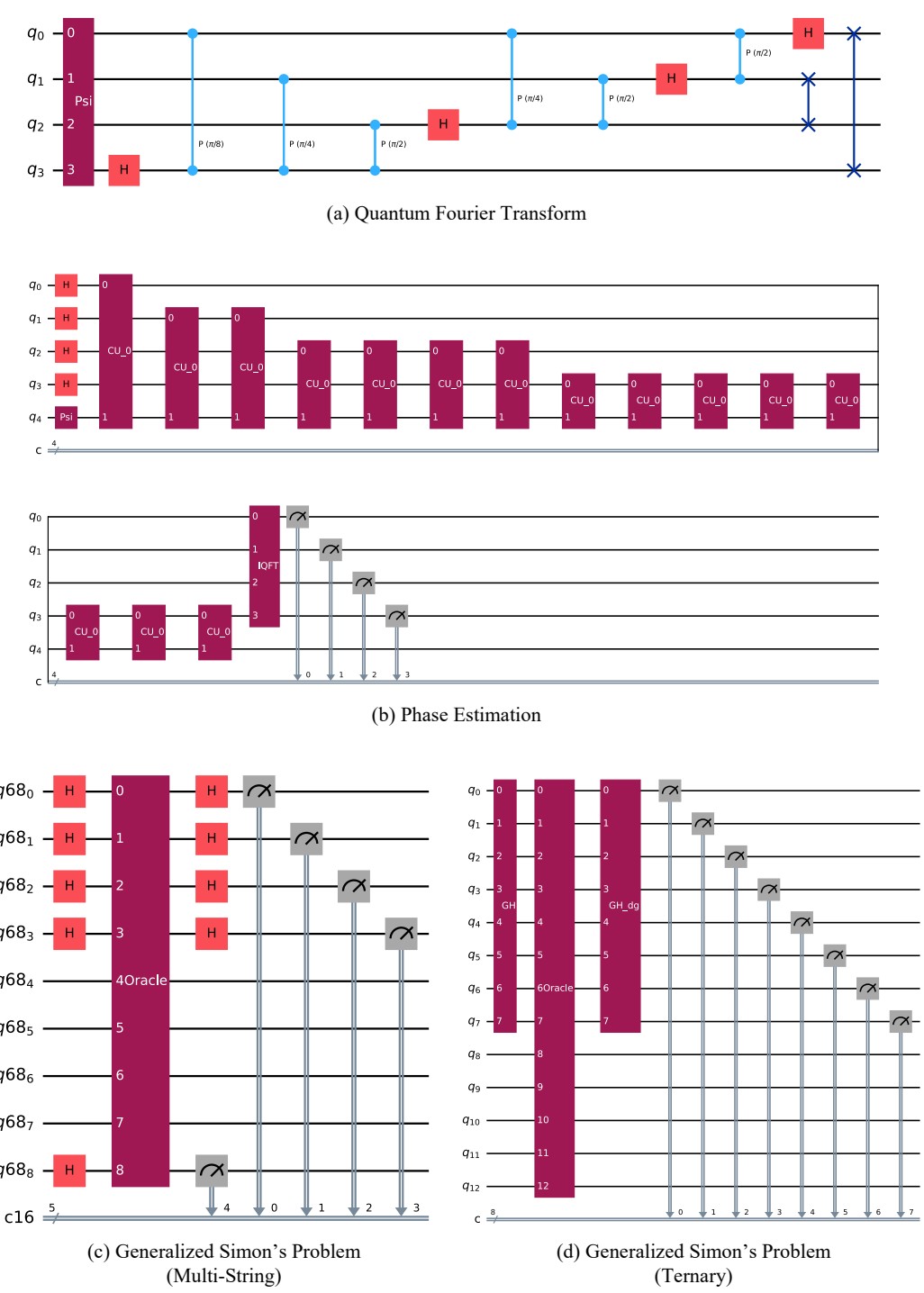

(a) Quantum Fourier Transform

(b) Phase Estimation

(c) Generalized Simon's Problem
(Multi-String)

(d) Generalized Simon's Problem
(Ternary)

Figure 5: Visualization of Reference Quantum Circuits for Quantum Algorithm Design Task (II).

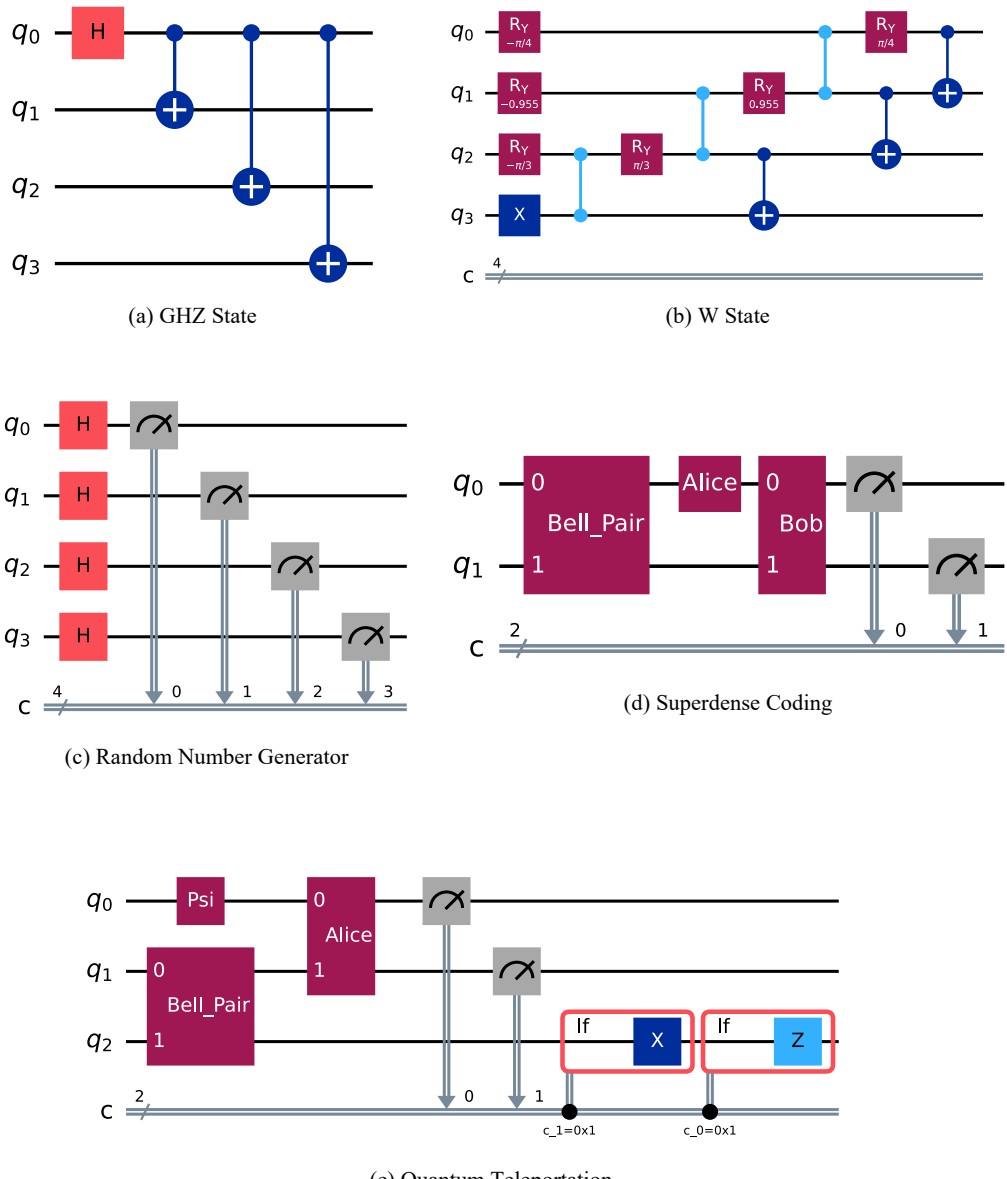

(a) GHZ State

(b) W State

(c) Random Number Generator

(d) Superdense Coding

(e) Quantum Teleportation

Figure 6: Visualization of Reference Quantum Circuits for Quantum Algorithm Design Task (III).

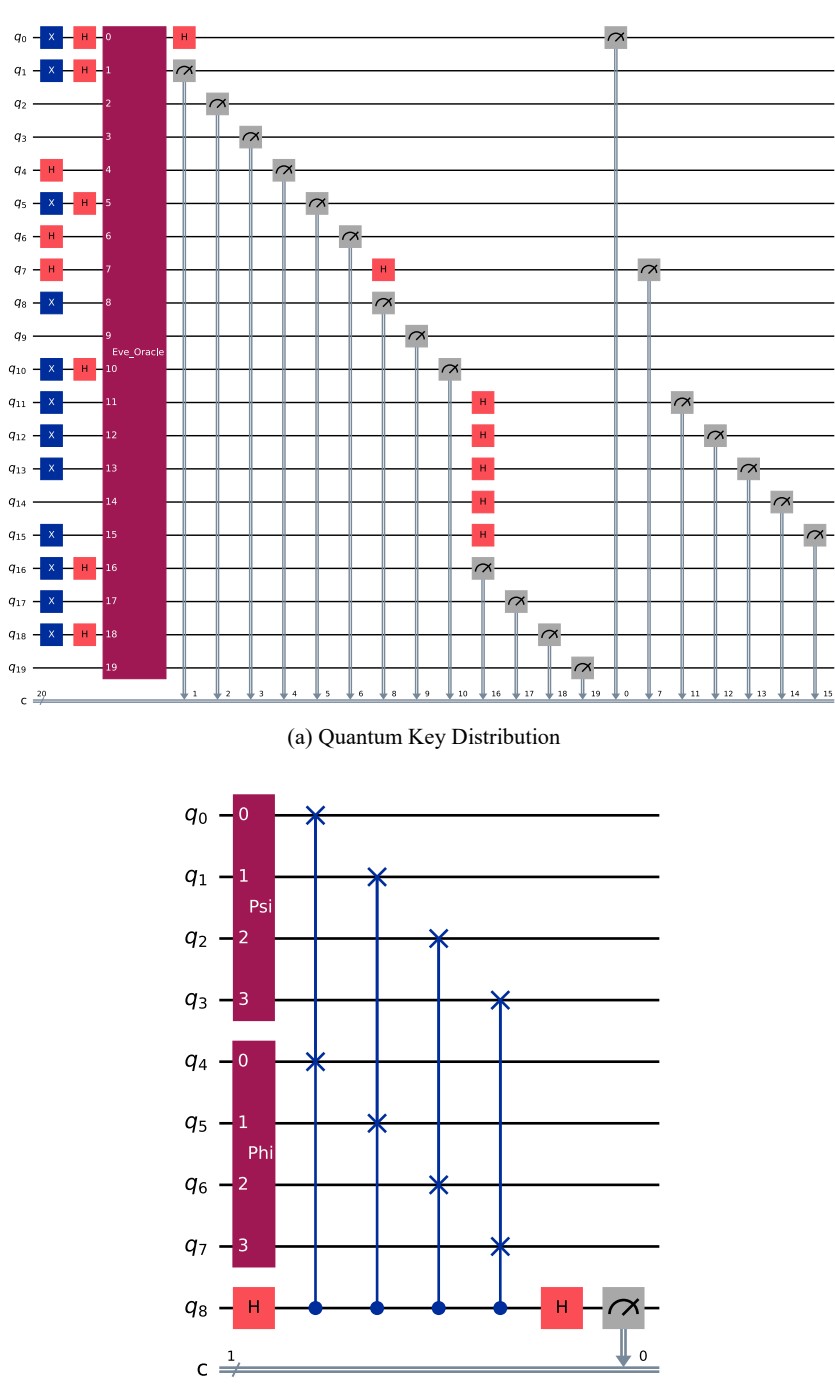

(a) Quantum Key Distribution

(b) Swap Test

Figure 7: Visualization of Reference Quantum Circuits for Quantum Algorithm Design Task (IV).

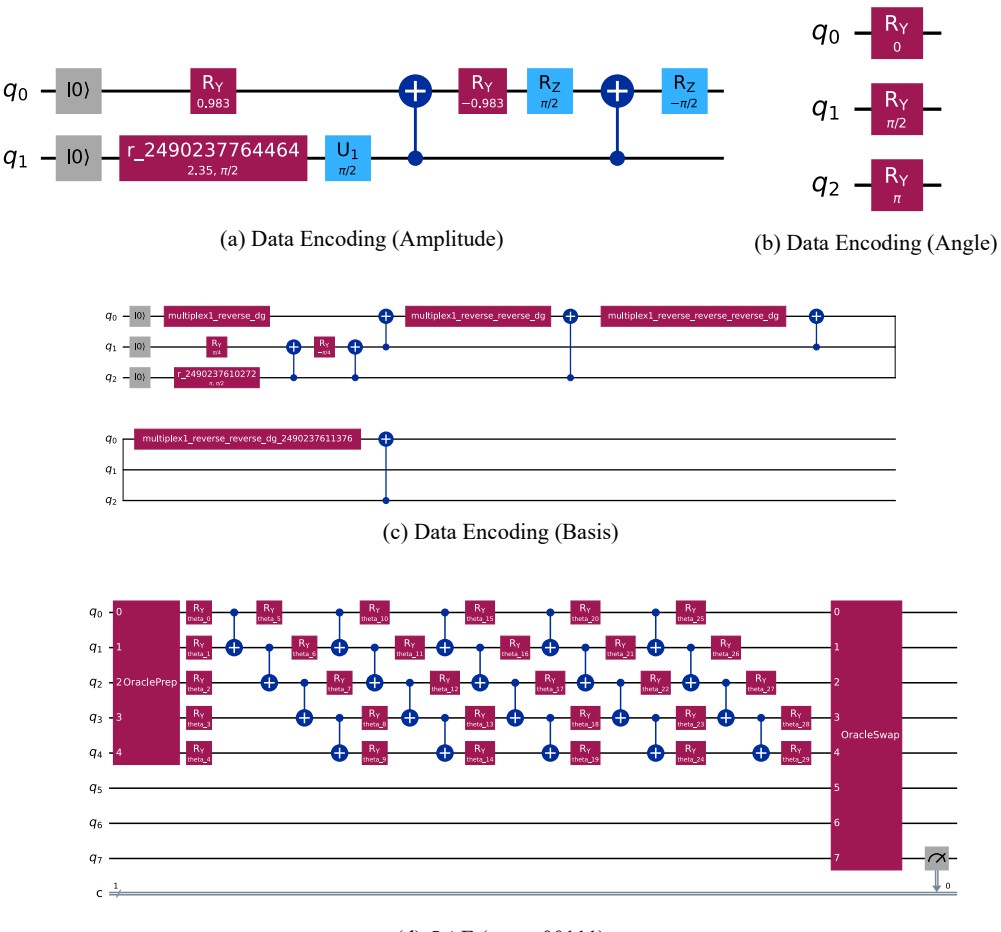

(a) Data Encoding (Amplitude)

(b) Data Encoding (Angle)

(c) Data Encoding (Basis)

(d) QAE (state=00111)

Figure 8: Visualization of Reference Quantum Circuits for Quantum Algorithm Design Task (V).

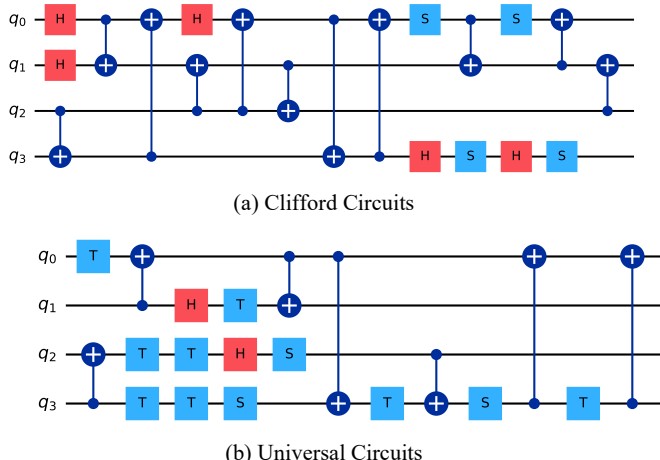

(a) Clifford Circuits

(b) Universal Circuits

Figure 9: Visualization of Reference Quantum Circuits for Random Circuit Synthesis Task.

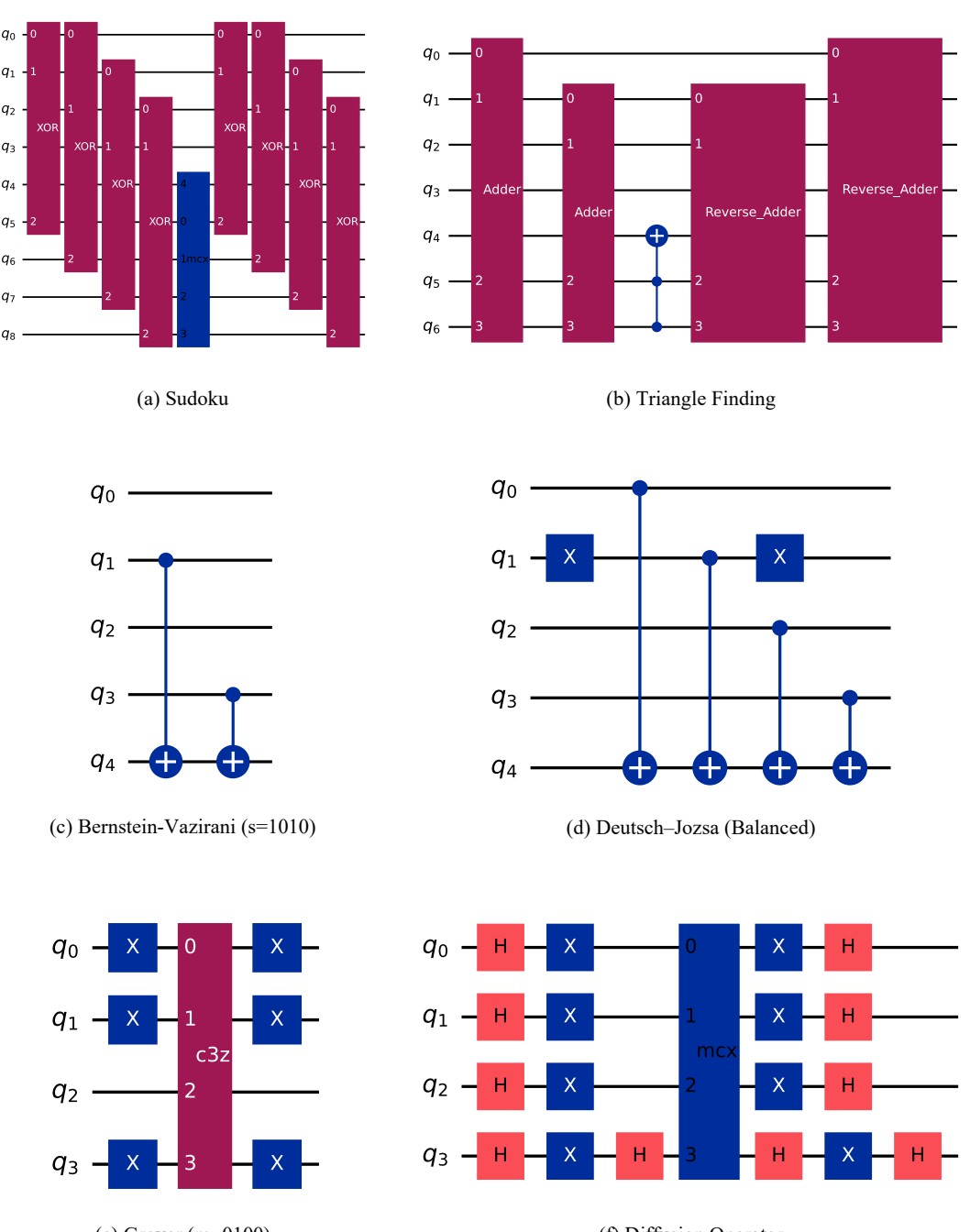

(a) Sudoku

(b) Triangle Finding

(c) Bernstein-Vazirani (s=1010)

(d) Deutsch–Jozsa (Balanced)

(e) Grover (m=0100)

(f) Diffusion Operator

Figure 10: Visualization of Reference Quantum Circuits for Oracle Construction Task (I).

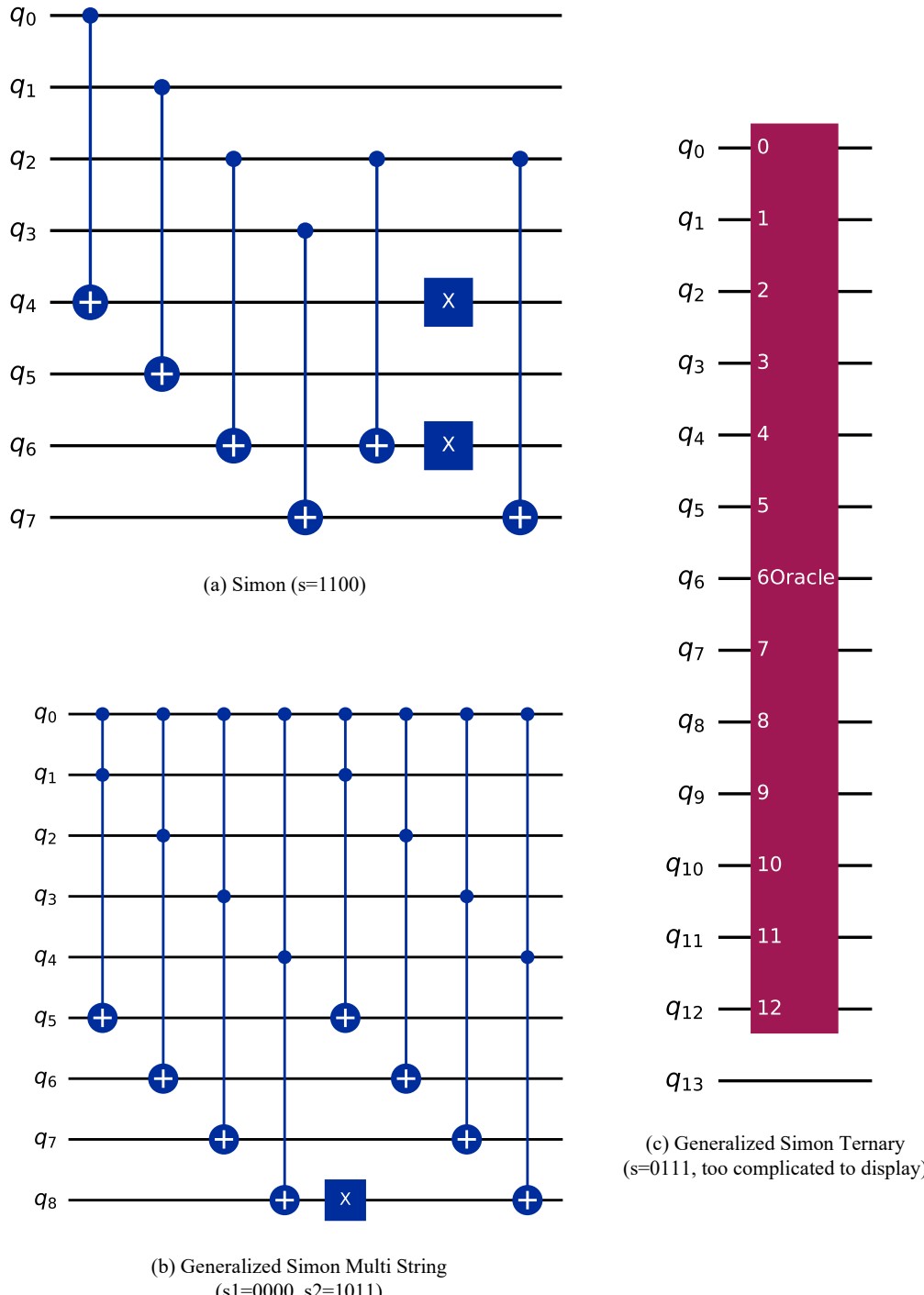

(a) Simon (s=1100)

(b) Generalized Simon Multi String
(s1=0000, s2=1011)

(c) Generalized Simon Ternary
(s=0111, too complicated to display)

Figure 11: Visualization of Reference Quantum Circuits for Oracle Construction Task (II).

## A.3 Human study

Here we present the instructions for the human study reported in Section 4.1.

**Quantum Programming Assessment**

📝 **Invitation to Participate in Quantum Programming Assessment**
    We sincerely invite you to participate in our **Quantum Computing Programming Assessment**. This assessment is based on the **QCircuitBench** dataset, which has been officially accepted to the **NeurIPS 2025 Datasets and Benchmarks Track**. It focuses on three categories of tasks from **QCircuitBench**: **Quantum Algorithm Design**, **Oracle Construction**, and **Random Circuit Synthesis**.
    **Contact Information of us: ypyangrui@pku.edu.cn**
    **Please make sure to include your contact information** so that we can conduct proper data collection and analysis.
    We greatly appreciate your participation and contribution to this study!

**Participation Details**
- Each participant will receive **3 randomly selected questions** from QCircuitBench.
- Different participants will be presented with different sets of questions.

**Rules**

**Open-Book**: You may refer to publicly available textbooks, lecture notes, research papers, or online resources. Possible references include: IBM Quantum Documentation (IBM Quantum Documentation); OpenQASM 3.0 Specification (OpenQASM 3.0 Specification — OpenQASM 3.0 Specification documentation) etc.
**Independent Work**: Plagiarism, copying existing answers, or directly using AI-generated content is **strictly prohibited**. Participants are expected to think independently and complete the tasks on their own.

**Submission Requirements**

**Circuit Implementation:**

Must be written in **OpenQASM 3.0**.

**Post-Processing Code:**

Must be written in **Python**, and may utilize the **Qiskit** library (recommended version: **Qiskit 0.46.0**). Other helper functions and libraries may be used freely.

Figure 12: Instructions for human study.

## A.4 Discussion of more tasks

**Problem Encoding.**    In Section 3.1.1, we mentioned another category of oracle construction tasks referred to as "Problem Encoding", which involves applying quantum algorithms, such as Grover's algorithm, to solve practical problems such as SAT and triangle finding. The crux of this process is encoding the problem constraints into Grover's oracle, thereby making this a type of oracle construction task. Unlike quantum logic synthesis, which encodes an explicit function $f(x)$ as a unitary operator $U_f$, this task involves converting the constraints of a particular problem into the required oracle form. We provide implementations of several concrete problems in this directory as demonstrations and will include more applications in future work.

**Quantum Information Protocols.**    In Section 3.1.2, we have also implemented three important quantum information protocols: Quantum Teleportation, Superdense Coding, and Quantum Key Distribution (BB84). A brief introduction to these protocols can be found in Appendix B. We did not include the experiments for these protocols as they involve communication between two parties, which is challenging to characterize with a single OpenQASM 3.0 file. We recommend revising the post-processing function as a general classical function to schedule the communication and processing between different parties specifically for these protocols. The fundamental quantum circuits and processing codes are provided in the repository.

# B  Preliminaries for Quantum Computing and Quantum Information

In this section, we will introduce necessary backgrounds for quantum computing related to this paper. A more detailed introduction to quantum computing can be found in the standard textbook by Nielsen and Chuang [2000].

**Quantum States.**  In classical computing, the basic unit is a bit. In quantum computing, the basic unit is a *qubit*. Mathematically, $n$ ($n \in \mathbb{N}$) qubits forms an $N$-dimensional Hilbert space for $N = 2^n$. An $n$-qubit *quantum state* $|\phi\rangle$ can be written as

$$|\phi\rangle = \sum_{i=0}^{N-1} \alpha_i |i\rangle, \quad \text{where} \quad \sum_{i=0}^{N-1} |\alpha_i|^2 = 1. \tag{1}$$

Here $|\cdot\rangle$ represents a column vector, also known as a ket state. The tensor product of two quantum states $|\phi_1\rangle = \sum_{i=0}^{N-1} \alpha_i |i\rangle$ and $|\phi_2\rangle = \sum_{j=0}^{M-1} \beta_j |j\rangle$ with $M = 2^m$, $m \in \mathbb{N}$ is defined as

$$|\phi_1\rangle \otimes |\phi_2\rangle = \sum_{i=0}^{N-1} \sum_{j=0}^{M-1} \alpha_i \beta_j |i, j\rangle, \tag{2}$$

where $|i, j\rangle$ is an $(n + m)$-qubit state with first $n$ qubits being the state $|i\rangle$ and the last $m$ qubits being the state $|j\rangle$. When there is no ambiguity, $|\phi_1\rangle \otimes |\phi_2\rangle$ can be abbreviated as $|\phi_1\rangle |\phi_2\rangle$.

**Quantum Oracles.**  To study a Boolean function $f \colon \{0, 1\}^n \to \{0, 1\}^m$, we need to gain its access. Classically, a standard setting is to being able to *query* the function, in the sense that if we input an $x \in \{0, 1\}^n$, we will get the output $f(x) \in \{0, 1\}^m$. In quantum computing, the counterpart is a quantum query, which is instantiated by a *quantum oracle*. Specifically, the function $f$ is encoded as an oracle $U_f$ such that for any $x \in \{0, 1\}^n$, $z \in \{0, 1\}^m$,

$$U_f |x\rangle |z\rangle = |x\rangle |z \oplus f(x)\rangle, \tag{3}$$

where $\oplus$ is the plus modulo 2. Note that a quantum query to the oracle is stronger than a classical query in the sense that the quantum query can be applied to a state in *superposition*: For an input state $\sum_i c_i |x_i\rangle |z_i\rangle$ with $\sum_i |c_i|^2 = 1$, the output state is $\sum_i c_i |x_i\rangle |z_i \oplus f(x_i)\rangle$; measuring this state gives $x_i$ and $z_i \oplus f(x_i)$ with probability $|c_i|^2$. A classical query for $x$ can be regarded as the special setting with $c_1 = 1$, $x_1 = x$, $z_1 = 0^m$, and $c_i = 0$ for all other $i$.

**Quantum Gates.**  Similar to classical computing that can stem from logic synthesis with AND, OR, and NOT, quantum computing is also composed of basic quantum gates. For instance, the Hadamard $H$ is the matrix $\frac{1}{\sqrt{2}} \begin{bmatrix} 1 & 1 \\ 1 & -1 \end{bmatrix}$, satisfying $H|0\rangle = \frac{1}{\sqrt{2}}(|0\rangle + |1\rangle)$ and $H|1\rangle = \frac{1}{\sqrt{2}}(|0\rangle - |1\rangle)$. In general, an $n$-qubit quantum gate is a unitary matrix from $\mathbb{C}^{2^n \times 2^n}$.

**Quantum Circuit Diagram.**  A quantum algorithm is composed of a series of quantum gates. By default, a quantum algorithm starts from the all-0 state $|0^n\rangle$. A quantum algorithm can be illustrated by its quantum gate diagram, drawn from left to right. The initial all-0 state is placed at the left side of the diagram. After that, whenever we apply a quantum gate, it is placed on the corresponding qubits, from left to right. At the end of the quantum gates, we need to measure and read the outputs, and these measurements are placed at the right side of the diagram. See Figure 13 for the quantum gate diagram of Simon's algorithm [Simon, 1997].

**Superdense Coding.**  Superdense coding [Bennett and Wiesner, 1992] is a quantum communication protocol that allows Alice to transmit two classical bits of information to Bob by sending only one qubit, given that they share a pair of entangled qubits. The protocol can be divided into five steps:

1. **Preparation:** Charlie prepares a maximally entangled Bell state, such as $|\beta_{00}\rangle = \frac{1}{\sqrt{2}}(|00\rangle + |11\rangle)$.

2. **Sharing:** Charlie sends the qubit 1 to Alice and the qubit 2 to Bob. Alice and Bob can be separated by an arbitrary distance.

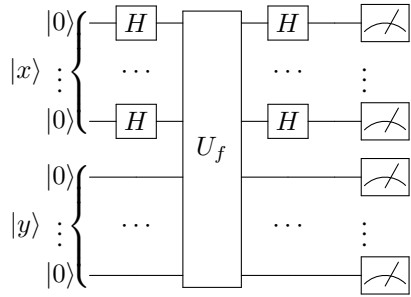

Figure 13: Quantum gate diagram of Simon's algorithm.

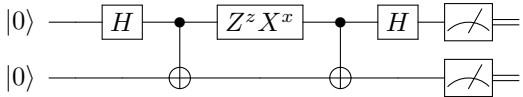

Figure 14: Quantum circuit diagram for superdense coding.

3. **Encoding:** Depending on the two classical bits $zx \in \{00, 01, 10, 11\}$ that Alice wants to send, she applies the corresponding quantum gate operation to her qubit, transforming the Bell state $|\beta_{00}\rangle$ into one of the four Bell states:

$$|\beta_{00}\rangle = \frac{1}{\sqrt{2}}(|00\rangle + |11\rangle) \ \text{ if } zx = 00$$

$$|\beta_{01}\rangle = \frac{1}{\sqrt{2}}(|01\rangle + |10\rangle) \ \text{ if } zx = 01$$

$$|\beta_{10}\rangle = \frac{1}{\sqrt{2}}(|00\rangle - |11\rangle) \ \text{ if } zx = 10$$

$$|\beta_{11}\rangle = \frac{1}{\sqrt{2}}(|01\rangle - |10\rangle) \ \text{ if } zx = 11$$

Alice achieves these transformations by applying the operation $Z^z X^x$ to her qubit, where $Z$ is the phase-flip gate, $X$ is the bit-flip gate. Specifically:

- If $zx = 00$, Alice applies $Z^0 X^0 = I$ (identity gate).
- If $zx = 01$, Alice applies $Z^0 X^1 = X$ (bit-flip gate).
- If $zx = 10$, Alice applies $Z^1 X^0 = Z$ (phase-flip gate).
- If $zx = 11$, Alice applies $Z^1 X^1 = ZX = iY$ gate.

4. **Sending:** Alice sends her qubit to Bob through a quantum channel.

5. **Decoding:** Bob applies a CNOT gate followed by a Hadamard gate to the two qubits, transforming the entangled state into the corresponding computational basis state $|zx\rangle$. By measuring the qubits, Bob obtains the two classical bits $zx$ sent by Alice.

Superdense coding exploits the properties of quantum entanglement to transmit two classical bits of information using only one qubit. The quantum circuit diagram for superdense coding is shown in Figure 14.

**Quantum Teleportation.**  Quantum teleportation [Bennett et al., 1993] is a technique for transferring quantum information from a sender (Alice) to a receiver (Bob) using shared entanglement and classical communication. The protocol can be described as follows:

1. **Preparation:** Telamon prepares a maximally entangled Bell state, such as $|\beta_{00}\rangle = \frac{1}{\sqrt{2}}(|00\rangle + |11\rangle)$.

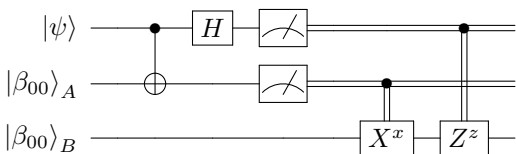

Figure 15: Quantum circuit diagram for quantum teleportation.

2. **Sharing:** Alice has qubit 1 in the state $|\psi\rangle = \alpha|0\rangle + \beta|1\rangle$, which she wants to teleport to Bob. Telamon shares qubit 2 with Alice and qubit 3 with Bob, creating the shared entangled state $|\beta_{00}\rangle_{23}$.

3. **Encoding:** Alice wants to teleport an unknown quantum state $|\psi\rangle = \alpha|0\rangle + \beta|1\rangle$ to Bob. She applies a CNOT gate to qubits 1 and 2, with qubit 1 as the control and qubit 2 as the target. Then, she applies a Hadamard gate to qubit 1. The resulting state of the three-qubit system is:

$$|\Psi\rangle = \frac{1}{2}[|\beta_{00}\rangle(\alpha|0\rangle + \beta|1\rangle) + |\beta_{01}\rangle(\alpha|1\rangle + \beta|0\rangle)$$
$$+ |\beta_{10}\rangle(\alpha|0\rangle - \beta|1\rangle) + |\beta_{11}\rangle(\alpha|1\rangle - \beta|0\rangle)].$$

4. **Measurement:** Alice measures qubits 1 and 2 in the Bell basis and obtains one of four possible outcomes: $|\beta_{00}\rangle$, $|\beta_{01}\rangle$, $|\beta_{10}\rangle$, or $|\beta_{11}\rangle$. This measurement collapses the three-qubit state into one of the following:

$$|\beta_{00}\rangle \otimes (\alpha|0\rangle + \beta|1\rangle)$$
$$|\beta_{01}\rangle \otimes (\alpha|1\rangle + \beta|0\rangle)$$
$$|\beta_{10}\rangle \otimes (\alpha|0\rangle - \beta|1\rangle)$$
$$|\beta_{11}\rangle \otimes (\alpha|1\rangle - \beta|0\rangle)$$

5. **Classical Communication:** Alice sends the result of her measurement (two classical bits) to Bob via a classical channel.

6. **Reconstruction:** Depending on the classical information received from Alice, Bob applies the operation $Z^z X^x$ to qubit 3, where $z$ and $x$ correspond to the two classical bits sent by Alice:

   - If Alice measured $|\beta_{00}\rangle$, she sends $zx = 00$, and Bob applies $Z^0 X^0 = I$ (identity operation).
   - If Alice measured $|\beta_{01}\rangle$, she sends $zx = 01$, and Bob applies $Z^0 X^1 = X$ (bit-flip).
   - If Alice measured $|\beta_{10}\rangle$, she sends $zx = 10$, and Bob applies $Z^1 X^0 = Z$ (phase-flip).
   - If Alice measured $|\beta_{11}\rangle$, she sends $zx = 11$, and Bob applies $Z^1 X^1 = ZX = iY$ (bit-flip and phase-flip).

   After applying the appropriate operation, Bob's qubit 3 will be in the state $|\psi\rangle = \alpha|0\rangle + \beta|1\rangle$, which is the original state that Alice wanted to teleport.

The quantum circuit diagram for quantum teleportation is shown in Figure 15.

**Quantum Key Distribution.** Quantum key distribution (QKD) [Bennett and Brassard, 1984] is a secure communication protocol that allows two parties, Alice and Bob, to produce a shared random secret key, which can then be used to encrypt and decrypt messages. The security of QKD is based on the fundamental principles of quantum mechanics that measuring a qubit can change its state. One of the most well-known QKD protocols is the BB84 protocol, which works as follows:

1. Alice randomly generates a bit string and chooses a random basis (X or Z) for each bit. She then encodes the bits into qubits using the chosen bases and sends them to Bob through a quantum channel.

2. Bob measures the received qubits in randomly chosen bases (X or Z) and records the results.

3. Alice and Bob communicate over a public classical channel to compare their basis choices. They keep only the bits for which their basis choices coincide and discard the rest.

4. Alice and Bob randomly select a subset of the remaining bits and compare their values. If the error rate is below a certain threshold, they conclude that no eavesdropping has occurred, and the remaining bits can be used as a secret key. If the error rate is too high, they abort the protocol, as it indicates the presence of an eavesdropper (Eve).

The security of the BB84 protocol relies on the fact that any attempt by Eve to measure the qubits during transmission will introduce detectable errors, alerting Alice and Bob to the presence of an eavesdropper.

# C   Additional Experiment Results

In this section, we provide additional experimental results. We present benchmarking experiments in Section C.1, with supplementary results for quantum algorithm design in Section C.1.1, and oracle construction and random circuit synthesis in Section C.1.2. The benchmarking results for Cirq implementation are provided in Section C.1.3. For fine-tuning experiments, we present in Section C.2, with probing experiments on random circuit synthesis in Section C.2.1, and quantum algorithm design results in Section C.2.2. In Section C.3, we demonstrate concrete cases of typical patterns observed in model outputs.

## C.1   Benchmarking Results

### C.1.1   Supplementary Analysis of Benchmarking Results for Quantum Algorithm Design

**Byte perplexity (PPL) scores.** As shown in Figure 16, byte-level perplexity in the zero-shot setting basically tracks BLEU trends, indicating consistent predictive performance across both metrics. For example, in quantum algorithm design tasks, the model finds Bernstein–Vazirani and Deutsch–Jozsa relatively easy, while struggling with Simon and Generalized Simon's Problem, reflecting their inherent differences in complexity.

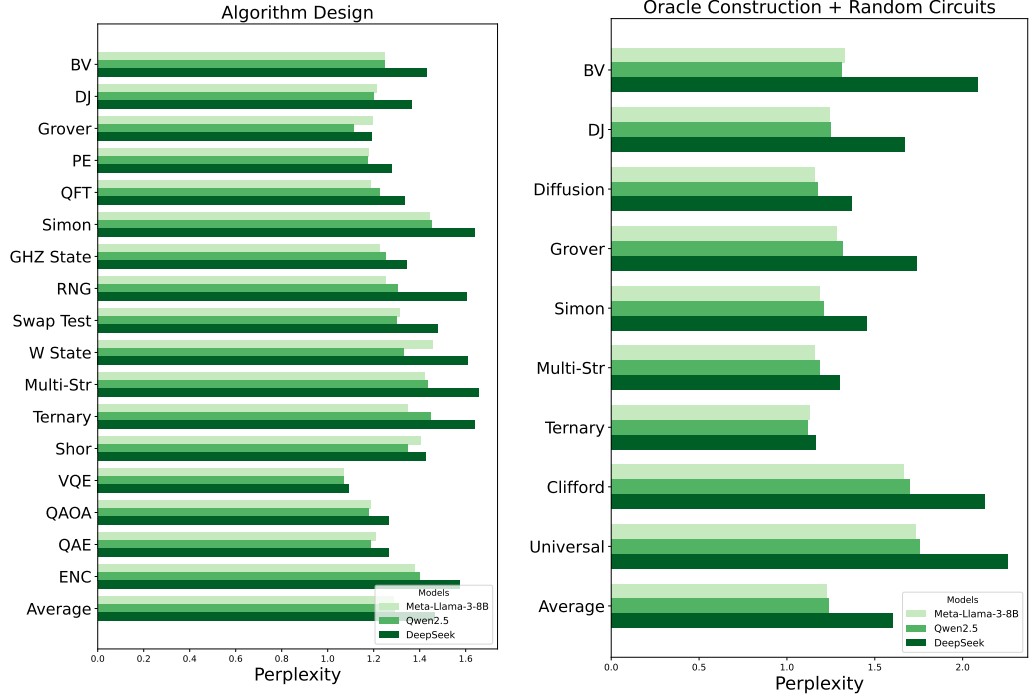

Figure 16: Benchmarking algorithm design and oracle construction in perplexity scores.

**Verification Scores.** The python syntax score is shown in Table 4. Across tasks, few-shot prompting generally improves both syntactic validity and resource efficiency, but the gains are uneven across models and algorithms. On python syntax, gpt-4o (few-shot) is near-perfect on many algorithm design tasks, while Qwen-2.5's performance remains poor or even deteriorates. Specifically, python syntax generation generally outperforms QASM syntax, probably reflecting a lack of quantum-specific data such as OpenQASM programs in the model's pre-training corpus.

Table 4: Python syntax score for benchmarking quantum algorithm design.

| Model | Shot | Bernstein Vazirani | Deutsch Jozsa | Grover | Phase Estimation | QFT | Simon | GHZ | Random Number Generator | Swap Test | W State | Generalized Simon (multi-str) | Generalized Simon (ternary) | Shor | VQE | QAOA | QAE | ENC | Avg |
|---|---|---|---|---|---|---|---|---|---|---|---|---|---|---|---|---|---|---|---|
| GPT-4o | 1 | 1.0000 (±0.0000) | 1.0000 (±0.0000) | 1.0000 (±0.0000) | 1.0000 (±0.0000) | 1.0000 (±0.0000) | 0.7500 (±0.2500) | 0.0000 (±0.0000) | 0.0000 (±0.0000) | 0.0000 (±0.0000) | 1.0000 (±0.0000) | 1.0000 (±0.0000) | 1.0000 (±0.0000) | 1.0000 (±0.0000) | 0.0000 (±0.0000) | 0.0000 (±0.0000) | 0.1667 (±0.0000) | N/A (±N/A) | 0.6422 |
| GPT-4o | 5 | 1.0000 (±0.0000) | 1.0000 (±0.0000) | 1.0000 (±0.0000) | 0.9231 (±0.0769) | 0.4615 (±0.1439) | 0.9231 (±0.0769) | 0.4286 (±0.2020) | 0.0000 (±0.0000) | 1.0000 (±0.0000) | 1.0000 (±0.0000) | 0.8462 (±0.1042) | 0.8889 (±0.1111) | 1.0000 (±0.0000) | 0.0000 (±0.0000) | 0.0000 (±0.0000) | 0.0000 (±0.0000) | N/A (±N/A) | 0.6545 |
| Llama3 | 1 | 0.3846 (±0.1404) | 0.5385 (±0.1439) | 0.6153 (±0.1404) | 0.1538 (±0.1042) | 0.9231 (±0.0769) | 0.3846 (±0.1404) | 0.5714 (±0.2020) | 0.3846 (±0.1404) | 0.7143 (±0.1253) | 0.4444 (±0.1757) | 0.2307 (±0.1216) | 0.0000 (±0.0000) | 0.3333 (±0.3333) | 0.0000 (±0.0000) | 0.0000 (±0.0000) | 0.0000 (±0.0000) | N/A (±N/A) | 0.3549 |
| Llama3 | 5 | 0.5385 (±0.1439) | 1.0000 (±0.0000) | 0.6154 (±0.1404) | 0.1538 (±0.1042) | 0.5385 (±0.1439) | 0.7692 (±0.1216) | 0.4286 (±0.2020) | 0.0769 (±0.0769) | 0.7143 (±0.1253) | 0.3333 (±0.1667) | 0.0769 (±0.0769) | 0.5556 (±0.1757) | 1.0000 (±0.0000) | 0.0000 (±0.0000) | 0.0000 (±0.0000) | 0.0000 (±0.0000) | N/A (±N/A) | 0.4251 |
| Qwen 2.5 | 1 | 0.6154 (±0.1404) | 0.3077 (±0.1332) | 0.9231 (±0.0769) | 0.7692 (±0.1216) | 0.7692 (±0.1216) | 0.6154 (±0.1404) | 0.2857 (±0.1429) | 0.7692 (±0.1216) | 0.5714 (±0.1373) | 0.6667 (±0.1667) | 0.3846 (±0.1404) | 0.6667 (±0.1667) | 0.6667 (±0.3333) | 0.0000 (±0.0000) | 0.0000 (±0.0000) | 0.2778 (±0.1086) | N/A (±N/A) | 0.5181 |
| Qwen 2.5 | 5 | 0.4615 (±0.1439) | 1.0000 (±0.0000) | 0.7692 (±0.1216) | 0.3846 (±0.1404) | 0.7692 (±0.1216) | 0.6154 (±0.1404) | 0.1429 (±0.1429) | 0.3846 (±0.1404) | 0.4286 (±0.1373) | 0.4444 (±0.1757) | 0.3846 (±0.1404) | 0.3333 (±0.1667) | 0.3333 (±0.3333) | 0.0042 (±0.0042) | 0.1042 (±0.0446) | 1.0000 (±0.0000) | N/A (±N/A) | 0.4725 |
| DeepSeek-R1 | 1 | 0.4615 (±0.1439) | 0.4615 (±0.1439) | 0.2308 (±0.1216) | 0.2308 (±0.1216) | 0.4615 (±0.1439) | 0.3077 (±0.1332) | 0.4286 (±0.2020) | 0.6923 (±0.1332) | 0.5000 (±0.1387) | 0.2222 (±0.1470) | 0.0769 (±0.0769) | 0.3333 (±0.1667) | 0.3333 (±0.3333) | 0.0000 (±0.0000) | 0.0000 (±0.0000) | 0.0000 (±0.0000) | N/A (±N/A) | 0.2788 |
| DeepSeek-R1 | 5 | 0.2308 (±0.1216) | 0.3077 (±0.1332) | 0.3846 (±0.1404) | 0.4615 (±0.1439) | 0.3077 (±0.1332) | 0.4615 (±0.1439) | 0.2857 (±0.1844) | 0.7692 (±0.1216) | 0.7143 (±0.1253) | 0.3333 (±0.1667) | 0.1538 (±0.1042) | 0.4444 (±0.1757) | 0.0000 (±0.0000) | 0.0042 (±0.0042) | 0.0000 (±0.0000) | 0.0000 (±0.0000) | N/A (±N/A) | 0.3037 |
| Human | - | 0.5000 | 1.0000 | 0.5000 | 1.0000 | 0.0000 | 0.0000 | 0.5000 | 1.0000 | 1.0000 | 0.5000 | 0.5000 | 1.0000 | 0.5000 | 0.0000 | 0.0000 | 0.0000 | N/A | 0.4705 |

**Efficiency Scores.** The three efficiency metrics, gate count ratio, shot count ratio, and time cost ratio, are presented in Table 5, Table 6, and Table 7. Llama-3 1-shot often yields bloated circuits, with its few-shot variant typically reducing gate/shot ratios. GPT-4o few-shot stays closer to reference but still shows outliers (e.g., phase estimation in gate count ratio). Numerous N/As indicate failures to produce executable outputs.[4]

Table 5: Gate count ratio for benchmarking quantum algorithm design.

| Model | Shot | Bernstein Vazirani | Deutsch Jozsa | Grover | Phase Estimation | QFT | Simon | GHZ | Random Number Generator | Swap Test | W State | Generalized Simon (multi-str) | Generalized Simon (ternary) | Shor | VQE | QAOA | QAE | ENC | Avg |
|---|---|---|---|---|---|---|---|---|---|---|---|---|---|---|---|---|---|---|---|
| GPT-4o | 1 | N/A (±N/A) | N/A (±N/A) | N/A (±N/A) | N/A (±N/A) | N/A (±N/A) | N/A (±N/A) | 0.6250 (±N/A) | 0.1250 (±N/A) | 0.0000 (±N/A) | N/A (±N/A) | N/A (±N/A) | N/A (±N/A) | N/A (±N/A) | N/A (±N/A) | N/A (±N/A) | N/A (±N/A) | 0.7898 (±0.0848) | 0.7898 |
| GPT-4o | 5 | 1.0000 (±0.0000) | 1.0000 (±0.0000) | N/A (±N/A) | 1.7050 (±0.1680) | 0.2913 (±0.1597) | 1.1715 (±0.0708) | 1.5603 (±0.3735) | 1.2381 (±0.1076) | 2.6001 (±0.0000) | 1.0000 (±0.0000) | 0.8000 (±0.1600) | N/A (±N/A) | N/A (±N/A) | N/A (±N/A) | N/A (±N/A) | N/A (±N/A) | 0.8041 (±0.0948) | 1.0570 |
| Llama3 | 1 | 2.6001 (±0.9492) | 1.8570 (±0.4962) | 1.6250 (±0.3707) | 6.5020 (±4.4051) | 1.0833 (±0.0902) | 2.6001 (±0.9492) | 1.7501 (±0.6187) | 2.6001 (±0.9492) | 1.4000 (±0.2456) | 2.2502 (±0.8897) | 4.3328 (±2.2828) | 1.6250 (±0.1972) | 3.0003 (±3.0003) | N/A (±N/A) | N/A (±N/A) | N/A (±N/A) | 0.9091 (±0.0610) | 2.5008 |
| Llama3 | 5 | 0.9352 (±0.0199) | 1.0212 (±0.0349) | 0.1070 (±0.0894) | 0.2450 (±0.0502) | 0.3637 (±0.0535) | 1.3333 (±0.0000) | 2.6589 (±0.3712) | 2.6076 (±0.2693) | 1.2067 (±0.1539) | 1.8369 (±0.4386) | 0.4134 (±0.0603) | 1.1199 (±0.1042) | N/A (±N/A) | N/A (±N/A) | N/A (±N/A) | N/A (±N/A) | 0.6336 (±0.1140) | 1.1140 |
| Qwen 2.5 | 1 | N/A (±N/A) | N/A (±N/A) | N/A (±N/A) | N/A (±N/A) | N/A (±N/A) | 1.4635 (±0.0945) | N/A (±N/A) | 1.0000 (±0.0000) | 1.1594 (±0.1075) | 0.7692 (±0.1775) | 0.3551 (±0.0477) | N/A (±N/A) | N/A (±N/A) | N/A (±N/A) | N/A (±N/A) | N/A (±N/A) | 1.0000 (±0.0000) | 0.9579 |
| Qwen 2.5 | 5 | 0.5681 (±0.1452) | 0.7149 (±0.1125) | 0.4902 (±0.2211) | N/A (±N/A) | 0.4761 (±0.1299) | N/A (±N/A) | 1.3249 (±0.1629) | 1.0000 (±0.1048) | 0.3854 (±0.3750) | N/A (±N/A) | N/A (±N/A) | N/A (±N/A) | 1.0000 (±0.0000) | 1.0000 (±0.0000) | 1.0000 (±0.0000) | N/A (±N/A) | 0.9698 (±0.0348) | 0.7899 |
| DeepSeek-R1 | 1 | N/A (±N/A) | N/A (±N/A) | N/A (±N/A) | N/A (±N/A) | N/A (±N/A) | N/A (±N/A) | N/A (±N/A) | N/A (±N/A) | 0.3200 (±0.0640) | N/A (±N/A) | N/A (±N/A) | N/A (±N/A) | N/A (±N/A) | N/A (±N/A) | N/A (±N/A) | N/A (±N/A) | 0.3706 (±0.0792) | 0.3453 |
| DeepSeek-R1 | 5 | 0.7150 (±0.1327) | N/A (±N/A) | N/A (±N/A) | N/A (±N/A) | N/A (±N/A) | N/A (±N/A) | 2.0951 (±0.0996) | 0.5238 (±0.0249) | N/A (±N/A) | 0.5310 (±0.1055) | N/A (±N/A) | N/A (±N/A) | N/A (±N/A) | N/A (±N/A) | N/A (±N/A) | N/A (±N/A) | 0.5876 (±0.1183) | 0.8905 |
| Human | - | 1.000 | 1.0000 | N/A | 0.5500 | 0.5714 | 1.0000 | 1.0000 | N/A | 1.0833 | 1.0694 | N/A | 0.3214 | N/A | N/A | N/A | N/A | 0.6667 | 0.8262 |

Table 6: Shot count ratio for benchmarking quantum algorithm design.

| Model | Shot | Bernstein Vazirani | Deutsch Jozsa | Grover | Phase Estimation | QFT | Simon | GHZ | Random Number Generator | Swap Test | W State | Generalized Simon (multi-str) | Generalized Simon (ternary) | Shor | VQE | QAOA | QAE | ENC | Avg |
|---|---|---|---|---|---|---|---|---|---|---|---|---|---|---|---|---|---|---|---|
| GPT-4o | 1 | N/A (±N/A) | N/A (±N/A) | N/A (±N/A) | N/A (±N/A) | N/A (±N/A) | N/A (±N/A) | N/A (±N/A) | N/A (±N/A) | N/A (±N/A) | N/A (±N/A) | N/A (±N/A) | N/A (±N/A) | N/A (±N/A) | N/A (±N/A) | N/A (±N/A) | N/A (±N/A) | N/A (±N/A) | N/A |
| GPT-4o | 5 | 1.0000 (±0.0000) | 1.0000 (±0.0000) | N/A (±N/A) | N/A (±N/A) | N/A (±N/A) | 0.1577 (±0.0335) | N/A (±N/A) | N/A (±N/A) | 1.0000 (±0.0000) | 0.1024 (±0.0000) | N/A (±N/A) | N/A (±N/A) | N/A (±N/A) | N/A (±N/A) | N/A (±N/A) | N/A (±N/A) | N/A (±N/A) | 0.6520 |
| Llama3 | 1 | 1.9980 (±1.9940) | N/A (±N/A) | 0.3331 (±0.1921) | N/A (±N/A) | N/A (±N/A) | 0.1998 (±0.1998) | N/A (±N/A) | 1.9980 (±1.9940) | N/A (±N/A) | N/A (±N/A) | N/A (±N/A) | N/A (±N/A) | N/A (±N/A) | N/A (±N/A) | N/A (±N/A) | N/A (±N/A) | N/A (±N/A) | 1.1322 |
| Llama3 | 5 | 3.4759 (±1.7474) | 1.0000 (±0.0000) | 0.0316 (±0.0125) | N/A (±N/A) | N/A (±N/A) | 0.3989 (±0.3969) | N/A (±N/A) | 1.1663 (±0.1945) | N/A (±N/A) | N/A (±N/A) | N/A (±N/A) | N/A (±N/A) | N/A (±N/A) | N/A (±N/A) | N/A (±N/A) | N/A (±N/A) | N/A (±N/A) | 1.2145 |
| Qwen 2.5 | 1 | 0.5456 | 0.3437 | N/A | N/A | 1.4000 | 1.4598 (±0.1411) | 2.0000 (±0.0000) | 1.0000 | 1.1594 (±0.1075) | 0.7692 (±0.1775) | 0.3551 (±0.0477) | N/A | 1.5000 | N/A | N/A | N/A | N/A | 1.0533 |
| Qwen 2.5 | 5 | 0.5680 (±0.0821) | 0.7150 (±0.1125) | 0.4902 (±0.2211) | 0.1463 (±N/A) | 0.4761 (±0.1299) | N/A | 1.3333 | 1.3249 | 0.8000 | 0.7500 (±0.3750) | 0.3854 (±0.0530) | N/A | 1.2606 | 1.0000 (±0.0000) | 1.0000 (±0.0000) | 1.0000 (±0.0000) | N/A | 0.8036 |
| DeepSeek-R1 | 1 | N/A (±N/A) | 1.2688 (±N/A) | N/A (±N/A) | N/A (±N/A) | N/A (±N/A) | N/A (±N/A) | 3.3630 | 1.0000 | 1.1000 | N/A (±N/A) | 0.3200 (±0.0640) | N/A (±N/A) | 0.3200 | N/A (±N/A) | N/A (±N/A) | N/A (±N/A) | N/A (±N/A) | 1.2286 |
| DeepSeek-R1 | 5 | 0.7152 (±0.1327) | 0.6523 (±N/A) | N/A (±N/A) | N/A (±N/A) | 0.5939 (±N/A) | N/A (±N/A) | N/A (±N/A) | 2.0951 (±0.0996) | 0.5238 (±0.0249) | N/A (±N/A) | 0.5310 (±0.1055) | N/A (±N/A) | N/A (±N/A) | N/A (±N/A) | N/A (±N/A) | N/A (±N/A) | N/A (±N/A) | 0.8519 |
| Human | - | 1.0000 | 1.0000 | N/A | N/A | N/A | 0.0500 | N/A | N/A | N/A | N/A | N/A | N/A | N/A | N/A | N/A | N/A | N/A | 0.6677 |

**Open-Book Setting** Considering that the knowledge of quantum computing might be limited in the pre-training dataset, we further evaluate model performance in an open-book setting by enabling web search (see Table 8). This allows the model to formulate a search query, retrieve results via Google, and incorporate the content from the most relevant link into its response. Notably, the semantic score is mostly lower with web search. For instance, scores dropped from 1.0000 (few-shot) to 0.3540 (web) on Bernstein–Vazirani, and from 1.0000 (few-shot) to 0.7690 (web) on Deutsch–Jozsa. This suggests that unguided retrieval may introduce noise or distract from task-specific structure. While few-shot prompting remains competitive, we believe future work in the open-book setting holds promise through the use of stronger structural priors and guided retrieval strategies to better align external information with task-specific objectives.

---

[4]For the mean value, we report N/A if none of the outputs is executable. For the standard error, we also report N/A if only one output is executable, since standard error is undefined in that case.

Table 7: Time count ratio for benchmarking quantum algorithm design.

| Model | Shot | Bernstein Vazirani | Deutsch Jozsa | Grover | Phase Estimation | QFT | Simon | GHZ | Random Number Generator | Swap Test | W State | Generalized Simon (multi-str) | Generalized Simon (ternary) | Shor | VQE | QAOA | QAE | ENC | Avg |
|---|---|---|---|---|---|---|---|---|---|---|---|---|---|---|---|---|---|---|---|
| GPT-4o | 1 | N/A (±N/A) | N/A (±N/A) | N/A (±N/A) | N/A (±N/A) | N/A (±N/A) | N/A (±N/A) | N/A (±N/A) | N/A (±N/A) | N/A (±N/A) | N/A (±N/A) | N/A (±N/A) | N/A (±N/A) | N/A (±N/A) | N/A (±N/A) | N/A (±N/A) | N/A (±N/A) | N/A (±N/A) | N/A |
| GPT-4o | 5 | 1.9854 (±0.0102) | 1.9772 (±0.0066) | N/A (±N/A) | 1.9027 (±0.1140) | 2.9622 (±2.8295) | 1.9380 (±0.0383) | N/A (±N/A) | 4.8838 (±1.6294) | 1.9677 (±0.1183) | 1.4431 (±0.2183) | N/A (±N/A) | 2.5558 (±N/A) | N/A (±N/A) | N/A (±N/A) | N/A (±N/A) | N/A (±N/A) | N/A (±N/A) | 2.4018 |
| Llama3 | 1 | 2.0149 (±N/A) | 1.6772 (±N/A) | 1.9051 (±N/A) | N/A (±N/A) | N/A (±N/A) | 1.8690 (±N/A) | N/A (±N/A) | 2.8721 (±N/A) | 1.4943 (±N/A) | 2.2403 (±N/A) | 1.8624 (±N/A) | N/A (±N/A) | N/A (±N/A) | N/A (±N/A) | N/A (±N/A) | N/A (±N/A) | N/A (±N/A) | 1.9919 |
| Llama3 | 5 | 1.9960 (±0.0524) | 2.0049 (±0.0153) | 2.4384 (±5.5055) | 2.1920 (±N/A) | 3.3379 (±2.0366) | 0.9216 (±N/A) | N/A (±N/A) | 3.2756 (±0.9467) | 2.0395 (±0.0541) | 1.3823 (±0.0539) | 1.8833 (±0.2511) | 1.9239 (±0.0648) | N/A (±N/A) | N/A (±N/A) | N/A (±N/A) | N/A (±N/A) | N/A (±N/A) | 2.1269 |
| Qwen 2.5 | 1 | N/A (±N/A) | N/A (±N/A) | N/A (±N/A) | N/A (±N/A) | N/A (±N/A) | 2.0146 (±0.1092) | N/A (±N/A) | 2.3333 (±N/A) | 1.3430 (±0.2538) | N/A (±N/A) | N/A (±N/A) | N/A (±N/A) | N/A (±N/A) | N/A (±N/A) | N/A (±N/A) | N/A (±N/A) | N/A (±N/A) | 1.8970 |
| Qwen 2.5 | 5 | 1.2136 (±0.2397) | 1.6008 (±0.2211) | 0.9677 (±N/A) | 1.0021 (±N/A) | N/A (±N/A) | 1.9147 (±N/A) | N/A (±N/A) | 4.0713 (±2.2684) | 1.0662 (±0.0587) | N/A (±N/A) | 5.2961 (±4.4214) | 1.2891 (±N/A) | N/A (±N/A) | 1.0000 (±0.0000) | 1.0000 (±0.0000) | 1.0000 (±0.0000) | N/A (±N/A) | 1.7851 |
| DeepSeek-R1 | 1 | N/A (±N/A) | 2.0155 (±N/A) | N/A (±N/A) | N/A (±N/A) | N/A (±N/A) | N/A (±N/A) | N/A (±N/A) | 2.4000 (±N/A) | 4.4643 (±N/A) | N/A (±N/A) | 1.9625 (±N/A) | N/A (±N/A) | N/A (±N/A) | N/A (±N/A) | N/A (±N/A) | N/A (±N/A) | N/A (±N/A) | 2.7106 |
| DeepSeek-R1 | 5 | 2.1455 (±0.1654) | N/A (±N/A) | N/A (±N/A) | N/A (±N/A) | N/A (±N/A) | N/A (±N/A) | N/A (±N/A) | 1.7661 (±0.1429) | N/A (±N/A) | N/A (±N/A) | 1.7763 (±0.2043) | N/A (±N/A) | N/A (±N/A) | N/A (±N/A) | N/A (±N/A) | N/A (±N/A) | N/A (±N/A) | 1.8960 |
| Human | - | 0.1081 | 0.4917 | N/A | 0.5537 | N/A | N/A | N/A | N/A | N/A | N/A | 0.9688 | N/A | N/A | N/A | N/A | N/A | N/A | 0.5306 |

Table 8: GPT-4o Web Search Results for Quantum Algorithm Design

| Metric | Bernstein Vazirani | Deutsch Jozsa | Grover | Phase Estimation | QFT | Simon | GHZ | Random Number Generator | Swap Test | W State | Generalized Simon (multi-str) | Generalized Simon (ternary) | Shor |
|---|---|---|---|---|---|---|---|---|---|---|---|---|---|
| Semantic | 0.3540 (±0.1300) | 0.7690 (±0.1220) | 0.0450 (±0.0420) | 0.0770 (±0.0520) | 0.0000 (±0.0000) | 0.0020 (±0.0020) | 0.0000 (±0.0000) | 0.0000 (±0.0000) | 0.4090 (±0.1160) | 0.0460 (±0.0380) | 0.0000 (±0.0000) | 0.0010 (±0.0010) | 0.0000 (±0.0000) |
| QASM | 0.6920 (±0.1330) | 0.8460 (±0.1040) | 0.3080 (±0.1330) | 0.7690 (±0.1220) | 0.2310 (±0.1220) | 0.5380 (±0.1440) | 0.8570 (±0.1430) | 1.0000 (±0.0000) | 0.9290 (±0.0710) | 0.4440 (±0.1760) | 0.0770 (±0.0770) | 0.1110 (±0.1110) | 0.0000 (±0.0000) |
| Code | 0.8460 (±0.1040) | 0.9230 (±0.0770) | 0.9230 (±0.0770) | 0.7690 (±0.1220) | 0.5380 (±0.1440) | 0.2310 (±0.1220) | 0.0000 (±0.0000) | 0.0000 (±0.0000) | 0.5000 (±0.1390) | 0.4440 (±0.1760) | 0.3850 (±0.1400) | 0.6670 (±0.1670) | 0.3330 (±0.3330) |
| Gate | 1.0000 (±0.0000) | 1.0000 (±0.0000) | 1.4145 (±0.2391) | 1.4453 (±0.0816) | 1.0000 (±0.0000) | 0.9170 (±N/A) | 0.6870 (±N/A) | 0.6250 (±N/A) | 1.0000 (±0.0000) | 1.2860 (±N/A) | N/A (±N/A) | 1.0830 (±N/A) | N/A (±N/A) |
| Shot | 1.1427 (±0.1631) | 1.3745 (±0.2658) | 0.0435 (±0.0197) | N/A (±N/A) | N/A (±N/A) | 0.1936 (±N/A) | N/A (±N/A) | N/A (±N/A) | 1.0000 (±0.0000) | 0.1024 (±N/A) | N/A (±N/A) | N/A (±N/A) | N/A (±N/A) |
| Time | 1.9830 (±0.0040) | 1.9970 (±0.0145) | 1.4509 (±0.3128) | 1.9302 (±0.0195) | N/A (±N/A) | 1.9470 (±0.0049) | N/A (±N/A) | 13.6940 (±N/A) | 2.0030 (±0.0045) | 1.6800 (±0.0103) | 1.5430 (±N/A) | 1.5430 (±N/A) | N/A (±N/A) |

## C.1.2 Benchmarking Results for Oracle Construction and Random Circuit Synthesis

We observe similar patterns on the oracle construction and random circuit synthesis task to the quantum algorithm design task.

- The syntax–semantic gap remains pronounced. Models frequently produce valid QASM syntax yet fail semantically, particularly on structurally complex tasks. For instance, GPT-4o achieves perfect syntax on Deutsch–Jozsa, but fails entirely in semantic score. This further reinforces the necessity of our multi-faceted verification framework.

- Few-shot learning improves results on structurally simple or template-like oracles. For example, the semantic score of Llama3 improves from 0.6667 to 1.0000 on diffusion operator and from 0.2863 to 0.6000 on Bernstein–Vazirani. However, performance remains near zero on generalized Simon and universal/Clifford circuits, underscoring the challenge of generalization.

- DeepSeek-R1 is a notable outlier. It fails completely in the 1-shot setting, both syntactically and semantically, but recovers in 5-shot prompting, achieving perfect semantic scores on Diffusion Operator and modest gains on Bernstein–Vazirani. Still, it fails on more complicated tasks.

Table 9: QASM syntax score for benchmarking oracle construction and random circuit synthesis.

| Model | Shot | Bernstein Vazirani | Deutsch Jozsa | Diffusion Operator | Grover | Simon | Generalized Simon (multi-str) | Generalized Simon (ternary) | Clifford | Universal | Avg |
|---|---|---|---|---|---|---|---|---|---|---|---|
| GPT-4o | 1 | 1.0000 (±0.0000) | 1.0000 (±0.0000) | 0.0000 (±0.0000) | 1.0000 (±0.0000) | 1.0000 (±0.0000) | 0.0167 (±0.0167) | 0.0000 (±0.0000) | 0.0000 (±0.0000) | 0.0000 (±0.0000) | 0.4463 |
| GPT-4o | 5 | 1.0000 (±0.0000) | 1.0000 (±0.0000) | 0.0000 (±0.0000) | 0.0000 (±0.0000) | 1.0000 (±0.0000) | 0.3333 (±0.3333) | 0.0000 (±0.0000) | 0.0000 (±0.0000) | 0.0000 (±0.0000) | 0.3703 |
| Llama3 | 1 | 1.0000 (±0.0000) | 0.8182 (±0.1220) | 0.6667 (±0.2108) | 0.0909 (±0.0909) | 0.8000 (±0.1333) | 0.4286 (±0.2020) | 0.0000 (±0.0000) | 0.0000 (±0.0000) | 0.0000 (±0.0000) | 0.4227 |
| Llama3 | 5 | 1.0000 (±0.0000) | 1.0000 (±0.0000) | 1.0000 (±0.0000) | 0.0000 (±0.0000) | 1.0000 (±0.0000) | 1.0000 (±0.0000) | 0.0000 (±0.0000) | 0.0000 (±0.0000) | 0.0000 (±0.0000) | 0.5556 |
| Qwen2.5 | 1 | 1.0000 (±0.0000) | 1.0000 (±0.0000) | 1.0000 (±0.0000) | 0.0909 (±0.0909) | 0.6000 (±0.1633) | 0.4286 (±0.2020) | 0.0000 (±0.0000) | 0.0000 (±0.0000) | 0.0000 (±0.0000) | 0.4577 |
| Qwen2.5 | 5 | 1.0000 (±0.0000) | 1.0000 (±0.0000) | 1.0000 (±0.0000) | 0.0000 (±0.0000) | 0.6000 (±0.1633) | 0.8571 (±0.1429) | 0.0000 (±0.0000) | 0.0000 (±0.0000) | 0.0000 (±0.0000) | 0.4952 |
| DeepSeek-R1 | 1 | 0.0000 (±0.0000) | 0.0000 (±0.0000) | 0.0000 (±0.0000) | 0.0000 (±0.0000) | 0.0000 (±0.0000) | 0.0000 (±0.0000) | 0.0000 (±0.0000) | 0.0000 (±0.0000) | 0.0000 (±0.0000) | 0.0000 |
| DeepSeek-R1 | 5 | 0.7000 (±0.1527) | 0.9091 (±0.0909) | 1.0000 (±0.0000) | 0.0909 (±0.0909) | 0.7000 (±0.1528) | 0.7143 (±0.1844) | 0.0000 (±0.0000) | 0.0000 (±0.0000) | 0.0000 (±0.0000) | 0.4571 |
| Human | - | 1.0000 | 1.0000 | 1.0000 | 0.0000 | 1.0000 | 1.0000 | 1.0000 | 1.0000 | 0.0000 | 0.7778 |

Table 10: Semantic score for benchmarking oracle construction and random circuit synthesis.

| Model | Shot | Bernstein Vazirani | Deutsch Jozsa | Diffusion Operator | Grover | Simon | Generalized Simon (multi-str) | Generalized Simon (ternary) | Clifford | Universal | Avg |
|---|---|---|---|---|---|---|---|---|---|---|---|
| GPT-4o | 1 | 0.6250 (±0.0856) | 0.0000 (±0.0000) | 0.0000 (±0.0000) | 0.0000 (±0.0000) | 0.2688 (±0.1508) | 0.0167 (±0.0167) | 0.0000 (±0.0000) | 0.0000 (±0.0000) | 0.0000 (±0.0000) | 0.1012 |
| GPT-4o | 5 | 0.6958 (±0.0980) | 0.0000 (±0.0000) | 0.0000 (±0.0000) | 0.0000 (±0.0000) | 0.5521 (±0.1601) | 0.0000 (±0.0000) | 0.0000 (±0.0000) | 0.0000 (±0.0000) | 0.0000 (±0.0000) | 0.1387 |
| Llama3 | 1 | 0.2863 (±0.0815) | 0.0000 (±0.0000) | 0.6667 (±0.2108) | 0.0909 (±0.0909) | 0.0250 (±0.0167) | 0.0000 (±0.0000) | 0.0000 (±0.0000) | 0.0000 (±0.0000) | 0.0000 (±0.0000) | 0.1188 |
| Llama3 | 5 | 0.6000 (±0.0898) | 0.0000 (±0.0000) | 1.0000 (±0.0000) | 0.0000 (±0.0000) | 0.1000 (±0.1000) | 0.0000 (±0.0000) | 0.0000 (±0.0000) | 0.0000 (±0.0000) | 0.0000 (±0.0000) | 0.1890 |
| Qwen2.5 | 1 | 0.5125 (±0.0394) | 0.0000 (±0.0000) | 1.0000 (±0.0000) | 0.0909 (±0.0909) | 0.0313 (±0.0313) | 0.0000 (±0.0000) | 0.0000 (±0.0000) | 0.0000 (±0.0000) | 0.0000 (±0.0000) | 0.1816 |
| Qwen2.5 | 5 | 0.5862 (±0.0758) | 0.0000 (±0.0000) | 1.0000 (±0.0000) | 0.0000 (±0.0000) | 0.2375 (±0.1324) | 0.0357 (±0.0357) | 0.0000 (±0.0000) | 0.0000 (±0.0000) | 0.0000 (±0.0000) | 0.2066 |
| DeepSeek-R1 | 1 | 0.0000 (±0.0000) | 0.0000 (±0.0000) | 0.0000 (±0.0000) | 0.0000 (±0.0000) | 0.0000 (±0.0000) | 0.0000 (±0.0000) | 0.0000 (±0.0000) | 0.0000 (±0.0000) | 0.0000 (±0.0000) | 0.0000 |
| DeepSeek-R1 | 5 | 0.3638 (±0.0835) | 0.0000 (±0.0000) | 1.0000 (±0.0000) | 0.0909 (±0.0909) | 0.0313 (±0.0251) | 0.0000 (±0.0000) | 0.0000 (±0.0000) | 0.0000 (±0.0000) | 0.0000 (±0.0000) | 0.1651 |
| Human | - | 0.2500 | 1.0000 | 1.0000 | 0.0000 | 0.0000 | 0.0000 | 0.0000 | 1.0000 | 1.0000 | 0.5139 |

### C.1.3 Benchmarking Results for Cirq Version

Here we present the benchmarking results for the Cirq implementation of QCircuitBench. Interestingly, LLMs demonstrate strong performance across both syntax and semantic metrics on several oracles. Notably, while all models score zero on the Deutsch-Jozsa task in the QASM setting, several achieve semantic scores as high as 0.9000 in the Cirq version. This discrepancy likely stems from the modified oracle organization strategy necessitated by Cirq's distinct grammar compared to OpenQASM. Nevertheless, the overall performance trends remain consistent with the QASM results, reinforcing the relative difficulty of each task.

Table 11: Semantic score for oracle construction and random circuit synthesis (Cirq)

| Model | Shot | Bernstein Vazirani | Deutsch Jozsa | Diffusion Operator | Grover | Simon | Generalized Simon (multi-str) | Generalized Simon (ternary) | Clifford | Universal | Avg |
|---|---|---|---|---|---|---|---|---|---|---|---|
| GPT-4o | 1 | 0.9500 (±0.0500) | 0.7750 (±0.1205) | 0.0000 (±0.0000) | 1.0000 (±0.0000) | 0.6525 (±0.1289) | 0.1500 (±0.0982) | 0.3476 (±0.1742) | 0.0000 (±0.0000) | 0.0000 (±0.0000) | 0.4306 |
| GPT-4o | 5 | 0.9400 (±0.0600) | 0.8300 (±0.1041) | 0.0000 (±0.0000) | 1.0000 (±0.0000) | 0.5900 (±0.1464) | 0.4000 (±0.1633) | 0.5238 (±0.1854) | 0.0000 (±0.0000) | 0.0000 (±0.0000) | 0.4760 |
| Llama3 | 1 | 0.7750 (±0.1083) | 0.2625 (±0.1094) | 0.0000 (±0.0000) | 0.7000 (±0.1528) | 0.0000 (±0.0000) | 0.0250 (±0.0250) | 0.1429 (±0.1429) | 0.0000 (±0.0000) | 0.0000 (±0.0000) | 0.2117 |
| Llama3 | 5 | 0.7450 (±0.1055) | 0.7100 (±0.1320) | 0.0000 (±0.0000) | 0.9000 (±0.1000) | 0.0875 (±0.0591) | 0.0000 (±0.0000) | 0.3206 (±0.1581) | 0.0000 (±0.0000) | 0.0000 (±0.0000) | 0.3070 |
| Qwen2.5 | 1 | 0.5700 (±0.1578) | 0.4725 (±0.1272) | 0.0000 (±0.0000) | 0.6000 (±0.1633) | 0.0250 (±0.0250) | 0.0500 (±0.0500) | 0.0000 (±0.0000) | 0.0000 (±0.0000) | 0.0000 (±0.0000) | 0.1908 |
| Qwen2.5 | 5 | 0.5900 (±0.1464) | 0.7950 (±0.1122) | 0.6786 (±0.1786) | 1.0000 (±0.0000) | 0.0625 (±0.0502) | 0.2500 (±0.1344) | 0.6238 (±0.1833) | 0.0000 (±0.0000) | 0.0000 (±0.0000) | 0.4444 |
| DeepSeek-R1 | 1 | 0.6550 (±0.1495) | 0.4000 (±0.1633) | 0.0000 (±0.0000) | 0.0000 (±0.0000) | 0.0063 (±0.0063) | 0.0000 (±0.0000) | 0.0000 (±0.0000) | 0.0000 (±0.0000) | 0.0000 (±0.0000) | 0.1181 |
| DeepSeek-R1 | 5 | 0.8450 (±0.1086) | 0.8000 (±0.1106) | 0.4286 (±0.2020) | 1.0000 (±0.0000) | 0.0125 (±0.0125) | 0.0000 (±0.0000) | 0.2429 (±0.1445) | 0.0000 (±0.0000) | 0.0000 (±0.0000) | 0.3699 |

Table 12: Circuit syntax score for oracle construction and random circuit synthesis (Cirq)

| Model | Shot | Bernstein Vazirani | Deutsch Jozsa | Diffusion Operator | Grover | Simon | Generalized Simon (multi-str) | Generalized Simon (ternary) | Clifford | Universal | Avg |
|---|---|---|---|---|---|---|---|---|---|---|---|
| GPT-4o | 1 | 1.0000 (±0.0000) | 0.9000 (±0.1000) | 0.0000 (±0.0000) | 1.0000 (±0.0000) | 1.0000 (±0.0000) | 1.0000 (±0.0000) | 0.8571 (±0.1429) | 0.0000 (±0.0000) | 0.0000 (±0.0000) | 0.6397 |
| GPT-4o | 5 | 1.0000 (±0.0000) | 0.9000 (±0.1000) | 0.0000 (±0.0000) | 1.0000 (±0.0000) | 0.9000 (±0.1000) | 1.0000 (±0.0000) | 1.0000 (±0.0000) | 0.0000 (±0.0000) | 0.0000 (±0.0000) | 0.6444 |
| Llama3 | 1 | 0.9000 (±0.1000) | 0.6000 (±0.1633) | 0.0000 (±0.0000) | 0.7000 (±0.1528) | 0.7000 (±0.1528) | 0.4000 (±0.1633) | 0.5714 (±0.2020) | 0.8462 (±0.1042) | 0.9231 (±0.0769) | 0.6267 |
| Llama3 | 5 | 1.0000 (±0.0000) | 0.9000 (±0.1000) | 0.0000 (±0.0000) | 0.9000 (±0.1000) | 0.9000 (±0.1000) | 0.8000 (±0.1333) | 0.8571 (±0.1429) | 1.0000 (±0.0000) | 1.0000 (±0.0000) | 0.8179 |
| Qwen2.5 | 1 | 0.9000 (±0.1000) | 0.8000 (±0.1333) | 0.0000 (±0.0000) | 0.7000 (±0.1528) | 0.6000 (±0.1633) | 0.2000 (±0.1333) | 0.0000 (±0.0000) | 1.0000 (±0.0000) | 1.0000 (±0.0000) | 0.5789 |
| Qwen2.5 | 5 | 1.0000 (±0.0000) | 0.9000 (±0.1000) | 0.7143 (±0.1844) | 1.0000 (±0.0000) | 1.0000 (±0.0000) | 1.0000 (±0.0000) | 0.8571 (±0.1429) | 1.0000 (±0.0000) | 1.0000 (±0.0000) | 0.9417 |
| DeepSeek-R1 | 1 | 0.7000 (±0.1528) | 0.4000 (±0.1633) | 0.0000 (±0.0000) | 0.6000 (±0.1633) | 0.2000 (±0.1333) | 0.0000 (±0.0000) | 0.0000 (±0.0000) | 1.0000 (±0.0000) | 0.9231 (±0.0769) | 0.4248 |
| DeepSeek-R1 | 5 | 1.0000 (±0.0000) | 0.9000 (±0.1000) | 0.8571 (±0.1429) | 1.0000 (±0.0000) | 0.3000 (±0.1528) | 0.0000 (±0.0000) | 0.8571 (±0.1429) | 1.0000 (±0.0000) | 1.0000 (±0.0000) | 0.7682 |

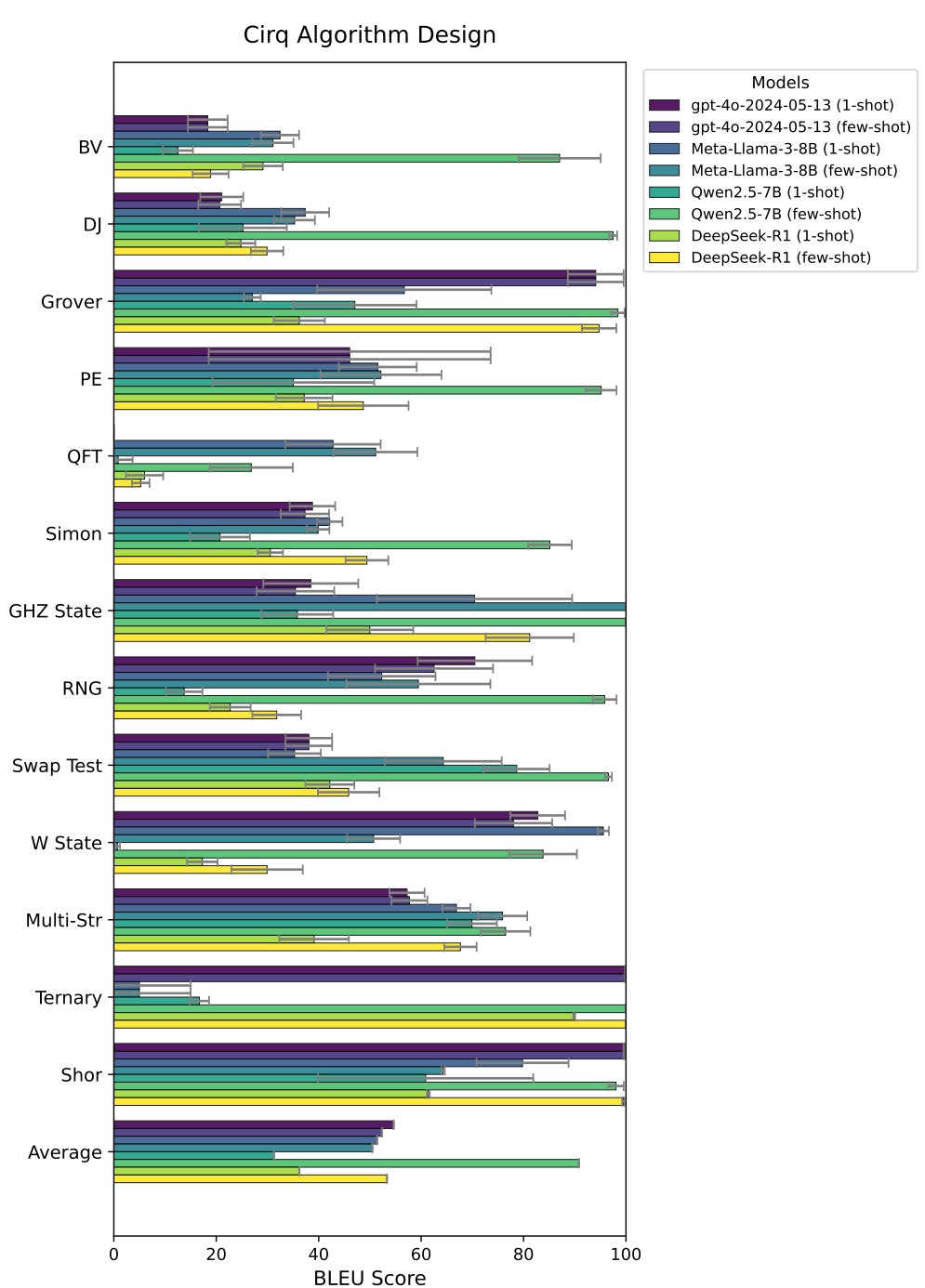

Figure 17: BLEU results for quantum algorithm design (Cirq).

## C.2 Fine-Tuning Results

### C.2.1 Analyzing Fine-Tuning Results for Random Circuit Synthesis

**Observations.** Regarding the counter-intuitive phenomenon where the performance on Clifford and universal random circuits decreases after fine-tuning, we conducted additional experiments and fine-tuned the model on 4,800 samples specifically for the Clifford task. Upon closer inspection, we observed that the model more frequently generated outputs with infinite loops and increased monotony, often producing repetitive gate patterns and repeatedly cycling over the same qubit after fine-tuning.

**Temperature.** We conducted experiments with different "temperature" parameters, which control the randomness of predictions. Formally, let $T > 0$ be the temperature, $z_i$ be the raw score for token $i$, the probability for token $i$ is computed as $p_i = \frac{e^{z_i/T}}{\sum_j e^{z_j/T}}$. Typically, lower temperatures make the model more conservative, while higher temperatures flatten the distribution, increasing the likelihood of generating originally less probable sequences. The results are shown in Table 13:

Table 13: Clifford Model Fine-Tuning Results Across Different Temperature Settings.

| Model | Setting | Temperature | BLEU | Aggr. Verification |
|---|---|---|---|---|
| Llama3 | 1-shots (5) | 0 | 13.3796 (±0.9508) | -0.6582 (±0.0360) |
| | | 0.2 | 12.5688 (±0.8276) | -0.6526 (±0.0372) |
| | | 1 | 53.0431 (±3.8422) | -0.1914 (±0.0361) |
| Llama3 | finetune | 0 | 7.6261 (±0.3433) | -0.8895 (±0.0247) |
| | | 0.2 | 13.8714 (±0.6536) | -0.7873 (±0.0306) |
| | | 1 | 32.5241 (±2.0548) | -0.2072 (±0.0358) |

**Entropy.** We also conducted an entropy analysis before and after fine-tuning. The output entropy decreased noticeably, suggesting the model became more deterministic and less diverse. This likely led to overfitting to high-probability surface patterns, sacrificing semantically correct but less frequent solutions. The issue is particularly detrimental for tasks requiring diverse outputs such as random circuit synthesis. This observation aligns with increases in BLEU scores and declines in verification scores, suggesting a mismatch between surface-level fluency and semantic correctness.

Table 14: Entropy comparison on random circuit synthesis before and after fine-tuning.

| Task | Entropy(Before) | Entropy(After) |
|---|---|---|
| Clifford | 0.7783 | 0.1429 |
| Universal | 0.8994 | 0.2115 |

**Explanations.** One possible explanation for this counter-intuitive result lies in the challenge of encoding quantum state vectors within a language model. In the problem description, the target quantum state is represented by a complex vector with four decimal places of precision, where the dimension scales as with the number of qubits. It is a well-known fact that LLMs generally struggle with very long floating-point numbers, which might contribute to the observed performance decline.

Another potential reason could be overfitting during fine-tuning, particularly for tasks that require high output diversity. The varying degrees of intrinsic difficulty and the amount of relevant pre-training knowledge across different tasks likely played a role. Oracle constructions are relatively simple for the model to learn. For example, in the Bernstein-Vazirani algorithm, the model only needs to apply a CNOT gate at positions corresponding to '1' bits. In contrast, the random circuits in the Clifford and Universal tasks involve more general and complex quantum state transformations, making them significantly more challenging. These tasks are also less common during pre-training, which could have hindered the model's ability to generalize without overfitting. This challenge is one of the reasons we initially considered a few-shot learning approach to be suitable.

While these are plausible hypotheses, we acknowledge that further investigation is required to draw definitive conclusions. We consider this an intriguing topic that warrants additional research. These

findings highlight the need for future work on diversity-aware objectives or regularization strategies during fine-tuning.

## C.2.2 Fine-Tuning Results for Quantum Algorithm Design

We also performed fine-tuning experiments on the quantum algorithm design task. To facilitate a more granular analysis, we present the unaggregated verification scores and efficiency metrics for this experiment. Fine-tuning introduces a notable shift in model behavior: while the QASM syntax score slightly declines, both the Python syntax score and the semantic accuracy improve substantially. This suggests that the model becomes more adept at capturing algorithmic intent and generating semantically correct circuits, even if they occasionally deviate from the rigid syntactic constraints of QASM. We hypothesize that this trade-off stems from quantum-specific bias induced by fine-tuning. These findings underscore the need for incorporating syntax-aware objectives or constrained decoding mechanisms to preserve formal correctness without compromising semantic fidelity after fine-tuning.

Table 15: Fine-Tuning Results for Quantum Algorithm Design

| Metric | Bernstein Vazirani | Deutsch Jozsa | Grover | Phase Estimation | Quantum Fourier Transform | Simon | GHZ | Random Number Generator | Swap Test | W State | Generalized Simon (multi-str) | Generalized Simon (ternary) | Shor |
|---|---|---|---|---|---|---|---|---|---|---|---|---|---|
| Semantic | 0.1296 (±0.0287) | 0.0366 (±0.0148) | 0.1174 (±0.0277) | 0.1012 (±0.0259) | 0.1255 (±0.0280) | 0.1284 (±0.0285) | 0.1203 (±0.0269) | 0.1276 (±0.0283) | 0.1283 (±0.0285) | 0.1245 (±0.0276) | 0.1296 (±0.0286) | 0.1233 (±0.0274) | 0.1201 (±0.0267) |
| QASM | 0.1504 (±0.0311) | 0.0602 (±0.0207) | 0.1278 (±0.0291) | 0.1128 (±0.0275) | 0.1504 (±0.0311) | 0.1504 (±0.0311) | 0.1439 (±0.0299) | 0.1504 (±0.0311) | 0.1515 (±0.0313) | 0.1460 (±0.0303) | 0.1504 (±0.0311) | 0.1460 (±0.0303) | 0.1399 (±0.0291) |
| Code | 0.7970 (±0.0350) | 0.7970 (±0.0350) | 0.8045 (±0.0345) | 0.8271 (±0.0329) | 0.8120 (±0.0340) | 0.8421 (±0.0317) | 0.8489 (±0.0305) | 0.7970 (±0.0350) | 0.7955 (±0.0352) | 0.8029 (±0.0341) | 0.8045 (±0.0345) | 0.8394 (±0.0315) | 0.8252 (±0.0319) |
| Gate | 1.0094 (±0.0197) | 1.0256 (±0.0557) | 1.0000 (±0.0000) | 1.0120 (±0.0251) | 1.0094 (±0.0197) | 1.0094 (±0.0197) | 1.0094 (±0.0197) | 1.0094 (±0.0197) | 1.0094 (±0.0197) | 1.0094 (±0.0197) | 1.0094 (±0.0197) | 1.0094 (±0.0197) | 1.0094 (±0.0197) |
| Shot | 1.1000 (±0.1000) | 1.7000 (±0.7000) | 1.0000 (±0.0000) | 1.1000 (±0.1000) | 1.1000 (±0.1000) | 1.1000 (±0.1000) | 1.1000 (±0.1000) | 1.1000 (±0.1000) | 1.1000 (±0.1000) | 1.1000 (±0.1000) | 1.1000 (±0.1000) | 1.1000 (±0.1000) | 1.1000 (±0.1000) |
| Time | 0.5099 (±0.0068) | 0.6213 (±0.1123) | 0.5124 (±0.0117) | 0.5067 (±0.0102) | 0.5203 (±0.0130) | 0.5045 (±0.0077) | 0.5050 (±0.0095) | 0.5472 (±0.0420) | 0.5070 (±0.0097) | 0.5141 (±0.0163) | 0.4961 (±0.0096) | 0.4918 (±0.0073) | 0.5589 (±0.0743) |

## C.3 Case Studies

After carefully examining the model's output, we observed several interesting patterns. We present a series of case studies to illustrate these observations and provide possible explanations.

### C.3.1 Improvisation Error

**Low Score for GPT-4o in One-Shot Setting.** At first glance, it is surprising that GPT-4o performs poorly on many quantum algorithms in the algorithm design task in the one-shot setting compared to Llama3-8B. Given that Llama3-8B has a relatively smaller parameter scale, the results should have been the other way around. A closer examination of the model's output reveals the potential reason: while Llama3-8B closely mimics the input examples, GPT-4o tends to improvise, resulting in outputs that are not well captured by the current syntax support. Here are several concrete examples.

This is the OpenQASM 3.0 code output for the W state with $n = 7$. In this code, GPT-4o uses the advanced "for" loop syntax newly introduced in OpenQASM 3.0 to create the circuit. Although the code fails to produce the W state, it is syntactically correct. However, the Qiskit.qasm3 import module, which converts OpenQASM 3.0 files to QuantumCircuit objects and is used in our verification function to check the correctness of the syntax of output OpenQASM codes, is still in the experimental stage and does not support many of OpenQASM 3.0's advanced features, including for loops. As a result, GPT-4o's use of these features causes the code to fail syntax validation, getting a score of -1.

```
OPENQASM 3.0;
include "stdgates.inc";
qubit[7] q;
h q[0];
for i in[1:6] {
    cx q[i-1], q[i];
}
```

Listing 8: OpenQASM 3.0 Code output by GPT-4o for W state with $n = 7$.

Here is another example where GPT-4o decides to assign novel names to its qubit registers, leading to a conflict in the symbol table in Scope.GLOBAL. If we substitute all the registers $x$, $y$, and $s$ with new names, the code can pass syntax validation successfully and is close to the correct solution.

```
OPENQASM 3.0;
include "stdgates.inc";
include "oracle.inc";
bit[9] s;
qubit[10] x;
qubit[11] y;
h x[0];
h x[1];
h x[2];
h x[3];
h x[4];
h x[5];
h x[6];
h x[7];
h x[8];
Oracle x[0], x[1], x[2], x[3], x[4], x[5], x[6], x[7], x[8], y;
h x[0];
h x[1];
h x[2];
h x[3];
h x[4];
h x[5];
h x[6];
h x[7];
h x[8];
measure x[0] -> s[0];
measure x[1] -> s[1];
measure x[2] -> s[2];
measure x[3] -> s[3];
measure x[4] -> s[4];
measure x[5] -> s[5];
measure x[6] -> s[6];
measure x[7] -> s[7];
measure x[8] -> s[8];
```

Listing 9: OpenQASM 3.0 Code output by GPT-4o for Bernstein-Vazirani Problem with $n = 9$.

Since Llama3-8B tends to follow the provided example more closely, it avoids such kind of errors. These factors may help explain why GPT-4o performs poorly in the one-shot setting compared to Llama3-8B.

**High Score for GPT-4o in Five-Shot Setting.** In contrast to its relatively low score in the one-shot setting, GPT-4o achieves the best overall performance in the five-shot setting for both oracle construction and algorithm design tasks.

Unlike in the one-shot setting, where GPT-4o frequently uses advanced OpenQASM 3.0 features, it learns from the examples in context and uses simpler syntax in the five-shot setting. For the W state task, here is an output in the five-shot setting:

```
OPENQASM 3.0;
include "stdgates.inc";
qubit[3] q;
h q[0];
cx q[0], q[1];
h q[1];
cx q[1], q[2];
h q[2];
```

Listing 10: OpenQASM 3.0 Code output by GPT-4o for W state with $n = 3$.

This output avoids the use of "for" loops and successfully passes the syntax validation test, although it still cannot generate the W state correctly.

In addition to adapting to plain syntax through in-context learning, GPT-4o achieves outstanding performance on more complicated tasks such as phase estimation. Here is the model output of GPT-4o on the phase estimation task with qubit number $n = 2$.

```
OPENQASM 3.0;
include "stdgates.inc";
include "oracle.inc";
bit[2] c;
qubit[2] q;
Psi q[0];
h q[1];
CU_0 q[0], q[1];
h q[1];
c[0] = measure q[1];
c[1] = measure q[0];
```

Listing 11: OpenQASM 3.0 Code output by GPT-4o for Phase Estimation with $n = 2$.

```
from qiskit import transpile

def run_and_analyze(circuit, aer_sim):
    """Run the circuit and analyze the result."""
    circ = transpile(circuit, aer_sim)
    result = aer_sim.run(circ, shots=1).result()
    counts = result.get_counts()
    phase_str = list(counts.keys())[0]
    phase = int(phase_str, 2) / 2**2
    return phase
```

Listing 12: Post-processing code output by GPT-4o for Phase Estimation with $n = 2$.

This suite of OpenQASM 3.0 circuits and post-processing functions successfully outputs the phase within the required precision for the test case, resulting in an impressive semantic score of 1.0. Despite the small number of qubits and differences from the reference implementation, the accuracy achieved is noteworthy.

These phenomena reflect that GPT-4o has impressive in-context learning abilities and overall better capabilities in designing and implementing quantum algorithms.

### C.3.2    Counting Error

If we ask the GPT-4o model to implement the code of Bernstein-Vazirani Problem for a given secret string, for example $s = 000101$, the model fails to apply the CX gate on the correct qubits.

```
OPENQASM 3.0;
include "stdgates.inc";

gate Oracle _gate_q_0, _gate_q_1, _gate_q_2, _gate_q_3, _gate_q_4,
    ↪ _gate_q_5, _gate_q_6 {
  // Apply CX gates according to the secret string "000101"
  cx _gate_q_2, _gate_q_6;
  cx _gate_q_5, _gate_q_6;
}

bit[6] c;
qubit[7] q;

// Apply Hadamard gates to the first 6 qubits
h q[0];
h q[1];
h q[2];
h q[3];
h q[4];
```

```
h q[5];

// Prepare the ancilla qubit in the |1> state and apply a Hadamard
    ↪ gate
x q[6];
h q[6];

// Call the oracle
Oracle q[0], q[1], q[2], q[3], q[4], q[5], q[6];

// Apply Hadamard gates to the first 6 qubits again
h q[0];
h q[1];
h q[2];
h q[3];
h q[4];
h q[5];

// Measure the first 6 qubits
c[0] = measure q[0];
c[1] = measure q[1];
c[2] = measure q[2];
c[3] = measure q[3];
c[4] = measure q[4];
c[5] = measure q[5];
```

Listing 13: OpenQASM code for Bernstein-Vazirani Problem with s = 000101 in few-shot setting.

We reminded the model of the mechanism with the following prompt:

> **Reminder Prompt**
>
> Remember that in the construction of oracles for Bernstein-Vazirani Problem, you should apply cx gates where the control qubits are where the string bit equals 1, and the target qubit is the ancillary qubit.

There was no improvement. We further asked the model about the indices of the bits in the string which equal 1. It provided the following answer:

> **GPT-4o's answer for counting indices**
>
> In the secret string s="000101", the bits that equal 1 are found at the following indices (0-based):
> Index 2
> Index 5
> These indices correspond to the positions where the string has a '1'.

This reveals the incapability for GPT models to count the bit of 1 and apply corresponding gates to construct the oracle.

### C.3.3  Data Contamination

In our experiments, we observed an interesting phenomenon of performance discrepancy between implementing quantum algorithms with Qiskit codes for the general setting and with OpenQASM codes for a concrete case.

We provided the following prompt:

**Prompt for Bernstein-Vazirani Problem with Qiskit**

Please write the code with Qiskit implementing Bernstein-Vazirani Algorithm with qubit
number n = 5.

The model answered with the following text and codes:

**Text Answer by GPT-4o**

Here's a Python code using Qiskit to implement the Bernstein-Vazirani algorithm with $n = 5$
qubits. The secret string used in this example is $s = 10101$ (you can change this to any string
of your choice).

```python
from qiskit import QuantumCircuit, Aer, transpile, assemble
from qiskit.visualization import plot_histogram

# Secret string s for the Bernstein-Vazirani problem
s = '10101'

# Number of qubits
n = len(s)

# Create a Quantum Circuit with n+1 qubits (one ancillary qubit)
bv_circuit = QuantumCircuit(n+1, n)

# Apply Hadamard gates to all qubits except the ancillary qubit
bv_circuit.h(range(n))

# Apply X and Hadamard gate to the ancillary qubit
bv_circuit.x(n)
bv_circuit.h(n)

# Construct the Oracle
for i, bit in enumerate(s):
    if bit == '1':
        bv_circuit.cx(i, n)

# Apply Hadamard gates to the first n qubits
bv_circuit.h(range(n))
# Measure the first n qubits
bv_circuit.measure(range(n), range(n))
# Visualize the circuit
bv_circuit.draw('mpl')

# Run the circuit using Qiskit Aer simulator
simulator = Aer.get_backend('qasm_simulator')
t_qc = transpile(bv_circuit, simulator)
qobj = assemble(t_qc)
result = simulator.run(qobj).result()

# Get the result counts and plot the histogram
counts = result.get_counts()
plot_histogram(counts)
```

Listing 14: Qiskit code for Bernstein-Vazirani Problem generated by GPT-4o in the zero-shot setting.

This code, although adopts an older version of Qiskit syntax, can be compiled and run successfully
with the correct answer. Similar performances can be observed with OpenQASM codes:

```
// OpenQASM 2.0
OPENQASM 2.0;
include "qelib1.inc";
```

```
// Declare 5 qubits for input and 1 ancilla qubit
qreg q[6];
creg c[5]; // classical register for measuring the result

// Step 1: Apply Hadamard gates to the input qubits
h q[0];
h q[1];
h q[2];
h q[3];
h q[4];

// Apply Hadamard to the ancillary qubit and set it to |1>
x q[5];
h q[5];

// Step 2: Oracle implementation (CNOT gates based on secret string s
    ↪ = 10110)
cx q[0], q[5]; // Secret string bit 1
// No cx for q[1] because the secret string bit is 0
cx q[2], q[5]; // Secret string bit 1
cx q[3], q[5]; // Secret string bit 1
// No cx for q[4] because the secret string bit is 0

// Step 3: Apply Hadamard gates to the input qubits again
h q[0];
h q[1];
h q[2];
h q[3];
h q[4];

// Step 4: Measure the qubits to get the result
measure q[0] -> c[0];
measure q[1] -> c[1];
measure q[2] -> c[2];
measure q[3] -> c[3];
measure q[4] -> c[4];
```

Listing 15: OpenQASM code for Bernstein-Vazirani Problem generated by GPT-4o.

This phenomenon contrasts with the results in Appendix C.3.2. Considering that Qiskit has implemented tutorial codes for several representative quantum algorithms, this seemingly outstanding performance might be a deceptive result caused by data contamination. Therefore, only a carefully designed quantum algorithm dataset can avoid the effects of data contamination, allowing for an effective evaluation of the model's genuine capability in quantum algorithm design and implementation. This dataset is also meaningful for testing general AI code generation and syntax learning, where no existing AI dataset could substitute us.

