# QCircuitBench: A Large-Scale Dataset for Benchmarking Quantum Algorithm Design

## Abstract

Quantum computing is an emerging field recognized for the significant speedup it offers over classical computing through quantum algorithms. However, designing and implementing quantum algorithms pose challenges due to the complex nature of quantum mechanics and the necessity for precise control over quantum states. Despite the significant advancements in AI, there has been a lack of datasets specifically tailored for this purpose. In this work, we introduce QCircuitBench, the first benchmark dataset designed to evaluate AI's capability in designing and implementing quantum algorithms in the form of quantum circuit codes. Unlike using AI for writing traditional codes, this task is fundamentally more complicated due to highly flexible design space. Our key contributions include:

1. A general framework which formulates the key features of quantum algorithm design task for Large Language Models.

2. Implementation for quantum algorithms from basic primitives to advanced applications, spanning 3 task suites, 23 algorithms, and 128,573 data points.

3. Automatic validation and verification functions, allowing for iterative and interactive evaluation without human inspection.

4. Promising potential as a training dataset through primitive fine-tuning results.

We observed several interesting experimental phenomena: fine-tuning does not always outperform few-shot learning, and LLMs tend to exhibit consistent error patterns. In all, QCircuitBench is a comprehensive benchmark for AI-driven quantum algorithm design, while it also reveals limitations of LLMs in this domain.

## 1 Introduction

Quantum computing is an emerging field in recent decades because algorithms on quantum computers may solve problems significantly faster than their classical counterparts. From the perspective of theoretical computer science, the design of quantum algorithms have been investigated in various research directions - see the survey [Dalzell et al., 2023] and the quantum algorithm zoo [Jordan, 2025]. However, the design of quantum algorithms on quantum computers has been completed manually by researchers. This process is notably challenging due to highly flexible design space and extreme demands for a comprehensive understanding of mathematical tools and quantum properties.

For these reasons, quantum computing is often considered to have high professional barriers. As the discipline evolves, we aim to explore more possibilities for algorithm design and implementation in the quantum setting. This is aligned with recent advances among AI for Science, including AlphaFold [Jumper et al., 2021], AlphaGeometry [Trinh et al., 2024], etc. Recently, large language models (LLMs) have also become widely applicable among AI for science approaches [Yang et al., 2024b, Zhang et al., 2024, Yu et al., 2024]. LLMs represent the best practice of sequential modeling

methods at current stage. They have an edge over other models in possessing abundant pre-training knowledge and providing human-friendly interfaces which support human-machine collaboration. Therefore, we gear LLMs for quantum algorithm design.

As far as we know, there has not been any dataset for AI in quantum algorithm design. Existing work combining quantum computing and AI mostly targets at exploiting quantum computing for AI; there are some papers applying AI for quantum computing, but they either consider niche problems [Nakayama et al., 2023, Schatzki et al., 2021] or limited functions [Tang et al., 2023, Fürrutter et al., 2024], not quantum algorithm datasets of general interest (see Section 2). However, unlike classical code generation where abundant data exist, the most challenging aspect for quantum algorithm design is the lack of sufficient data, and hence the difficulty of generalization in training AI models. Therefore, datasets for quantum algorithm design are solicited.

Descriptions of quantum algorithms in natural language could be verbose and vague. Mathematical formulas, while precise and succinct, are difficult to

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

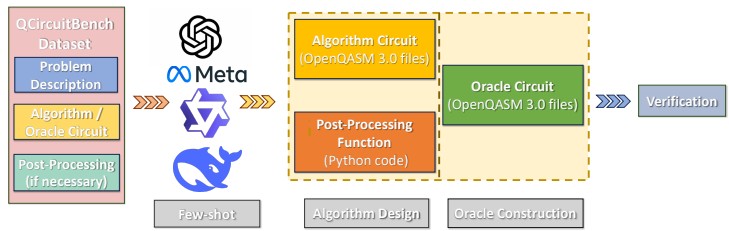

Figure 2: Flowchart of benchmarking QCircuitBench.

**Models.** Recently, the GPT series models have become the benchmark for generative models due to their exceptional performance. Specifically, we include two models from OpenAI, GPT-3.5-turbo [Brown et al., 2020] and GPT-4 [OpenAI et al., 2024], in our benchmark. Additionally, the LLAMA series models [Touvron et al., 2023a,b] are widely recognized as leading open-source models, and we have selected LLAMA-3-8B for our study. For a comprehensive evaluation, we also benchmark Qwen 2.5 [Yang et al., 2024a] and DeepSeek-R1 [Guo et al., 2025].

On these models, we employ a few-shot learning framework, a prompting technique that has shown considerable success in generative AI [Xie et al., 2021]. In this approach, we utilize either 1, 3, or 5 examples, followed by a problem description. To ensure we do not train and test on the same quantum algorithm, we implement $k$-fold validation among all algorithms.

**Evaluation Metrics.** We use three evaluation metrics (see Appendix C.1 for more details):

1. BLEU Score: This metric measures how closely the generated code matches the reference code, with a higher BLEU score indicating more similarity. Formally, the BLEU score is defined as:

$$\text{BLEU} = \text{BP} \cdot \exp\left(\sum_{n=1}^{N} w_n \log p_n\right),$$

where BP is the acronym for brevity penalty, $w_n$ is the weight for the n-gram precision (typically $\frac{1}{N}$ for uniform weights), $p_n$ is the precision for n-grams. BP is calculated as:

$$\text{BP} = \begin{cases} 1 & \text{if } c > r \\ e^{1-\frac{r}{c}} & \text{if } c \leq r \end{cases},$$

---

[2]The verification function explicitly integrates the oracle / gate definition library with output algorithm circuit since Qiskit importer for OpenQASM 3.0 does not support non-standard gate libraries currently.

Table 1: Benchmarking algorithm design in verification function scores.

| Model | Shot | Bernstein Vazirani | Deutsch Jozsa | Grover | Phase Estimation | QFT | Simon | GHZ | Random Number Generator | Swap Test | W State | Generalized Simon (multi-str) | Generalized Simon (ternary) | VQE | QAOA | Shor | Avg |
|---|---|---|---|---|---|---|---|---|---|---|---|---|---|---|---|---|---|
| gpt4o | 1 | 0.8461 (±0.1538) | 0.8307 (±0.1533) | 0.7644 (±0.1618) | 0.6638 (±0.1141) | -1.0000 (±0.0000) | -0.2015 (±0.1268) | -1.0000 (±0.0000) | -1.0000 (±0.0000) | 0.5292 (±0.1745) | -0.0900 (±0.2279) | -1.0000 (±0.0000) | -0.4152 (±0.1857) | -1.0000 (±0.0000) | -1.0000 (±0.0000) | -1.0000 (±0.0000) | -0.7452 |
| gpt4o | few | 0.9592 (±0.0408) | 0.9692 (±0.0208) | 0.9165 (±0.0766) | 0.4400 (±0.1784) | -1.0000 (±0.0000) | 0.0600 (±0.0364) | -1.0000 (±0.0000) | -1.0000 (±0.0000) | 0.8090 (±0.0292) | 0.4022 (±0.0776) | -1.0000 (±0.0000) | -0.3222 (±0.1698) | -1.0000 (±0.0000) | -1.0000 (±0.0000) | -1.0000 (±0.0000) | -0.8188 |
| Llama3 | 1 | -0.7538 (±0.1727) | -0.1077 (±0.2214) | -0.6154 (±0.2130) | -0.9231 (±0.0769) | -1.0000 (±0.0000) | -0.9231 (±0.0769) | -1.0000 (±0.0000) | -1.0000 (±0.0000) | -1.0000 (±0.0000) | -0.6933 (±0.2031) | -1.0000 (±0.0000) | -1.0000 (±0.0000) | -1.0000 (±0.0000) | -1.0000 (±0.0000) | -1.0000 (±0.0000) | -0.8583 |
| Llama3 | few | -0.5962 (±0.2184) | 0.3962 (±0.2308) | -0.4605 (±0.2155) | -0.6615 (±0.1492) | -1.0000 (±0.0000) | -0.5846 (±0.1543) | -1.0000 (±0.0000) | -1.0000 (±0.0000) | -0.4572 (±0.2143) | -0.6122 (±0.2599) | -0.8889 (±0.1111) | -1.0000 (±0.0000) | -1.0000 (±0.0000) | -1.0000 (±0.0000) | -1.0000 (±0.0000) | -0.7893 |
| gpt3.5 | 1 | 0.0769 (±0.2392) | 0.2731 (±0.1794) | -0.0779 (±0.2099) | -0.1277 (±0.2722) | -1.0000 (±0.0000) | -0.2181 (±0.1238) | -1.0000 (±0.0000) | -1.0000 (±0.0000) | 0.1563 (±0.2399) | -0.6922 (±0.2111) | -1.0000 (±0.0000) | -0.2211 (±0.0000) | -1.0000 (±0.0000) | -1.0000 (±0.0000) | -1.0000 (±0.0000) | -0.6670 |
| gpt3.5 | few | 0.2292 (±0.1534) | 0.2038 (±0.1708) | 0.0010 (±0.2508) | -0.5577 (±0.2331) | -1.0000 (±0.0000) | -0.1296 (±0.1078) | -1.0000 (±0.0000) | -1.0000 (±0.0000) | -0.3742 (±0.2332) | -0.8778 (±0.1222) | -1.0000 (±0.0000) | -0.3267 (±0.1684) | -1.0000 (±0.0000) | -1.0000 (±0.0000) | -1.0000 (±0.0000) | -0.7514 |
| Qwen 2.5 | 1 | -0.8923 (±0.1077) | -0.6285 (±0.1967) | -0.9231 (±0.0769) | -0.8462 (±0.1042) | -1.0000 (±0.0000) | -1.0000 (±0.0000) | -1.0000 (±0.0000) | -1.0000 (±0.0000) | -0.6270 (±0.1982) | -0.8444 (±0.1556) | -1.0000 (±0.0000) | -1.0000 (±0.0000) | -1.0000 (±0.0000) | -1.0000 (±0.0000) | -1.0000 (±0.0000) | -0.9115 |
| Qwen 2.5 | few | -0.4123 (±0.2268) | -0.0746 (±0.2211) | -0.9230 (±0.0769) | -0.8307 (±0.1151) | -1.0000 (±0.0000) | -0.6838 (±0.1371) | -1.0000 (±0.0000) | -1.0000 (±0.0000) | -0.3552 (±0.2403) | -1.0000 (±0.0000) | -0.8888 (±0.1111) | -1.0000 (±0.0000) | -1.0000 (±0.0000) | -1.0000 (±0.0000) | -1.0000 (±0.0000) | -0.7978 |
| DeepSeek-R1 | 1 | -0.8462 (±0.1538) | -1.0000 (±0.0000) | -1.0000 (±0.0000) | -1.0000 (±0.0000) | -1.0000 (±0.0000) | -0.8392 (±0.1090) | -1.0000 (±0.0000) | -1.0000 (±0.0000) | -1.0000 (±0.0000) | -0.6000 (±0.2651) | -1.0000 (±0.0000) | -1.0000 (±0.0000) | -1.0000 (±0.0000) | -1.0000 (±0.0000) | -1.0000 (±0.0000) | -0.9490 |
| DeepSeek-R1 | few | -0.5385 (±0.2152) | -0.2892 (±0.2607) | -1.0000 (±0.0000) | -0.7712 (±0.1618) | -1.0000 (±0.0000) | -1.0000 (±0.0000) | -1.0000 (±0.0000) | -1.0000 (±0.0000) | -0.6182 (±0.0000) | -0.8833 (±0.2031) | -1.0000 (±0.1167) | -1.0000 (±0.0000) | -0.5000 (±0.4999) | -1.0000 (±0.0000) | -1.0000 (±0.0000) | -0.8286 |

where $c$ is the length of the generated text and $r$ is the length of the reference text. Furthermore, n-gram precision $p_n$ is calculated as:

$$p_n = \frac{\sum_{C \in \text{Candidates}} \sum_{n\text{gram} \in C} \min(\text{Count}(n\text{gram in candidate}), \text{Count}(n\text{gram in references}))}{\sum_{C \in \text{Candidates}} \sum_{n\text{gram} \in C} \text{Count}(n\text{gram in candidate})}.$$

2. Verification function: This function checks the syntax validation and the result correctness of the code produced by the language model. To be specific, we evaluate the result using three criteria:

   (a) **QASM Syntax Verification**: We first check the syntax of the QASM code provided by the model. The syntax verification function $V_{\text{QASM}}(q)$ is set to be 1 if the QASM syntax is correct, and 0 otherwise.

   (b) **Python Syntax Verification**: Similarly, the syntax of the post-processing Python code (which includes the run_and_analyze function), denoted $V_{\text{code}}(c)$, is set to be 1 if the Python syntax is correct, and 0 otherwise.

   (c) **Execution and Evaluation**: If at least one syntax check passes, we proceed to evaluating the functional correctness. For each test case $t$, we run the quantum circuit simulation for a number of shots $M$, and compare the result with the ground truth. The success rate is calculated as:

$$\text{acc} = \frac{\sum_{t=1}^{T} \sum_{m=1}^{M} \mathbb{I}[\text{result} = \text{ground-truth}]}{T \times M}.$$

The final verification score is a triplet $(V_{\text{QASM}}(q), V_{\text{code}}(c), \text{acc})$. In addition, all the verification functions were executed by classical simulations in our experiments, but the APIs we implemented are compatible with IBM hardware and can be easily adapted to quantum computers.

3. Byte Perplexity: This metric evaluates the model's ability to predict the next byte in a sequence. Formally, the Perplexity score is defined as:

$$\text{PPL}(x) = 2^{-\frac{1}{N} \sum_{i=1}^{N} \log_2 p(x_i|x_{<i})},$$

where $p(x_i|x_{<i})$ is the probability of the $i$-th byte $x_i$ given the preceding bytes $x_{<i}$ and $N$ is the length of the byte sequence. Lower byte perplexity indicates better performance by reflecting the model's predictive accuracy.

The results for BLEU scores are shown in Figure 3. The verification scores of algorithm design tasks are shown in Table 1. We include the results of Byte Perplexity and the verification scores of oracle construction tasks in Appendix C.2.

We observe the following phenomena from the results:

- Most models achieve better scores in the few-shot setting than the 1-shot setting. This indicates their capability to learn effectively from contextual examples. Specifically, the score of tasks such as Deutsch-Jozsa were notably increased by 0.7108 after few-shot learning in the DeepSeek-R1 model. However, all models struggle with more complicated algorithms such as Grover, phase estimation, and quantum Fourier transform, with a score increase of 0.2616 by the Llama3 model on phase estimation. This highlights the differences in task difficulty.

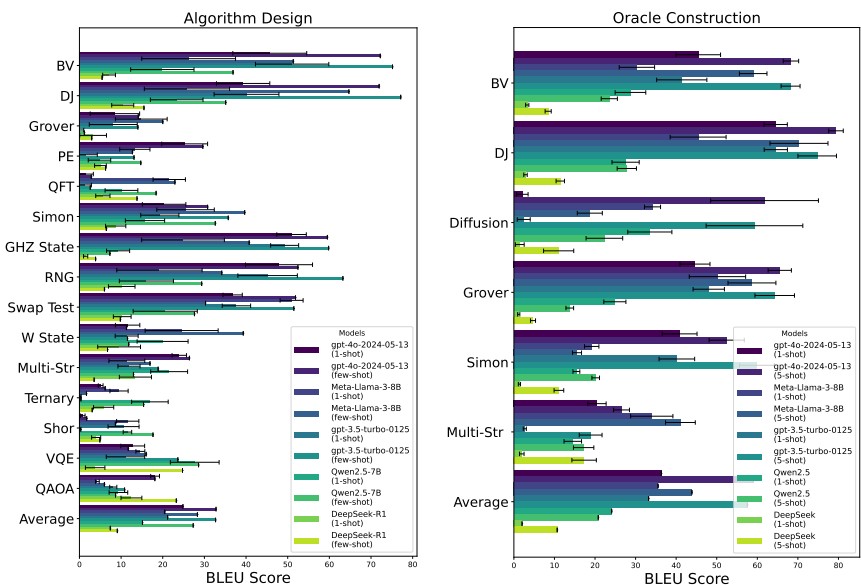

Figure 3: Benchmarking algorithm design and oracle construction tasks in BLEU scores.

- There exist challenges for near-term quantum algorithms. In particular, for the VQE and QAOA tasks, the models often fail to construct right parameterized circuits or apply optimization strategies correctly, with a score of at most -0.5000 by the DeepSeek-R1 model. This reflects the limitation of LLMs in handling hybrid quantum-classical workflows.

- GPT-4o and GPT-3.5 consistently excel in long-context comprehension, significantly outperforming other models across tasks, which highlights their superior in-context learning capabilities. In contrast, DeepSeek-R1 underperforms due to its long-chain reasoning style, which often exceeds the context length before producing a complete and verifiable solution.

- Although BLEU scores generally align with verification results, some discrepancies arise, such as the swap test showing relatively high BLEU scores but incorrect algorithm generation by most models. This observation emphasizes the need for complementary evaluation metrics such as our verification function.

**Types of Errors Made by LLMs.** In Appendix C.3, we include several case studies to illustrate and analyze various types of errors made by LLMs. In particular, they can be summarized as follows:

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

## 5 Conclusions and Future Work

In this paper, we propose QCircuitBench, the first comprehensive, structured universal quantum algorithm dataset and quantum circuit generation benchmark for AI models. This framework formulates quantum algorithm design from the programming language perspective and includes detailed descriptions and implementation of most established and important quantum algorithms / primitives, allowing for automatic verification methodologies. Benchmarking of QCircuitBench on up-to-date LLMs is systematically conducted. Fine-tuning results also showcase the potential of QCircuitBench as a training dataset, and implementation of the Generalized Simon's Problem mentioned in Section 3.1.2 showcases the compatibility of our framework with more complex algorithms. In addition, our framework is designed to scale with increasing qubit numbers and support complex quantum algorithms as long as they are efficiently implementable with polynomial gates.

Our work leaves several open questions for future investigation:

- QCircuitBench is a benchmarking dataset for LLMs. It is of general interest to extend benchmarking to training, which will help LLMs better maneuver quantum algorithm design. We have implemented advanced algorithms such as the Generalized Simon's Problem, but this in general needs implementations of more advanced algorithms to make it impactful.

- Since quantum algorithms have fundamental difference from classical algorithms, novel fine-tuning methods to attempt quantum algorithm design and quantum circuit implementation, or even developments of new quantum algorithms by LLMs are solicited.

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

## A  Details of QCircuitBench

The QCircuitBench Dataset, along with its Croissant metadata, is available on Harvard Dataverse at the following link: https://doi.org/10.7910/DVN/ZC4PNI

QCircuitBench has the following directory structure:

```
QCircuitBench

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

    t = 1
    with open(f"test_oracle/n{n}/trial{t}/oracle.inc", "r") as file
        ↪ :
        oracle_def = file.read()
    full_qasm = plug_in_oracle(qasm_string, oracle_def)
    circuit = verify_qasm_syntax(full_qasm)
    if circuit is None:
        return -1
    try:
        exec(code_string, globals())
        aer_sim = AerSimulator()
        total_success = 0
        total_fail = 0
        t_range = min(10, 4 ** (n - 2))
```

```
        shots = 10
        for t in range(1, 1 + t_range):
            print(f"    Running Test Case {t}")
            with open(f"test_oracle/n{n}/trial{t}/oracle.inc", "r")
                ↪ as file:
                oracle_def = file.read()
            full_qasm = plug_in_oracle(qasm_string, oracle_def)
            circuit = loads(full_qasm)
            with open(f"test_oracle/n{n}/trial{t}/oracle_info.txt",
                ↪ "r") as file:
                content = file.read()
            match = re.search(r"Secret string: ([01]+)", content)
            if match:
                secret_string = match.group(1)
            else:
                raise ValueError("Secret string not found in the
                    ↪ file.")

            cnt_success = 0
            cnt_fail = 0
            for shot in range(shots):
                prediction = run_and_analyze(circuit.copy(),
                    ↪ aer_sim)
                if not isinstance(prediction, str):
                    raise TypeError("Predicted secret string should
                        ↪ be a string.")
                if prediction == secret_string:
                    cnt_success += 1
                else:
                    cnt_fail += 1
            print(f"      Success: {cnt_success}/{shots}, Fail: {
                ↪ cnt_fail}/{shots}")
            total_success += cnt_success
            total_fail += cnt_fail
        print(f"Total Success: {total_success}; Total Fail: {
            ↪ total_fail}")
        return total_success / (total_fail + total_success)

    except Exception as e:
        print(f"Error: {e}")
        return -1
```

Listing 5: Verification function for Simon's algorithm.

This verification function is accompanied with an "{algorithm_name}_utils.py" file to provide
necessary utility functions.

```
from Qiskit.qasm3 import loads
from Qiskit_aer import AerSimulator
import re

def print_and_save(message, text):
    print(message)
    text.append(message)

def plug_in_oracle(qasm_code, oracle_def):
    """Plug-in the oracle definition into the QASM code."""
    oracle_pos = qasm_code.find('include "oracle.inc";')
    if oracle_pos == -1:
        raise ValueError("Oracle include statement not found in the
            ↪ file")
    full_qasm = (
        qasm_code[:oracle_pos]
```

```
                + oracle_def
                + qasm_code[oracle_pos + len('include "oracle.inc";') :]
        )
        return full_qasm

def verify_qasm_syntax(output):
    """Verify the syntax of the output and return the corresponding
        ↪  QuantumCircuit (if it is valid)."""
    assert isinstance(output, str)
    try:
        # Parse the OpenQASM 3.0 code
        circuit = loads(output)
        print(
            "    The OpenQASM 3.0 code is valid and has been
                ↪ successfully loaded as a QuantumCircuit."
        )
        return circuit
    except Exception as e:
        print(f"    Error: The OpenQASM 3.0 code is not valid.
            ↪ Details: {e}")
        return None
```

Listing 6: Utility functions for verification of Simon's algorithm.

7. **Dataset Creation Script:** this script involves all the code necessary to create the data points from scratch. The file is named as "{algorithm_name}_dataset.py". The main function looks like this:

```
def main():
    parser = argparse.ArgumentParser()
    parser.add_argument(
        "-f",
        "--func",
        choices=["qasm", "json", "gate", "check"],
        help="The function to call: generate qasm circuit, json
            ↪ dataset or extract gate definition.",
    )
    args = parser.parse_args()
    if args.func == "qasm":
        generate_circuit_qasm()
    elif args.func == "json":
        generate_dataset_json()
    elif args.func == "gate":
        extract_gate_definition()
    elif args.func == "check":
        check_dataset()
```

Listing 7: Main function of the dataset script for Simon's algorithm.

Here the "generate_circuit_qasm()" function generates the raw data of quantum circuits in Open-QASM 3.0 format where the algorithm circuit and the oracle definition are blended, then "extract_gate_definition()" function extracts the definition of oracles and formulates the algorithm circuits into the format suitable for model output. The "check_dataset()" function is used to check the correctness of the created data points and "generate_dataset_json()" function to combine the data into json format for easy integration with the benchmarking pipeline.

## A.2 Discussion of more tasks

**Problem Encoding.** In Section 3.1.1, we mentioned another category of oracle construction tasks referred to as "Problem Encoding", which involves applying quantum algorithms, such as Grover's algorithm, to solve practical problems such as SAT and triangle finding. The crux of this process is encoding the problem constraints into Grover's oracle, thereby making this a type of oracle construction task. Unlike quantum logic synthesis, which encodes an explicit function $f(x)$ as a unitary operator $U_f$, this task involves converting the constraints of a particular problem into the

required oracle form. We provide implementations of several concrete problems in this directory as demonstrations and will include more applications in future work.

**Quantum Information Protocols.**  In Section 3.1.2, we have also implemented three important quantum information protocols: Quantum Teleportation, Superdense Coding, and Quantum Key Distribution (BB84). A brief introduction to these protocols can be found in Appendix B. We did not include the experiments for these protocols as they involve communication between two parties, which is challenging to characterize with a single OpenQASM 3.0 file. We recommend revising the post-processing function as a general classical function to schedule the communication and processing between different parties specifically for these protocols. The fundamental quantum circuits and processing codes are provided in the repository.

## A.3   Datasheet

Here we present a datasheet for the documentation of QCircuitBench.

### Motivation

- *For what purpose was the dataset created?* It was created as a benchmark for the capability of designing and implementing quantum algorithms for LLMs.

- *Who created the dataset (e.g., which team, research group) and on behalf of which entity (e.g., company, institution, organization)?* The authors of this paper.

- *Who funded the creation of the dataset?* We will reveal the funding resources in the Acknowledgement section of the final version.

### Composition

- *What do the instances that comprise the dataset represent (e.g., documents, photos, people, countries)?* The dataset comprises problem description, generation code, algorithm circuit, post-processing function, oracle / gate definition, verification function, and dataset creation script for various quantum algorithms.

- *How many instances are there in total (of each type, if appropriate)?* The dataset has 3 task suites, 23 algorithms, and 128,573 data points. There are additional quantum information protocols and problem encoding tasks not included for experiments.

- *Does the dataset contain all possible instances or is it a sample (not necessarily random) of instances from a larger set?* The dataset contains instances with restricted qubit numbers due to the current scale of real quantum hardware.

- *What data does each instance consist of?* Qiskit codes, OpenQASM 3.0 codes, python scripts, and necessary text information.

- *Are relationships between individual instances made explicit?* Yes, the way to create different instances are clearly described in Appendix A.1.

- *Are there recommended data splits?* Yes, we recommend splitting the data according to different algorithms in algorithm design task.

- *Are there any errors, sources of noise, or redundancies in the dataset?* There might be some small issues due to the dumping process of Qiskit and programming mistakes (if any).

- *Is the dataset self-contained, or does it link to or otherwise rely on external resources (e.g., websites, tweets, other datasets)?* The dataset is self-contained.

- *Does the dataset contain data that might be considered confidential (e.g., data that is protected by legal privilege or by doctor-patient confidentiality, data that includes the content of individuals' non-public communications)?* No.

- *Does the dataset contain data that, if viewed directly, might be offensive, insulting, threatening, or might otherwise cause anxiety?* No.

**Collection Process**

- *How was the data associated with each instance acquired?* The data is created by first composing Qiskit codes for each algorithm and then converting to OpenQASM 3.0 files using Qiskit.qasm3.dump function, with additional processing procedure.

- *What mechanisms or procedures were used to collect the data (e.g., hardware apparatuses or sensors, manual human curation, software programs, software APIs)?* Manual human programming and Qiskit APIs.

- *Who was involved in the data collection process (e.g., students, crowd workers, contractors), and how were they compensated (e.g., how much were crowd workers paid)?* Nobody other than the authors of the paper.

- *Over what timeframe was the data collected?* The submitted version of the dataset was created in May 2025.

**Uses**

- *Has the dataset been used for any tasks already?* It has been used in this paper to benchmark LLM's ability for quantum algorithm design.

- *Is there a repository that links to any or all papers or systems that use the dataset?* The only paper which uses the dataset for now is this paper.

**Distribution**

- *Will the dataset be distributed to third parties outside of the entity (e.g., company, institution, organization) on behalf of which the dataset was created?* Yes, the dataset will be made publicly available on the Internet after the review process.

- *How will the dataset be distributed (e.g., tarball on website, API, GitHub)?* It will be distributed on the GitHub platform.

- *Will the dataset be distributed under a copyright or other intellectual property (IP) license, and/or under applicable terms of use (ToU)?* The dataset is distributed under CC BY 4.0.

- *Have any third parties imposed IP-based or other restrictions on the data associated with the instances?* No.

- *Do any export controls or other regulatory restrictions apply to the dataset or to individual instances?* No.

**Maintenance**

- *Who will be supporting/hosting/maintaining the dataset?* The authors of this paper.

- *How can the owner/curator/manager of the dataset be contacted (e.g., email address)?* The email for contact will be provided after the review process.

- *Is there an erratum?* Not at this time.

- *Will the dataset be updated (e.g., to correct labeling errors, add new instances, delete instances)?* Yes, it will be continually updated.

- *If others want to extend/augment/build on/contribute to the dataset, is there a mechanism for them to do so?* Yes, they can do so with the GitHub platform.

## A.4 Copyright and Licensing Terms

This work is distributed under a CC BY 4.0 license. The implementation of the code references open-source projects such as Qiskit, QuantumKatas, Cirq, and NWQBench. We bear responsibility in case of violation of rights.

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

 include detailed analysis of the experiments and additional experiment results. In Section C.1, we introduce the metrics: BLEU score, verification score, and byte perplexity, and provide a detailed analysis for the experiments on BLEU and verification score. In Section C.2, we include additional experimental results. In Section C.3, we present concrete cases of typical patterns observed in model outputs.

### C.1  Metrics

**BLEU Score.**  Bilingual Evaluation Understudy (BLEU) score is a metric used to evaluate the quality of machine-translated text compared to human-translated text. It measures how close the machine translation is to one or more reference translations. The BLEU score evaluates the quality of text generated by comparing it with one or more reference texts. It does this by calculating the n-gram precision, which means it looks at the overlap of n-grams (contiguous sequences of n words) between the generated text and the reference text. Originally the BLEU score ranges from 0 to 1, where 1 indicates a perfect match with the reference translations. Here rescaling the score makes it ranges from 0 to 100.

The BLEU score, originally designed for machine translation, can also be effectively used for evaluating algorithm generation tasks. Just as BLEU measures the similarity between machine-translated text and human reference translations, it can measure the similarity between a generated algorithm and a gold-standard algorithm. This involves comparing sequences of tokens to assess how closely the generated output matches the reference solution. In the context of algorithm generation, n-grams can represent sequences of tokens or operations in the code. BLEU score captures the precision of these n-grams, ensuring that the generated code aligns closely with the expected sequences found in the reference implementation.

The formula for BLEU score is given by:

$$\text{BLEU} = \text{BP} \cdot \exp\left(\sum_{n=1}^{N} w_n \log p_n\right).$$

where BP is the acronym for brevity penalty, $w_n$ is the weight for the n-gram precision (typically $\frac{1}{N}$ for uniform weights), $p_n$ is the precision for n-grams. BP is calculated as:

$$BP = \begin{cases} 1 & \text{if } c > r \\ e^{1-\frac{r}{c}} & \text{if } c \leq r \end{cases}.$$

where $c$ is the length of the generated text and $r$ is the length of the reference text. Furthermore, n-gram precision $p_n$ is calculated as:

$$p_n = \frac{\sum_{C \in \text{Candidates}} \sum_{n-\text{gram} \in C} \min(\text{Count}(n-\text{gram in candidate}), \text{Count}(n-\text{gram in references}))}{\sum_{C \in \text{Candidates}} \sum_{n-\text{gram} \in C} \text{Count}(n-\text{gram in candidate})}.$$

This formulation ensures that the BLEU score takes into account both the precision of the generated n-grams and the overall length of the translation, providing a balanced evaluation metric.

In our experiments, the BLEU scores for various quantum algorithm design tasks are illustrated in Figure 3(a). This figure not only displays the average performance of each model but also highlights the differences in performance across individual quantum algorithm tasks. The first notable observation is that the figure clearly demonstrates the varying levels of difficulty among quantum algorithms. For example, models achieve higher BLEU scores on tasks such as Bernstein-Vazirani and Deutsch-Jozsa, whereas they perform significantly worse on tasks like Grover, phase estimation, and quantum Fourier transform. This indicates that the former tasks are considerably easier than the latter ones. Another significant observation is that most models score higher in a five-shot prompt compared to a one-shot prompt, which confirms the large language models' ability to improve performance through contextual learning.

Similar patterns are observed in oracle construction tasks, as illustrated in Figure 3(b). The figure highlights that the Diffusion Operator task is notably more challenging than the Grover oracle construction task. Interestingly, we found that adding more in-context examples actually reduced the performance of the Qwen 2.5 and DeepSeek-R1 models. This decline in performance could be attributed to the significant differences between each oracle construction task, which may be too out-of-distribution. Consequently, the additional examples might cause the models to overfit to the specific examples provided in the context, rather than generalizing well across different tasks.

**Detailed Analysis of Verification Score.** In addition to evaluating the BLEU score, we conducted an experiment to measure the correctness of the machine-generated algorithms, and the results are shown in Table 1. By running a verification function, we discovered that phase estimation and the swap test are significantly more challenging than other problems, leading most models to score -1 (indicating they cannot even generate the correct syntax). Notably, the BLEU score for the swap test is above average compared to other algorithms, yet almost none of the models produced a correct algorithm. This discrepancy highlights a critical limitation of using BLEU as a metric for algorithm evaluation. BLEU measures average similarity, but even a single mistake in an algorithm can render it entirely incorrect, thus failing to capture the true accuracy and functionality of the generated algorithms. Another important finding is that in a five-shot setting, GPT-4 and GPT-3.5 surpass all other models by a large margin. This demonstrates their exceptional capabilities, particularly in long-context comprehension and in-context learning. These models not only excel in understanding and generating text based on minimal examples but also maintain high performance over extended sequences, highlighting their advanced architecture and training methodologies.

As variational algorithms with parametric quantum circuits, VQE and QAOA require specifically designed metrics. For VQE, we compare the energy obtained from the machine-generated ansatz with the ground truth and compute a correctness score as follows:

$$1 - \frac{|E_{\text{LLM}} - E_{\text{expected}}|}{|E_{\text{expected}}|} \tag{4}$$

This ratio-based metric is used because VQE optimizes in a continuous space, where solutions are approximations rather than exact values. In contrast, for QAOA, when applied to solving the MaxCut problem in a discrete space, is directly evaluated against the ground truth partition, with $0$ or $1$ correctness score.

Since quantum algorithms based on parametric quantum circuits often share a common structure in their variational optimization process, a refined verification function is required to evaluate both the quantum ansatz and its associated classical optimization step. Therefore, the final verification score is decomposed into two components: one for the quantum circuit (QASM) generation and another for the optimization code (Python implementation). The evaluation criteria are as follows:

- **QASM Syntax Check:**
    - If the machine-generated QASM contains syntax errors, the score is $-1$.
- **Optimization Code Check:**
    - If the QASM is valid but the Python code has syntax errors, the QASM output is evaluated using a ground truth implementation of the optimization code.
    - If the result matches the ground truth, the score is $0.5 \times correctness\_score$.
    - If the result is incorrect, the score is $0$.
    - If further syntax errors occur during evaluation, the score remains $-1$.

We found that GPT-4 outperforms other models in ansatz design. However, most models frequently fail to generate correct optimization code, often encountering syntax errors. A detailed analysis suggests that a major source of these errors is inconsistency in qiskit versions, leading to incorrect function calls and deprecated API usage.

The verification results of the oracle construction task, as shown in Table 3, confirm our previous conclusions. In the five-shot setting, GPT-4 and GPT-3.5 consistently outperform all other models. Additionally, this table highlights the inconsistency between BLEU scores and verification scores. For instance, while the Diffusion Operator task achieves the lowest BLEU score, it is the Grover oracle construction that receives the lowest verification score. This discrepancy suggests that BLEU scores may not fully capture the performance of models in certain complex tasks, and it is necessary to include verification score as a comprehensive evaluation.

**Byte Perplexity.** Perplexity is a measure of how well a probability distribution or a probabilistic model predicts a sample. In the context of language models, it quantifies the uncertainty of the model when it comes to predicting the next element in a sequence. Byte perplexity specifically deals with sequences of bytes, which are the raw binary data units used in computer systems. For our purposes, we consider byte perplexity under UTF-8 encoding, a widely used character encoding standard that represents each character as one or more bytes.

For a given language model, let $p(x_i|x_{<i})$ be the probability of the $i$-th byte $x_i$ given the preceding bytes $x_{<i}$. If we have a sequence of bytes $x = (x_1, x_2, \ldots, x_N)$, the perplexity $\text{PPL}(x)$ of the model on this sequence is defined as:

$$\text{PPL}(x) = 2^{-\frac{1}{N} \sum_{i=1}^{N} \log_2 p(x_i|x_{<i})}.$$

A notable feature of byte perplexity is that, it does not rely on any specific tokenizer, making it versatile for comparing different models. Therefore, byte perplexity can be used to measure the performance in quantum algorithm generation tasks. In such tasks, a lower byte perplexity indicates a better-performing model, as it means the model is more confident in its predictions of the next byte in the sequence.

### C.2 Additional Experimental Results

**Byte perplexity (PPL) scores.** The Byte Perplexity results, shown in Figure 7, provide valuable insights into the performance of our model. Evaluated in a zero-shot setting, byte perplexity trends closely mirror those observed with BLEU scores. This alignment suggests that our model's predictive capabilities are consistent across Perplexity and BLEU evaluation metrics. Specifically, in the context of quantum algorithm design tasks, the results indicate that the Bernstein-Vazirani and Deutsch-Jozsa algorithms are relatively straightforward for the model, whereas the Simon algorithm presents greater difficulty. This differentiation highlights the varying levels of complexity inherent in these quantum algorithms.

**Oracle construction.** The details are presented in Table 3. GPT-4o and GPT-3.5 consistently outperform other models, with both showing substantial improvements under few-shot prompting. GPT-4o achieves the highest overall performance, raising its average score from -0.3912 (1-shot) to -0.1245 (5-shot). GPT-3.5 follows closely, improving from -0.5910 to -0.2474. They are the only models capable of generating partially correct solutions with positive scores for challenging tasks such as the diffusion operator, showcasing strong in-context learning and generalization capabilities of the GPT series. In contrast, Qwen 2.5 struggles to generalize, with only marginal improvement from -0.6216 to -0.5258 and persistent failures on advanced tasks like Grover and Generalized-Simon. DeepSeek-R1 performs the worst overall, with highly negative scores in both settings. Its long-chain reasoning often leads to outputs that exceed the maximum context length, resulting in truncated or invalid circuits, highlighting the inefficiency of reasoning models for oracle construction tasks.

**Temperature.** Regarding the counter-intuitive phenomenon where the performance on Clifford and universal random circuits decreases after fine-tuning, we conducted additional experiments and fine-tuned the model on 4,800 samples specifically for the Clifford task. Upon closer inspection, we observed that the model more frequently generated outputs with infinite loops and increased

Table 3: Benchmarking oracle construction in verification function scores.

| Model | Shot | Bernstein-Vazirani | Deutsch-Jozsa | Diffusion-Operator | Grover | Simon | Clifford | Universal | Generalized-Simon (multi-str) | Avg |
|---|---|---|---|---|---|---|---|---|---|---|
| gpt4o | 1 | 0.3600 (±0.0659) | 0.1600 (±0.0801) | -1.0000 (±0.0000) | -0.9540 (±0.0323) | -0.4348 (±0.0542) | -0.4348 (±0.0224) | -0.1144 (±0.0341) | -0.7188 (±0.0808) | -0.3912 |
| gpt4o | 5 | 0.5400 (±0.0521) | 0.3700 (±0.0677) | 0.0769 (±0.2878) | -0.9770 (±0.0230) | -0.1739 (±0.0453) | 0.1052 (±0.0210) | -0.1111 (±0.0361) | -0.6250 (±0.0870) | -0.1245 |
| Llama3 | 1 | -0.7600 (±0.0571) | -0.7000 (±0.0595) | -0.8462 (±0.1538) | -0.9770 (±0.0230) | -0.8261 (±0.0397) | -0.1862 (±0.0349) | -0.1424 (±0.0338) | -0.8438 (±0.0652) | -0.6602 |
| Llama3 | 5 | 0.0300 (±0.0771) | -0.3400 (±0.0807) | -1.0000 (±0.0000) | -0.9310 (±0.0394) | -0.3587 (±0.0503) | -0.1348 (±0.0325) | -0.1572 (±0.0329) | -0.0313 (±0.0951) | -0.3654 |
| gpt3.5 | 1 | -0.1000 (±0.0859) | -0.1100 (±0.0764) | -1.0000 (±0.0000) | -0.9540 (±0.0323) | -0.4130 (±0.0561) | 0.0650 (±0.0178) | 0.0538 (±0.0190) | -0.9688 (±0.0313) | -0.5910 |
| gpt3.5 | 5 | 0.2700 (±0.0723) | 0.1300 (±0.0734) | 0.2308 (±0.2809) | -0.7701 (±0.0688) | -0.3043 (±0.0482) | 0.0816 (±0.0163) | 0.0723 (±0.0159) | -0.5625 (±0.0891) | -0.2474 |
| Qwen 2.5 | 1 | -0.5100 (±0.0689) | -0.5500 (±0.0687) | -0.8462 (±0.1538) | -0.8391 (±0.0587) | -0.9891 (±0.0109) | -0.1065 (±0.0345) | -0.1318 (±0.0337) | -1.0000 (±0.0000) | -0.6216 |
| Qwen 2.5 | 5 | -0.0900 (±0.0889) | -0.4000 (±0.0739) | -0.3846 (±0.2665) | -0.8391 (±0.0587) | -0.8913 (±0.0326) | -0.2895 (±0.0442) | -0.3434 (±0.0415) | -0.9688 (±0.0313) | -0.5258 |
| DeepSeek-R1 | 1 | -0.8900 (±0.0424) | -0.9800 (±0.0141) | -1.0000 (±0.0000) | -1.0000 (±0.0000) | -1.0000 (±0.0000) | -0.8885 (±0.0000) | -0.8644 (±0.0232) | -1.0000 (±0.02589) | -0.9529 |
| DeepSeek-R1 | 5 | -0.6700 (±0.0697) | -0.7900 (±0.0518) | -0.8462 (±0.1538) | -0.9310 (±0.0394) | -0.9565 (±0.0214) | -0.4355 (±0.0701) | -0.5496 (±0.0375) | -0.8125 (±0.0369) | -0.7489 |

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

> **Prompt for Bernstein-Vazirani Problem with Qiskit**
>
> Please write the code with Qiskit implementing Bernstein-Vazirani Algorithm with qubit number n = 5.

The model answered with the following text and codes:

1749

**Text Answer by GPT4o**

Here's a Python code using Qiskit to implement the Bernstein-Vazirani algorithm with $n = 5$ qubits. The secret string used in this example is $s = 10101$ (you can change this to any string of your choice).

```python
from qiskit import QuantumCircuit, Aer, transpile, assemble
from qiskit.visualization import plot_histogram

# Secret string s for the Bernstein-Vazirani problem
s = '10101'

# Number of qubits
n = len(s)

# Create a Quantum Circuit with n+1 qubits (one ancillary qubit)
bv_circuit = QuantumCircuit(n+1, n)

# Apply Hadamard gates to all qubits except the ancillary qubit
bv_circuit.h(range(n))

# Apply X and Hadamard gate to the ancillary qubit
bv_circuit.x(n)
bv_circuit.h(n)

# Construct the Oracle
for i, bit in enumerate(s):
    if bit == '1':
        bv_circuit.cx(i, n)

# Apply Hadamard gates to the first n qubits
bv_circuit.h(range(n))
# Measure the first n qubits
bv_circuit.measure(range(n), range(n))
# Visualize the circuit
bv_circuit.draw('mpl')

# Run the circuit using Qiskit Aer simulator
simulator = Aer.get_backend('qasm_simulator')
t_qc = transpile(bv_circuit, simulator)
qobj = assemble(t_qc)
result = simulator.run(qobj).result()

# Get the result counts and plot the histogram
counts = result.get_counts()
plot_histogram(counts)
```

1791

Listing 14: Qiskit code for Bernstein-Vazirani Problem generated by GPT4o in the zero-shot setting.

1792  This code, although adopts an older version of Qiskit syntax, can be compiled and run successfully
1793  with the correct answer. Similar performances can be observed with OpenQASM codes:

```
// OpenQASM 2.0
OPENQASM 2.0;
include "qelib1.inc";

// Declare 5 qubits for input and 1 ancilla qubit
qreg q[6];
creg c[5]; // classical register for measuring the result

// Step 1: Apply Hadamard gates to the input qubits
h q[0];
h q[1];
h q[2];
```

```
h q[3];
h q[4];

// Apply Hadamard to the ancillary qubit and set it to |1>
x q[5];
h q[5];

// Step 2: Oracle implementation (CNOT gates based on secret string s
    ↪ = 10110)
cx q[0], q[5]; // Secret string bit 1
// No cx for q[1] because the secret string bit is 0
cx q[2], q[5]; // Secret string bit 1
cx q[3], q[5]; // Secret string bit 1
// No cx for q[4] because the secret string bit is 0

// Step 3: Apply Hadamard gates to the input qubits again
h q[0];
h q[1];
h q[2];
h q[3];
h q[4];

// Step 4: Measure the qubits to get the result
measure q[0] -> c[0];
measure q[1] -> c[1];
measure q[2] -> c[2];
measure q[3] -> c[3];
measure q[4] -> c[4];
```

Listing 15: OpenQASM code for Bernstein-Vazirani Problem generated by GPT4o.

This phenomenon contrasts with the results in Appendix C.3.2. Considering that Qiskit has implemented tutorial codes for several representative quantum algorithms, the seemingly outstanding performance of quantum algorithm code generation might be a deceptive result caused by data contamination. Therefore, only a carefully designed quantum algorithm dataset can avoid the effects of data contamination, allowing for an effective evaluation of the model's genuine capability in quantum algorithm design and implementation. This dataset is also meaningful for testing general AI code generation and syntax learning, where no existing AI dataset could substitute us.