# OpenReview forum: "QCircuitBench: A Large-Scale Dataset for Benchmarking Quantum Algorithm Design"
_NeurIPS.cc/2025/Datasets_and_Benchmarks_Track — NeurIPS 2025 Datasets and Benchmarks Track poster_

### Official Review · Reviewer_A8iu · 2025-06-26

**Rating:** 5
**Confidence:** 4

**Summary:**

The submission introduces QCircuitBench, the first benchmark dataset designed to evaluate AI's capability in designing and implementing quantum algorithms as quantum circuit codes. Its main contributions are as follows:
1. A general framework for formulating quantum algorithm design tasks for large language models (LLMs), covering problem descriptions, quantum circuit codes, classical post-processing, and verification functions.
2. Implementation of 3 task suites, 23 algorithms, and 128,573 data points, ranging from basic primitives to advanced applications like the Generalized Simon’s Problem.
3. Automatic validation and verification functions to enable iterative, human-free evaluation.
4. Potential as a training dataset, with fine-tuning experiments revealing that LLMs have limitations in this domain and may exhibit consistent error patterns.

**Additional Feedback:**

- Q1: Develop unified interface specifications across frameworks, such as standardized interfaces for quantum gate definition, circuit construction, and measurement operations, to ensure that algorithm implementations are portable across different frameworks.
- Q2: Quantum algorithm evaluation metrics have been added, such as quantifying the degree of entanglement of qubits in a circuit, and measuring the quantum characteristics of the algorithm.
- Q3: Provide a complete set of visualizations in the appendix, including quantum circuits for all algorithms, as well as the quantum circuits generated by LLMs for different algorithm tasks.
- Q4: Variational quantum algorithms (VQAs) such as VQE and QAOA have low success rates in benchmarks, and models often fail to generate correct parameterized circuits, so more variational quantum algorithms may be added to extend QCircuitBench to increase the success rates.
- Q5: Will the dataset be expanded to support other quantum programming frameworks, and what challenges do you anticipate in this process?

**Dataset Code Accessibility:**

Yes

**Dataset Code Comments:**

The directory structure is documented in Appendix A, detailing organized subfolders for each task suite (e.g., Oracle Construction, Algorithm Design) and algorithm (e.g., Simon’s problem), with clear file formats (QASM, Python, JSON), ensuring the reproducibility and collaboration in quantum AI. The dataset is publicly hosted in a standard format, accompanied by executable code and detailed documentation, and the benchmarking protocols are transparent. These ensure researchers can readily use, validate, and extend the work, aligning with open scientific practices.

**Ethical Considerations:**

No, there are no or only very minor ethics concerns

**Final Justification:**

The authors thoroughly addressed all of my initial concerns. As a result, I have raised my score and now confidently recommend acceptance. Here is a summary of how my concerns were addressed:

1.  My primary concern was the benchmark's limitation to Qiskit, which impacted its generalizability. The authors have since integrated Cirq, providing new experimental results and committing to full coverage in the final version. This was a substantive update that fully resolved my main reservation.
2.  I had questioned the evaluation metrics. The authors' clarification was convincing; they explained that the `semantic score` (based on state fidelity or task success rate) is indeed a direct and appropriate measure of quantum performance for the specific tasks in their benchmark. Their reasoning is sound, and I consider this point resolved.
3. I pointed out the need for visualizations of the quantum circuits. The authors have agreed to add these to the final paper and provided text-based examples in the rebuttal as a placeholder. This commitment is sufficient to address my feedback.

There are no remaining unresolved issues from my perspective. The most heavily weighted issue was the benchmark's generalizability, and the addition of Cirq support was the primary driver for my score increase. The successful resolution of my other points further strengthened this decision.

**Limitations Weaknesses:**

1. The dataset focuses on OpenQASM 3.0 and Qiskit, ignoring other quantum programming frameworks (e.g., Cirq, ProjectQ) or hardware-specific languages. This limits generalizability to diverse quantum platforms.
2. The verification function only checks the circuit syntax and measurement results. There is a lack of evaluation indicators specific to quantum algorithms. Since it is aimed at generating quantum algorithms, it is necessary to add evaluation metrics specific to quantum algorithms to evaluate the performance of large language models.
3. For the experiments, the paper does not adequately present the visualization results of the generated circuits. The generated results are only shown in code in the appendix, but the visualization gives the reader more readability.

**Strengths Contributions:**

1. QCircuitBench stands out for its novel focus on AI-driven quantum algorithm design, filling a critical research gap.
2. The paper’s clear presentation, comprehensive dataset, and rigorous benchmarks establish it as a foundational resource for advancing AI in quantum computing.
3. Its distinction from prior work is well-justified, and the experimental results provide actionable insights for future research.

---

> ### Author Rebuttal · Authors · 2025-07-31
>
> We thank the reviewer for acknowledging the novelty of QCircuitBench, the insights from the experimental results, and the potential to serve as a foundational resource for advancing AI in quantum computing.
>
> (1)  We appreciate the reviewer's valuable suggestion to integrate diverse quantum platforms. We have now implemented 8 algorithms in Cirq and plan to extend coverage to all algorithms and additional platforms in the final version. In the Cirq implementation, both the algorithm circuits and post-processing functions are written entirely in Python using Cirq. The circuits are constructed gate by gate, and post-processing leverages Cirq’s simulator and measurement tools. The Oracle / Gate Definition component is no longer required, as these gates are now generated dynamically within the verification functions.
>
>  Here are the benchmarking results.
>
> **Table 6: Benchmarking Results with Cirq Version for Quantum Algorithm Design**
> | metric               | Bernstein Vazirani | Deutsch Jozsa    | QFT              | Simon             | Random Number Generator | GHZ               | Swap Test         | W State           |
> | :------------------- | :----------------- | :--------------- | :--------------- | :---------------- | :---------------------- | :---------------- | :---------------- | :---------------- |
> | **semantic score**   | 1.0000 (±0.0000)   | 1.0000 (±0.0000) | 0.2055 (±0.0883) | 0.0000 (±0.0000)  | 0.0000 (±0.0000)        | 0.0000 (±0.0000)  | 0.0000 (±0.0000)  | 0.0000 (±0.0000)  |
> | **qasm syntax**      | 1.0000 (±0.0000)   | 1.0000 (±0.0000) | 0.8000 (±0.2000) | 1.0000 (±0.0000)  | 0.0000 (±0.0000)        | 0.0000 (±0.0000)  | 0.0000 (±0.0000)  | 0.2000 (±0.2000)  |
> | **code syntax**      | 1.0000 (±0.0000)   | 0.8333 (±0.1667) | 0.8000 (±0.2000) | 1.0000 (±0.0000)  | 0.0000 (±0.0000)        | 0.0000 (±0.0000)  | 1.0000 (±0.0000)  | 0.0000 (±0.0000)  |
> | **gate count ratio** | 1.0000 (±0.0000)   | 1.0000 (±0.0000) | 0.8798 (±0.0128) | 1.3954 (±0.2536)  | N/A (±N/A)              | N/A (±N/A)        | N/A (±N/A)        | N/A (±N/A)        |
> | **shot ratio**       | 1.0000 (±0.0000)   | 1.0000 (±0.0000) | 1.0000 (±0.0000) | 1.0000 (±0.0000)  | N/A (±N/A)              | N/A (±N/A)        | N/A (±N/A)        | N/A (±N/A)        |
> | **time ratio**       | 1.0283 (±0.0042)   | 0.9992 (±0.0124) | 1.0459 (±0.0234) | 1.3049 (±0.0092)  | N/A (±N/A)              | N/A (±N/A)        | N/A (±N/A)        | N/A (±N/A)        |
>
> (2)  We thank the reviewer for raising this point. We would like to clarify that our verification function assesses not only circuit syntax and measurement correctness but also **quantum semantics**. Specifically, for semantic evaluation, the final score lies in [0, 1], reflecting either the fidelity of the quantum state (for state preparation and random circuit synthesis tasks) or the success rate on test cases (for algorithm design and oracle construction tasks). This task-specific metric provides a precise and meaningful assessment of the quantum semantics relevant to each algorithm. As our quantum algorithm design tasks primarily target quantum speedups for classical problems, metrics such as the degree of entanglement of qubits may not apply to this setting.
>
> (3) We thank the reviewer for the advice to add visualizations of generated circuits. For each algorithm, we have drawn both the reference circuit and a representative sample generated by LLMs. However, due to the rebuttal policy prohibiting PDF uploads and external links, we are unable to display the figures here. As a simple alternative, we provide several text-based circuit diagrams below. The full visualizations will be included in the final version.
>
> Bernstein-Vazirani Algorithm (n=6):
>
> - Reference / Model:
>
>   ```text
>        ┌───┐     ┌─────────┐┌───┐┌─┐
>   q_0: ┤ H ├─────┤0        ├┤ H ├┤M├───────────────
>        ├───┤     │         │├───┤└╥┘┌─┐
>   q_1: ┤ H ├─────┤1        ├┤ H ├─╫─┤M├────────────
>        ├───┤     │         │├───┤ ║ └╥┘┌─┐
>   q_2: ┤ H ├─────┤2        ├┤ H ├─╫──╫─┤M├─────────
>        ├───┤     │         │├───┤ ║  ║ └╥┘┌─┐
>   q_3: ┤ H ├─────┤3 Oracle ├┤ H ├─╫──╫──╫─┤M├──────
>        ├───┤     │         │├───┤ ║  ║  ║ └╥┘┌─┐
>   q_4: ┤ H ├─────┤4        ├┤ H ├─╫──╫──╫──╫─┤M├───
>        ├───┤     │         │├───┤ ║  ║  ║  ║ └╥┘┌─┐
>   q_5: ┤ H ├─────┤5        ├┤ H ├─╫──╫──╫──╫──╫─┤M├
>        ├───┤┌───┐│         │└───┘ ║  ║  ║  ║  ║ └╥┘
>   q_6: ┤ X ├┤ H ├┤6        ├──────╫──╫──╫──╫──╫──╫─
>        └───┘└───┘└─────────┘      ║  ║  ║  ║  ║  ║
>   c: 6/═══════════════════════════╩══╩══╩══╩══╩══╩═
>                                   0  1  2  3  4  5
>   ```
>
> Quantum Fourier Transform (n=5):
>
> - Reference:
>
>   ```text
>        ┌──────┐                                                                 »
>   q_0: ┤0     ├──────■─────────────────────────────────────────■────────────────»
>        │      │      │                                         │                »
>   q_1: ┤1     ├──────┼─────────■───────────────────────────────┼────────■───────»
>        │      │      │         │                               │        │       »
>   q_2: ┤2 Psi ├──────┼─────────┼────────■──────────────────────┼────────┼───────»
>        │      │      │         │        │                ┌───┐ │P(π/8)  │P(π/4) »
>   q_3: ┤3     ├──────┼─────────┼────────┼────────■───────┤ H ├─■────────■───────»
>        │      │┌───┐ │P(π/16)  │P(π/8)  │P(π/4)  │P(π/2) └───┘                  »
>   q_4: ┤4     ├┤ H ├─■─────────■────────■────────■──────────────────────────────»
>        └──────┘└───┘                                                            »
>   «                                                   ┌───┐
>   «q_0: ───────────────■──────────────────────■───────┤ H ├─X─
>   «                    │                ┌───┐ │P(π/2) └───┘ │
>   «q_1: ───────────────┼────────■───────┤ H ├─■─────────X───┼─
>   «              ┌───┐ │P(π/4)  │P(π/2) └───┘           │   │
>   «q_2: ─■───────┤ H ├─■────────■───────────────────────┼───┼─
>   «      │P(π/2) └───┘                                  │   │
>   «q_3: ─■──────────────────────────────────────────────X───┼─
>   «                                                         │
>   «q_4: ────────────────────────────────────────────────────X─
>   «
>   ```
>
> - Model:
>
>   ```text
>        ┌──────┐┌───┐                                                            »
>   q_0: ┤0     ├┤ H ├─■────────■─────────────■─────────────────■─────────────────»
>        │      │└───┘ │P(π/2)  │       ┌───┐ │                 │                 »
>   q_1: ┤1     ├──────■────────┼───────┤ H ├─┼────────■────────┼─────────■───────»
>        │      │               │P(π/4) └───┘ │        │P(π/2)  │         │       »
>   q_2: ┤2 Psi ├───────────────■─────────────┼────────■────────┼─────────┼───────»
>        │      │                             │P(π/8)           │         │P(π/4) »
>   q_3: ┤3     ├─────────────────────────────■─────────────────┼─────────■───────»
>        │      │                                               │P(π/16)          »
>   q_4: ┤4     ├───────────────────────────────────────────────■─────────────────»
>        └──────┘                                                                 »
>   c: 5/═════════════════════════════════════════════════════════════════════════»
>                                                                                 »
>   «                                                                    ┌─┐
>   «q_0: ───────────────────────────────────────────────────────X───────┤M├───
>   «                                                            │ ┌─┐   └╥┘
>   «q_1: ──────■────────────────────────────────────────────X───┼─┤M├────╫────
>   «     ┌───┐ │                                       ┌─┐  │   │ └╥┘    ║
>   «q_2: ┤ H ├─┼────────■────────■─────────────────────┤M├──┼───┼──╫─────╫────
>   «     └───┘ │        │P(π/2)  │       ┌───┐         └╥┘  │   │  ║ ┌─┐ ║
>   «q_3: ──────┼────────■────────┼───────┤ H ├─■────────╫───X───┼──╫─┤M├─╫────
>   «           │P(π/8)           │P(π/4) └───┘ │P(π/2)  ║ ┌───┐ │  ║ └╥┘ ║ ┌─┐
>   «q_4: ──────■─────────────────■─────────────■────────╫─┤ H ├─X──╫──╫──╫─┤M├
>   «                                                    ║ └───┘    ║  ║  ║ └╥┘
>   «c: 5/═══════════════════════════════════════════════╩══════════╩══╩══╩══╩═
>   «                                                    2          1  3  0  4
>   ```
>
> W State (n=2):
>
> - Reference:
>
>   ```text
>        ┌──────────┐   ┌─────────┐
>   q_0: ┤ Ry(-π/4) ├─■─┤ Ry(π/4) ├──■──
>        └──┬───┬───┘ │ └─────────┘┌─┴─┐
>   q_1: ───┤ X ├─────■────────────┤ X ├
>           └───┘                  └───┘
>   c: 2/═══════════════════════════════
>
>   ```
>
> - Model:
>
>   ```text
>        ┌───┐     ┌──────────┐     ┌─────────┐┌─┐
>   q_0: ┤ H ├──■──┤ Ry(-π/4) ├──■──┤ Ry(π/4) ├┤M├
>        └───┘┌─┴─┐└──────────┘┌─┴─┐└───┬─┬───┘└╥┘
>   q_1: ─────┤ X ├────────────┤ X ├────┤M├─────╫─
>             └───┘            └───┘    └╥┘     ║
>   c: 2/════════════════════════════════╩══════╩═
>                                        1      0
>   ```
>
> (4)  We appreciate the suggestion to include more variational quantum algorithms. We will add more VQAs in the final version, and see if it helps improve the success rate.
>
> (5)  We have now integrated support for Cirq, as mentioned in (1). We plan to expand compatibility to other quantum programming frameworks in the final version. We anticipate that this will primarily involve engineering efforts, rather than facing fundamental obstacles.

---

> > ### Comment · Reviewer_A8iu · 2025-08-05
> >
> > Thank you to the authors for the detailed and thoughtful rebuttal. Your responses have thoroughly addressed the concerns I raised in my initial review. I am particularly impressed by the prompt integration of Cirq into the benchmark, which directly addresses my main point about generalizability. The clarification regarding the semantic score as a task-specific quantum metric is well-reasoned and satisfactory, and I appreciate the commitment to include full circuit visualizations in the final version. Given that my primary concerns have been effectively resolved, I have raised my score and now support the acceptance of this paper.
> >
> > I have a question that is not related to the paper, I want to know how the author uses words to draw quantum circuits. This is a good presentation.

---

> > > ### Author Response · Authors · 2025-08-05
> > >
> > > We thank the reviewer for the response and are glad that our rebuttal has addressed your concerns.
> > >
> > > The quantum circuits are visualized using Qiskit’s built-in drawing tool. Below is a code snippet demonstrating the usage:
> > > ```
> > > from qiskit import QuantumCircuit
> > > circuit = QuantumCircuit(3, 3)
> > > circuit.x(1)
> > > circuit.h(range(3))
> > > circuit.cx(0, 1)
> > > circuit.measure(range(3), range(3))
> > > print(circuit) # Or use circuit.draw()
> > > ```
> > > This code produces the following text-based circuit diagram, which can be displayed in Markdown using a ```text block:
> > > ```text
> > >      ┌───┐          ┌─┐
> > > q_0: ┤ H ├───────■──┤M├───
> > >      ├───┤┌───┐┌─┴─┐└╥┘┌─┐
> > > q_1: ┤ X ├┤ H ├┤ X ├─╫─┤M├
> > >      ├───┤└┬─┬┘└───┘ ║ └╥┘
> > > q_2: ┤ H ├─┤M├───────╫──╫─
> > >      └───┘ └╥┘       ║  ║
> > > c: 3/═══════╩════════╩══╩═
> > >             2        0  1
> > > ```
> > > Qiskit also supports circuit visualization as a `matplotlib.Figure` object via `circuit.draw(output="mpl")`, which we plan to use for circuit visualization in the final version.

---

### Official Review · Reviewer_igHa · 2025-07-01

**Rating:** 4
**Confidence:** 3

**Summary:**

The paper proposes a novel benchmark for evaluating the knowledge of LLMs on the quantum algorithm design. The proposed benchmark consists of three tasks, including oracle construction, quantum algorithm design, and random circuit synthesis. Extensive experiments are conducted to verify the quantum knowledge of current mainstream LLMs.

**Dataset Code Accessibility:**

Partly

**Dataset Code Comments:**

The dataset is released but not the full code.

**Ethical Considerations:**

No, there are no or only very minor ethics concerns

**Final Justification:**

Authors' rebuttal well address my concerns.

**Limitations Weaknesses:**

(1) The writing can be improved. For example, authors can consider adding more related work and background to let more readers quickly understand. The table and figures can also be polished to improve the readability.   Some citations are also not correct, for example the line 128( [Ambainis, 2004]), line 138 (uhrman et al., 2001]).
(2)  Main conclusions and takeaways of the experiments can be highlighted.
(3) Lack of human performance to benchmark the quality of the proposed dataset.
(4) Lack of evaluation code, as well as the data processing code for paper reproduction.  While it is the dataset and benchmark track, I believe the open-source would be important to evaluate a work.   A github repo with a detailed README file is also suggested.
(5) While correctness is essential, the evaluation of the efficiency of generated codes/algorithms is also important. I suggest authors include this evaluation in the work.
(6) The knowledge of quantum computing might be limited in the pre-training dataset. While the proposed benchmark uses the few-shot setting for QA task, I believe the open-book setting, or building a RAG system, might be more efficient to solve these questions, rather than simply fine-tuning.

**Strengths Contributions:**

(1) The task is novel and interesting.
(2) Extensive experiments are conducted to verify the ability of different LLMs.
(3) Fine-tuning experiments are conducted to derive new insights and suggest the difficulty in this task.

---

> ### Author Rebuttal · Authors · 2025-07-31
>
> We thank the reviewer for acknowledging the novelty and relevance of the task, the thoroughness of our benchmarking experiments, and the insights derived from the fine-tuning results.
>
> (1)  We thank the reviewer for the suggestions with respect to the writing. In the submitted version, we have included a Related Work section in Section 2 and a Background section in Appendix B, and we will further refine these sections in the final version. Additionally, we have polished the tables and figures for better readability.
>
> We have also corrected the citations: "[Ambainis, 2004] for 3-SAT problem, and *Quantum Algorithms for the Triangle Problem*[Magniez et al., 2005] for the triangle finding problem". We believe "[Buhrman et al., 2001]" is a proper citation for the SWAP Test problem - Figure 1 in its PRL version of this paper explicitly presents the SWAP Test circuit.
>
> (2)  Here are the highlights of main conclusions and takeaways of the experiments, which will be added in the final version.
>
> - **Main Conclusions:**
>   - QCircuitBench introduces the first comprehensive benchmark dataset tailored for AI-driven quantum algorithm design, addressing a critical research gap at the intersection of quantum computing and LLMs, with valuable implications for both communities.
>   - The dataset defines a general framework which formulates the key components of quantum algorithm design for LLMs, with careful considerations for various subtle challenges. Its code-generation perspective, modular and extensible structure, automatic verification functions, and broad coverage of diverse quantum algorithms together establish a rigorous benchmark for evaluating the capabilities of LLMs in quantum algorithm design.
>   - Extensive experiments on mainstream LLMs reveal both their potential and limitations. While benchmark results expose consistent error patterns and open challenges, fine-tuning experiments demonstrate early promise in helping models internalize structural patterns. These insights position QCircuitBench not only as a benchmark, but also as a forward-looking testbed for developing and training next-generation AI models for quantum computing.
> - **Takeaways of Experiments:**
>   - The experimental results indicate that QCircuitBench is challenging even for the most advanced LLMs available today, making it a valuable benchmark for AI models.
>   - We identify three predominant types of errors made by LLMs: (1) *improvisation errors*, where models hallucinate or deviate from task constraints, such as syntax or required structure; (2) *counting errors*, particularly in qubit or gate indices; and (3) *data contamination*, likely stemming from prior exposure to known implementations.
>   - Fine-tuning experiments demonstrate early promise, showing that the dataset can help models internalize structural patterns in oracle construction tasks.
>
> (3)  We thank the reviewer for the suggestion. In terms of human evaluation of the quality of the dataset, we have included **human-examined case studies** in Section 4.1 *Types of Errors Made by LLMs* and Appendix C.3, which provide **manual analysis** of model performance on the dataset. If the reviewer is referring to inviting human participants to solve the dataset tasks to derive a human performance baseline, we note that due to time constraints during the rebuttal period, a comprehensive human study is not feasible at this stage. We plan include such results in the final version.
>
> (4)  We would like to point out that the evaluation code and data processing code **are provided in the .zip file submitted**. They are included in the directory of each algorithm, named as *{algorithm_name}_verification.py*, *{algorithm_name}_generation.py*, and *{algorithm_name}_dataset.py*. A detailed description of the dataset structure, serving as a README, is provided in Appendix A. The benchmarking and fine-tuning scripts follow standard practice. We are happy to provide these, or any additional code the reviewer would like us to release.
>
> We did not initially host the dataset on GitHub, as this was not the recommended platform per the NeurIPS 2025 Call for Papers. Since the rebuttal policy prohibits the use of external links, we will include the GitHub repo link in the final version.
>
> (5)  We have incorporated three new metrics into the verification function:
>
> - **Gate Count Ratio:** the number of quantum gates used by the model, divided by that of the reference implementation. A lower ratio indicates higher gate-level efficiency.
> - **Shot Count Ratio:** the query complexity required by the model, divided by the reference. Again, a smaller value indicates better efficiency.
> - **Time Count Ratio:** the total execution time of the model, divided by the reference. A lower ratio reflects better runtime performance.
>
> We can only include Gate Count Ratio result for 10 algorithms here due to character limit.
>
> **Table 4: Gate Count Ratios in Quantum Algorithm Design**
>
> | Model       | Shot | Bernstein Vazirani | Deutsch Jozsa    | Grover           | Phase Estimation | QFT              | Simon            | GHZ              | Random Number Generator | Swap Test        | W State          |
> | ----------- | ---- | ------------------ | ---------------- | ---------------- | ---------------- | ---------------- | ---------------- | ---------------- | ----------------------- | ---------------- | ---------------- |
> | gpt4o       | few  | 1.0000 (±0.0000)   | 1.0000 (±0.0000) | N/A (±N/A)       | 1.7050 (±0.1680) | 0.2913 (±0.1597) | 1.1715 (±0.0708) | 1.5603 (±0.3735) | 1.2381 (±0.1076)        | 1.0000 (±0.0000) | 0.8000 (±0.1600) |
> | Llama3      | 1    | 2.6001 (±0.9492)   | 1.8570 (±0.4962) | 1.6250 (±0.3707) | 6.5020 (±4.4051) | 1.0833 (±0.0902) | 2.6001 (±0.9492) | 1.7501 (±0.6187) | 2.6001 (±0.9492)        | 1.4000 (±0.2456) | 2.2502 (±0.8897) |
> | Llama3      | few  | 0.9352 (±0.0199)   | 1.0212 (±0.0349) | 0.1070 (±0.0894) | 0.2450 (±0.0502) | 0.3637 (±0.0535) | 1.3333 (±0.0000) | 2.6589 (±0.3712) | 2.6076 (±0.2693)        | 1.2067 (±0.1539) | 1.8369 (±0.4386) |
> | gpt3.5      | 1    | 1.1774 (±0.3244)   | 0.7532 (±0.0647) | 0.7426 (±0.0999) | N/A (±N/A)       | 0.0335 (±0.0092) | 1.5748 (±0.0801) | 1.8185 (±0.7441) | N/A (±N/A)              | N/A (±N/A)       | N/A (±N/A)       |
> | gpt3.5      | few  | 0.7361 (±0.1426)   | 0.2889 (±0.0884) | 0.4746 (±0.1607) | N/A (±N/A)       | 0.0359 (±0.0150) | 1.1104 (±0.0547) | 1.6667 (±0.2778) | 1.0000 (±0.0000)        | 0.6111 (±0.0568) | 0.7185 (±0.0598) |
> | Qwen 2.5    | 1    | N/A (±N/A)         | N/A (±N/A)       | N/A (±N/A)       | N/A (±N/A)       | N/A (±N/A)       | 1.4635 (±0.0945) | N/A (±N/A)       | 1.0000 (±0.0000)        | 1.1594 (±0.1075) | 0.7692 (±0.1775) |
> | Qwen 2.5    | few  | 0.5681 (±0.1452)   | 0.7149 (±0.1125) | 0.4902 (±0.2211) | N/A (±N/A)       | 0.4761 (±0.1299) | N/A (±N/A)       | N/A (±N/A)       | 1.3249 (±0.1629)        | 0.8000 (±0.1048) | 0.7500 (±0.3750) |
> | DeepSeek-R1 | 1    | N/A (±N/A)         | N/A (±N/A)       | N/A (±N/A)       | N/A (±N/A)       | N/A (±N/A)       | N/A (±N/A)       | N/A (±N/A)       | N/A (±N/A)              | N/A (±N/A)       | 0.3200 (±0.0640) |
> | DeepSeek-R1 | few  | 0.7150 (±0.1327)   | N/A (±N/A)       | N/A (±N/A)       | N/A (±N/A)       | N/A (±N/A)       | N/A (±N/A)       | 2.0951 (±0.0996) | 0.5238 (±0.0249)        | N/A (±N/A)       | 0.5310 (±0.1055) |
>
>
>
> (6)  We have now enabled web search for GPT models to simulate an open-book scenario. Here are the results.
>
> **Table 5: Web Search Results for Quantum Algorithm Design**
>
> | Metric   | Bernstein Vazirani | Deutsch Jozsa   | Grover          | Phase Estimation | QFT             | Simon           | GHZ             | Random Number Generator | Swap Test       | W State         |
> | -------- | ------------------ | --------------- | --------------- | ---------------- | --------------- | --------------- | --------------- | ----------------------- | --------------- | --------------- |
> | Semantic | 0.3540(±0.1300)    | 0.7690(±0.1220) | 0.0450(±0.0420) | 0.0770(±0.0520)  | 0.0000(±0.0000) | 0.0020(±0.0020) | 0.0000(±0.0000) | 0.0000(±0.0000)         | 0.4090(±0.1160) | 0.0460(±0.0380) |
> | QASM     | 0.6920(±0.1330)    | 0.8460(±0.1040) | 0.3080(±0.1330) | 0.7690(±0.1220)  | 0.2310(±0.1220) | 0.5380(±0.1440) | 0.8570(±0.1430) | 1.0000(±0.0000)         | 0.9290(±0.0710) | 0.4440(±0.1760) |
> | Code     | 0.8460(±0.1040)    | 0.9230(±0.0770) | 0.9230(±0.0770) | 0.7690(±0.1220)  | 0.5380(±0.1440) | 0.2310(±0.1220) | 0.0000(±0.0000) | 0.0000(±0.0000)         | 0.5000(±0.1390) | 0.4440(±0.1760) |
> | Gate     | 1.0000(±0.0000)    | 1.0000(±0.0000) | 1.4145(±0.2391) | 1.4453(±0.0816)  | 1.0000(±0.0000) | 0.9170(±N/A)    | 0.6870(±N/A)    | 0.6250(±N/A)            | 1.0000(±0.0000) | 1.2860(±N/A)    |
> | Shot     | 1.1427(±0.1631)    | 1.3745(±0.2658) | 0.0435(±0.0197) | N/A(±N/A)        | N/A(±N/A)       | 0.1936(±N/A)    | N/A(±N/A)       | N/A(±N/A)               | 1.0000(±0.0000) | 0.1024(±N/A)    |
> | Time     | 1.9830(±0.0040)    | 1.9970(±0.0145) | 1.4509(±0.3128) | 1.9302(±0.0195)  | N/A(±N/A)       | 1.9470(±0.0049) | N/A(±N/A)       | 13.6940(±N/A)           | 2.0030(±0.0045) | 1.6800(±0.0103) |
>
> The web search setting allows the model to formulate a search query, retrieve results via Google, and incorporate the content from the most relevant link into its response. Notably, the semantic score is mostly lower with web search. For instance, scores dropped from 1.0 (3-shot) to 0.354 (web) on Bernstein–Vazirani, and from 1.0 (3-shot) to 0.769 (web) on Deutsch–Jozsa. This suggests that unguided retrieval may introduce noise or distract from task-specific structure. While few-shot prompting remains competitive, we believe future work in the open-book setting holds promise through the use of stronger structural priors and guided retrieval strategies to better align external information with task-specific objectives.

---

> > ### Comment · Reviewer_igHa · 2025-08-05
> >
> > Thanks for your response. It well address my concerns. Hope to see the revision in the final paper.

---

> > > ### Author Response · Authors · 2025-08-06
> > >
> > > We appreciate the reviewer’s response and are happy that our rebuttal addressed your concerns. We will ensure the revisions are reflected in the final version of the paper.

---

### Official Review · Reviewer_J2nE · 2025-07-02

**Rating:** 5
**Confidence:** 2

**Summary:**

QCircuitBench introduces the first large-scale dataset tailored for benchmarking quantum algorithm design using LLMs. The benchmark includes 128,573 data points spanning three task suites—oracle construction, quantum algorithm design, and random circuit synthesis—covering 23 representative quantum algorithms. The dataset is structured to include problem descriptions, quantum circuit code, post-processing functions, and automated verification, enabling precise and human-free evaluation.

**Dataset Code Accessibility:**

Yes

**Dataset Code Comments:**

The code is clear and professional.

**Ethical Considerations:**

No, there are no or only very minor ethics concerns

**Final Justification:**

The additional experiments make the work more comprehensive and robust. I hence recommend accept the paper.

**Limitations Weaknesses:**

- **Evaluation Metric Fragility and Ambiguity:** The custom verification function outputs a triplet, but the interpretability of negative scores (as shown in Table 1 and Table 2) is not well explained. For example, some models receive scores like -1.0000 or -0.8462 without clear semantic grounding (Sec. 4.1). It is unclear what range these scores span and how the components are aggregated. A clearer explanation or normalization approach would enhance interpretability.

- **Overreliance on Syntax-Level Metrics:** Although BLEU and QASM syntax checks are used, they do not fully reflect semantic correctness or quantum algorithm fidelity. For instance, the swap test is noted to have high BLEU but low functional accuracy (Sec. 4.1, Lines 283–285). While the authors note this discrepancy, they stop short of proposing domain-specific fidelity metrics (e.g., state fidelity or oracle distinguishability) that could better represent quantum functionality.

- **Limited Fine-Tuning Scope:** The fine-tuning experiments are limited to oracle construction only and do not explore quantum algorithm design or variational circuits, despite claiming dataset readiness for those tasks (Sec. 4.2). Additionally, performance on Clifford and universal gate sets degrades after fine-tuning (Sec. 4.2, Table 2), which deserves more detailed analysis than merely adjusting temperature (as deferred to Appendix C.2).

**Strengths Contributions:**

- The modular and extensible dataset structure is particularly well-thought-out. It includes not only OpenQASM 3.0 circuit files and Python-based post-processing logic but also unit test-based automatic verification functions and detailed problem descriptions using natural language and LaTeX math (Sec. 3.2, Fig. 1).

- Fine-tuning experiments on LLaMA-3 demonstrate early promise, showing that the dataset can help models internalize structural patterns in oracle construction tasks (Sec. 4.2). These results make QCircuitBench not only a benchmark but also a testbed for training strategies, with forward-looking implications.

- The paper is very well-written and organized, with clear sectioning, well-chosen examples, and informative figures/tables. Figures like the benchmark architecture (Fig. 2) and algorithm-specific BLEU scores (Fig. 3) are easy to read and insightful. Appendix references are abundant and useful for reproducibility.

---

> ### Author Rebuttal · Authors · 2025-07-31
>
> We thank the reviewer for recognizing the novelty of QCircuitBench, and for highlighting the ``well-thought-out'' modular and extensible dataset structure, its potential as both a benchmark and a training testbed, the quality of the codebase, and the writing of the paper.
>
> (1)  **Evaluation Metric Fragility and Ambiguity:** We apologize for any confusion regarding the verification function scores. The triplet $(V_\text{QASM}(q), V_\text{code}(c), \text{acc})$ in Section 4.1 is the output of the verification function in our codebase. This triplet is subsequently used to compute the final score, which ranges from -1 to 1 (as explained in Section 3.2).  A score of -1 indicates that the program contains syntax errors and cannot be executed. If the program runs successfully, it receives a score in the range [0,1], reflecting the success rate on test cases. The final scores reported in Table 1 are averaged over all the datapoints for each specific quantum algorithm.
>
> To mitigate potential confusion around the verification score, we now clearly separate **syntax checking** from **semantic evaluation**. Specifically, syntax checking is binary: 1 for syntactically correct code, and 0 for code with syntax errors. Semantic evaluation yields a score in the range [0,1], reflecting the fidelity of the generated quantum state or the success rate on test cases.
>
> Here is the updated verification score table (due to the 10,000-character limit, we are only able to provide the semantic scores for 10 algorithms here. The full results will be included in the final version).
>
> **Table 1: Semantic Scores on the Quantum Algorithm Design Task**
> | Model       | Shot | Bernstein Vazirani | Deutsch Jozsa    | Grover           | Phase Estimation | QFT              | Simon            | GHZ              | Random Number Generator | Swap Test        | W State          |
> | ----------- | ---- | ------------------ | ---------------- | ---------------- | ---------------- | ---------------- | ---------------- | ---------------- | ----------------------- | ---------------- | ---------------- |
> | gpt4o       | few  | 1.0000 (±0.0000)   | 1.0000 (±0.0000) | 0.0000 (±0.0000) | 0.0846 (±0.0576) | 0.0000 (±0.0000) | 0.0923 (±0.0769) | 0.0000 (±0.0000) | 0.0000 (±0.0000)        | 0.7852 (±0.0203) | 0.1156 (±0.0585) |
> | Llama3      | 1    | 0.0000 (±0.0000)   | 0.0154 (±0.0154) | 0.0019 (±0.0019) | 0.0000 (±0.0000) | 0.0000 (±0.0000) | 0.0000 (±0.0000) | 0.0000 (±0.0000) | 0.0000 (±0.0000)        | 0.0591 (±0.0591) | 0.0000 (±0.0000) |
> | Llama3      | few  | 0.0442 (±0.0442)   | 0.2269 (±0.1017) | 0.0000 (±0.0000) | 0.0000 (±0.0000) | 0.0000 (±0.0000) | 0.0000 (±0.0000) | 0.0000 (±0.0000) | 0.0000 (±0.0000)        | 0.0986 (±0.0673) | 0.0000 (±0.0000) |
> | gpt3.5      | 1    | 0.0000 (±0.0000)   | 0.1231 (±0.0769) | 0.0000 (±0.0000) | 0.0023 (±0.0023) | 0.0000 (±0.0000) | 0.0062 (±0.0042) | 0.0000 (±0.0000) | 0.0000 (±0.0000)        | 0.0000 (±0.0000) | 0.0000 (±0.0000) |
> | gpt3.5      | few  | 0.1062 (±0.0594)   | 0.4327 (±0.1004) | 0.0000 (±0.0000) | 0.0000 (±0.0000) | 0.0000 (±0.0000) | 0.0138 (±0.0115) | 0.0000 (±0.0000) | 0.0000 (±0.0000)        | 0.2219 (±0.0986) | 0.0000 (±0.0000) |
> | Qwen 2.5    | 1    | 0.0000 (±0.0000)   | 0.0000 (±0.0000) | 0.0000 (±0.0000) | 0.0000 (±0.0000) | 0.0000 (±0.0000) | 0.0000 (±0.0000) | 0.0000 (±0.0000) | 0.0000 (±0.0000)        | 0.0000 (±0.0000) | 0.0000 (±0.0000) |
> | Qwen 2.5    | few  | 0.0154 (±0.0154)   | 0.2854 (±0.0947) | 0.0000 (±0.0000) | 0.0000 (±0.0000) | 0.0000 (±0.0000) | 0.0000 (±0.0000) | 0.0000 (±0.0000) | 0.0000 (±0.0000)        | 0.0000 (±0.0000) | 0.0000 (±0.0000) |
> | DeepSeek-R1 | 1    | 0.0000 (±0.0000)   | 0.0600 (±0.0600) | 0.0000 (±0.0000) | 0.0000 (±0.0000) | 0.0000 (±0.0000) | 0.0000 (±0.0000) | 0.0000 (±0.0000) | 0.0000 (±0.0000)        | 0.0605 (±0.0605) | 0.0000 (±0.0000) |
> | DeepSeek-R1 | few  | 0.0788 (±0.0768)   | 0.0000 (±0.0000) | 0.0000 (±0.0000) | 0.0000 (±0.0000) | 0.0000 (±0.0000) | 0.0000 (±0.0000) | 0.0000 (±0.0000) | 0.0000 (±0.0000)        | 0.0000 (±0.0000) | 0.0000 (±0.0000) |
>
>
> (2)  **Overreliance on Syntax-Level Metrics:** We would like to clarify that our verification function score **explicitly takes semantic correctness or quantum algorithm fidelity into consideration**. The score between [0, 1] measures fidelity of the quantum state for state preparation and random circuit synthesis tasks, and the success rate on test cases for algorithm design and oracle construction tasks. This task-specific scoring framework is designed to rigorously evaluate the quantum functionality and correctness of each algorithm.
>
> (3)  **Limited Fine-Tuning Scope:** We thank the reviewer for the suggestion to broaden the scope of fine-tuning experiments.
>
> - We further conducted fine-tuning experiments on the quantum algorithm design task. Results show clear improvements in code syntax correctness and QASM structural validity. For instance, the code syntax score on the Simon task improved from 0.385 to 0.846, and the QASM syntax score on the Bernstein-Vazirani task rose from 0.154 to 0.385. These gains indicate that fine-tuning helps the model learn the correct syntax of quantum and python programs. However, semantic scores remain near zero on most tasks, with only marginal improvements in a few cases (e.g., 0.015 to 0.327 on Deutsch-Jozsa). This suggests that the model still lacks the functional understanding required to produce correct quantum behavior despite improved syntax.
>
> **Table 2: Fine-Tuning Results for Quantum Algorithm Design**
>
> | Metric   | BV               | DJ               | Grover           | PE               | QFT                          | Simon            | GHZ              | RNG              | Swap Test        | W State          |
> | -------- | ---------------- | ---------------- | ---------------- | ---------------- | ---------------------------- | ---------------- | ---------------- | ---------------- | ---------------- | ---------------- |
> | Semantic | 0.0000 (±0.0000) | 0.3269 (±0.1309) | 0.0000 (±0.0000) | 0.0000 (±0.0000) | 0.0000         (±0.0000)     | 0.0000 (±0.0000) | 0.0000 (±0.0000) | 0.0000 (±0.0000) | 0.0574 (±0.0574) | 0.0000 (±0.0000) |
> | QASM     | 0.3846 (±0.1404) | 0.3846 (±0.1404) | 0.3846 (±0.1404) | 0.3846 (±0.1404) | 0.1538         (±0.1042)     | 0.0769 (±0.0769) | 0.1429 (±0.1429) | 0.5385 (±0.1439) | 0.1429 (±0.0971) | 0.1111 (±0.1111) |
> | Code     | 0.4615 (±0.1439) | 0.7692 (±0.1216) | 0.7692 (±0.1216) | 0.3846 (±0.1404) | 0.8462         (±0.1042)     | 0.8462 (±0.1042) | 0.7143 (±0.1844) | 0.4615 (±0.1439) | 0.7143 (±0.1253) | 0.8889 (±0.1111) |
> | Exact    | 0.0000 (±0.0000) | 0.0000 (±0.0000) | 0.0000 (±0.0000) | 0.0000 (±0.0000) | 0.0000         (±0.0000)     | 0.0000 (±0.0000) | 0.0000 (±0.0000) | 0.0000 (±0.0000) | 0.0000 (±0.0000) | 0.0000 (±0.0000) |
> | Gate     | 0.9648 (±0.0200) | 0.9375 (±0.0586) | 0.7018 (±0.1766) | 0.4591 (±0.1513) | 0.8636         (±0.1179)     | 1.0609       (±N/A)    | 2.0915          (±N/A)    | 2.6929 (±0.4006) | 1.3720 (±0.2417) | 2.5000          (±N/A)    |
> | Shot     | 1.2308 (±0.2841) | 1.0000 (±0.0000) | 0.0370 (±0.0211) | N/A     (±N/A)   | N/A         (±N/A)           | 0.0909          (±N/A)    | N/A          (±N/A)       | N/A         (±N/A)   | 1.9980 (±1.9941) | 0.0001          (±N/A)    |
> | Time     | 2.0801          (±N/A)    | 2.2279 (±0.2977) | 1.8035 (±0.0932) | 2.0692          (±N/A)    | 741.1795         (±101.3296) | 1.9110 (±N/A)    | N/A          (±N/A)       | 5.4995 (±5.4787) | 1.3804 (±0.4477) | 2.3134          (±N/A)    |
>
> - To understand the performance drop on universal/Clifford synthesis tasks, we conducted an entropy analysis before and after fine-tuning. The output entropy decreased noticeably, suggesting the model became more deterministic and less diverse. This likely led to overfitting to high-probability surface patterns, sacrificing semantically correct but less frequent solutions—an issue particularly detrimental for tasks requiring diverse outputs such as random circuit synthesis. This aligns with observed increases in BLEU scores alongside declines in verification scores and points to future directions such as diversity-aware objectives or regularization during fine-tuning. We are happy to include additional experiments that the reviewer considers important for further investigating this phenomenon.
>
> **Table 3: Entropy Comparison Before and After Fine-Tuning**
> | Task | Entropy (Before) | Entropy (After) |
> |------|------------------|-----------------|
> | CL   | 0.7783           | 0.1429          |
> | UN   | 0.8994           | 0.2115          |

---

> > ### Comment · Reviewer_J2nE · 2025-08-05
> >
> > Thank you for the additional experiments, my concerns have been addressed and I have modified my score accordingly.

---

> > > ### Author Response · Authors · 2025-08-06
> > >
> > > We thank the reviewer for the reply. We are glad the additional experiments addressed your concerns and appreciate the score update.

---

### Official Review · Reviewer_rJsa · 2025-07-02

**Rating:** 4
**Confidence:** 4

**Summary:**

Quantum algorithm design is a uniquely challenging problem; and so to be able to leverage AI to design algorithms, a large dataset is required and quick evaluation is needed. As the paper points out, algorithm design for classic computers already has a wealth of information available for training; yet quantum algorithm design is a much younger field with far less such information available. This makes the problem interesting from the point of view of an open AI challenge, as well as having the potential to be transformational to the field of quantum computing research.

Overall, the paper presents an interesting and novel dataset. However, the weaknesses may outweigh the strengths; primarily because there is not enough detail provided to make the dataset evaluation accessible to AI researchers/developers.

**Dataset Code Accessibility:**

Yes

**Ethical Comments:**

A discussion of ethics should be included. Answers of NA still require an explanation.

**Ethical Considerations:**

No, there are no or only very minor ethics concerns

**Final Justification:**

Authors addressed confusion about the evaluation metric. Splitting the evaluation score into separate syntax and semantic scores is a big improvement; and it clarified how the numbers in the table are calculated. They also answered my question about what an optimal score would be (1 is optimal). However, it is clear that the choice of evaluation metric can change the results drastically in terms of evaluating whether an LLM is capable of designing quantum algorithms. The new Table 1 in the rebuttal gives a different overall picture of the state of LLM performance on this task - with many entries zero-ing out while at least one entry improved to 100%.

In summary, the dataset itself is valuable and a significant contribution with potential for high impact. A limitation is that the evaluation approach needs to be carefully considered; including discussion of what level of performance would make it practically useful for quantum computing algorithm designers to rely on LLMs.

**Limitations Weaknesses:**

The major weakness of the paper is in its reproducibility / clarity which limits the usefulness of the dataset. As an AI researcher with some quantum computing experience, it is difficult to understand the description of the data in Section 3. In particular, if the goal is algorithm design and there are 23 algorithms, then what is a datapoint?

I would like to see a metric that clearly indicates whether the problem is “solved”. In other words if an AI model is perfect at generating algorithms, what score would it earn? (The optimal value for BLEU is clear, but BLEU has limitations in evaluating novel algorithms and alternative algorithms). Without knowing the answer to this question, it is very difficult to tell if this dataset is challenging enough to leave room for improvement that AI models would strive for.

Table 1 “verification function scores” do not make sense according to the definition in the previous section. The score should be a triplet according to the definition. In the definition, the values of the triplet are all non-negative; yet the table lists just a scalar value which is often negative.

**Strengths Contributions:**

This is an interesting and novel dataset for quantum algorithm design spanning several types of algorithms with careful consideration about the complexity of the algorithms and the information already available to LLM training (such as online tutorials about specific algorithms). Connections to other datasets in the field are clearly articulated and differentiated.

The potential impact is twofold. First, the field of AI gains insight into the power of LLMs (or any code-generator) to solve challenging algorithm design problems in a space where there is not a lot of training information already available. This potential also makes the paper highly relevant to the field of AI research. Second, if AI is successful on this benchmark, this could be transformational to quantum algorithm design.

Verification uses a variety of standard code verification metrics including BLEU, syntax verification, execution evaluation and perplexity.

---

> ### Author Rebuttal · Authors · 2025-07-31
>
> We thank the reviewer for acknowledging the novelty and transformational potential of QCircuitBench, and for offering valuable insights from the perspective of the AI community.
>
> (1)  Regarding the description of the data in Section 3, we reframe the narrative of the 7 components of the dataset structure to better align with the conventions of the AI community.
>
> - **Input:** Problem Description.
> - **Output:**
>   - For *Quantum Algorithm Design* task: Algorithm Circuit
>         (OpenQASM 3.0) + Post-Processing Function (Python \& Qiskit).
>   - For *Oracle Construction* / *Random Circuit Synthesis*
>         task: Algorithm Circuit (OpenQASM 3.0).
>
> - **Evaluation:**
>   - Verification Function: Returns the syntax / semantic score.
>   - Oracle / Gate Definition: Provides test cases required for the Verification Function to operate.
>
> - **Dataset Preparation:** Dataset Creation Script + Generation Code. These two components are provided to facilitate reproducibility and transparency for those interested in the dataset construction process. These components are not required for AI researchers who only intend to use the dataset for model training or evaluation.
>
>    A detailed dataset description, along with concrete example cases, is provided in Appendix A. We would be happy to provide further clarification on the dataset structure and task suites if the reviewer finds any aspect unclear or would like additional details.
>
> (2)  The 23 algorithms are: *Bernstein-Vazirani Algorithm,
>   Deutsch-Jozsa Algorithm, Simon's Problem, Grover's Algorithm, Shor's Algorithm, Quantum Fourier Transform, Phase Estimation, Generalized Simon's Problem with multiple strings, Generalized Simon's Problem with Tenary Basis, W State Preparation, GHZ State Preparation, Swap Test, Random Number Generator, Superdense Coding, Quantum Teleportation, Quantum Key Distribution, QAOA (for Max-Cut), VQE (with Efficient SU2), Universal Random Circuit Synthesis, Clifford Random Circuit Synthesis, Diffusion Operator Construction, Triangle Finding, Sudoku.*
>
>    Actually there are indeed 15 algorithms in Figure 3 as stated. We politely ask the reviewer to double check the counting. The scope is limited to 15 because some of the algorithms, such as *Diffusion Operator Construction* only apply to the Oracle Construction task, and some of the quantum information protocols, such as *Quantum Key Distribution*, rely on interactive design based on intermediate measurement results, which makes them difficult to integrate into our static circuit / post-processing framework. Nevertheless, we provide tentative implementations for these protocols to support further exploration.
>
>    A datapoint in the Quantum Algorithm Design task corresponds to a quantum algorithm **tailored to a specific setting**, for example, the Bernstein-Vazirani algorithm for a given number of qubits, or the preparation of a particular quantum state such as a GHZ state of fixed size. For certain problems, such as random circuit synthesis, constructing a general solution for all instances is computationally inefficient. For algorithms where a general approach is available, constructing solutions for specific instances may be more tractable, and such instance-level solutions can also offer valuable insights to human researchers.
>
>    We view this as a strategy to **decompose the complex challenge of universal algorithm design** into the more tractable subproblem of designing quantum circuits and post-processing procedures for specific cases. This instance-specific focus serves as a stepping stone toward the more ambitious goal of universal algorithm design. Moreover, defining datapoints in this way helps mitigate the issue of data contamination. Many general algorithm implementations (e.g., from IBM’s Qiskit tutorials) are publicly available, which complicates the evaluation of a model’s genuine ability to design quantum algorithms. By focusing on specific, potentially unseen instances, we create a more robust benchmark for assessing model performance.
>
> (3)  Our proposed **verification function score** is precisely designed to clearly **indicate whether the problem is "solved"**. In the submitted version, the score ranges from -1 to 1 (Section 3.2). A score of -1 indicates that the program contains syntax errors and cannot be executed. If the program runs successfully, it receives a score in the range [0,1], reflecting the success rate on test cases. A score of 1 denotes a correct and fully functional solution, which may differ from the reference implementation, thus allowing for novel but valid alternatives. The final scores reported in Table 1 are averaged over all the datapoints for each algorithm. In this sense, the dataset presents a significant challenge for AI models, as most scores are currently negative.
>
>    To mitigate potential confusion around the verification score, we now clearly separate **syntax checking** from **semantic evaluation**. Syntax checking is performed separately for the OpenQASM quantum circuit and the Python post-processing code, with a binary score: 1 for syntactically correct code, and 0 for code with syntax errors. Semantic evaluation, by contrast, yields a score in the range [0,1]. For state preparation and random circuit synthesis tasks, this score reflects the fidelity between the generated and target quantum states; for algorithm design and oracle construction tasks, it reflects the success rate over test cases. Overall, this task-specific semantic score provides a principled and quantitative measure of quantum correctness tailored to each benchmark. A perfect generation from the model would **achieve a score of 1 in both syntax checking and semantic evaluation**, without any constraints on the BLEU score.
>
>    Due to the 10,000-character limit for responses to individual reviewers, we are unable to include the updated verification score table here. Please refer to tables in Response (1) to Reviewer J2nE for details.
>
>    Additionally, we have introduced three new metrics to evaluate the efficiency of the generated algorithms. For more information, please see Response (5) to Reviewer igHa.
>
> (4)  We apologize for any confusion regarding the verification function scores. The triplet $(V_\text{QASM}(q), V_\text{code}(c), \text{acc})$ in Section 4.1 is the output of the verification function in our codebase. This triplet is subsequently used to compute the final score, which ranges from -1 to 1. For the formal definition and further clarification of the verification function score, please refer to the first paragraph of Response (3).
>
> (5)  We will include a paragraph on ethical considerations in the final version. As quantum computing remains a nascent technology at the moment, we do not foresee any immediate negative societal impacts arising from our work. Looking ahead, we believe our dataset can contribute positively to quantum algorithm design and the field of quantum computing as a whole. We remain committed to monitoring the ethical landscape as the technology evolves.
>
> (6)  We observed that the reviewer selected "Partly" for Dataset Code Accessibility. We would like to clarify that the full code—from dataset generation to evaluation—has been included in the submitted .zip file, along with detailed documentation provided in Appendix A. We are happy to address any confusion or provide further clarification regarding code availability if needed.

---

> > ### Comment · Reviewer_rJsa · 2025-08-08
> >
> > The separation of syntax and semantic evaluation is very helpful; and re-formulating to remove negative scores. If I understand Table 1 in the rebuttal correctly, then if gpt4o gets a syntax-correct solution for BV, then it gets 100% of test cases correct; although it had a negative average score in the original Table 1.
> >
> > I am still confused about why there are so many 0 +/- 0 results in the new Table 1. Many of those entires had numbers > -1 in the original Table 1 (e,g, Llama3-few on Grover and Phase Estimation).
> >
> > I'll try to clarify my question about whether the quantum algorithm design problem is "solved". From your response, it sounds like if an LLM gets a correct solution just once, then it has solved that algorithm design. So any score greater than 0 is success? One advantage of AI is that we can run it cheaply to produce many potential solutions. If those solutions are easy to check, then getting 1 good solution out of a million wrong ones is fine. But if they are difficult to check, then how many incorrect solutions are we willing to evaluate to find one good one? Negative scores indicate syntax problems which are easy to check for and eliminate those from consideration. The fact that the score for gpt4o-few on BV went from negative in the old Table 1 to 1.0000 +/- 0.0000 in the new Table 1 demonstrates just how important the choice of evaluation metric is for validating whether or not the problem is solved. A small change in your metric means that the unsolved problem is now solved.

---

> > > ### Author Response · Authors · 2025-08-09
> > >
> > > We thank the reviewer for the response. We are glad that the separation of syntax and semantic evaluation is helpful.
> > >
> > > In the original Table 1, if the model generated a syntactically incorrect quantum circuit, we replaced it with the reference circuit to test the post-processing function. If the final answer was correct, we awarded half of the full score, as this indicated the post-processing function was both syntactically and semantically correct. Consequently, **partial syntax errors could still yield non-negative scores**.
> > > In the new Table 1, to remove any ambiguity, we **only proceed to semantic evaluation when both** the generated quantum circuit and the post-processing function **are syntactically correct**. This stricter criterion results in many 0 +/- 0  entries in the new table, whereas the corresponding entries in the original table could be greater than –1.
> > >
> > > As explained in the rebuttal, a perfect generation achieves a score of 1 in both syntax and semantic evaluation. Thus, **a score > 0 does not mean the algorithm design is “solved”**. It only indicates that the generated solution passes some test cases or partially overlaps with the target state.
> > >
> > > “How many incorrect solutions we are willing to check” is a fascinating question. The answer depends primarily on the computational resources available for simulation, which constitutes the sole cost in our verification procedure. Consequently, this threshold can vary substantially across different parties.
> > >
> > > We respectfully disagree with the statement that “a small change in your metric means that the unsolved problem is now solved.” While GPT-4o now produces a syntax-correct solution for BV and achieves 100% correctness in the new table, this outcome is unsurprising given that BV is the simplest textbook quantum algorithm consisting of only a single layer of Hadamard gates before and after the oracle, and a trivial post-processing step that directly returns the measurement results. Crucially, the shift from a negative score to a full score is not due to a minor change in our evaluation metric, but rather reflects **changes in the underlying model**. We have observed performance fluctuations in GPT-4o throughout our experiments. We tested GPT-4o on non-quantum tasks at different timestamps, and found markedly different outputs and response styles. This suggests that OpenAI may have updated or iterated on the model while keeping the same GPT-4o name. Our evaluation metric does not create the change—it is simply **sensitive enough to faithfully capture the variations resulting from model updates**.

---

> ### Author Response · Authors · 2025-08-06
>
> Dear Reviewer rJsa:
>
> Thank you again for your thoughtful review and for offering valuable insights from the perspective of the AI community.
>
> As the discussion period ends in two days, we kindly follow up to see if you have any further comments on our rebuttal. We would be grateful for the opportunity to address any remaining concerns.
>
> Best regards,
>
> Authors

---

### Note · Authors · 2025-08-12

We thank the AC and reviewers for their constructive engagement during the rebuttal and discussion period.

**(1) Contributions**

QCircuitBench introduces the first comprehensive benchmark dataset tailored for AI-driven quantum algorithm design. It defines a general framework which formulates the key components of quantum algorithm design for LLMs. Insights from extensive experiments position QCircuitBench as both a benchmark and a forward-looking testbed for developing next-generation AI models for quantum computing.

We appreciate the reviewers’ recognition of:
- **Novelty and impact**: “Highly relevant to the field of AI research… transformational to quantum algorithm design” (rJsa); “Novel and interesting” (igHa); “Stands our for its novel focus… filling a critical research gap… a foundational resource” (A8iu).
- **Dataset design**: “Modular and extensible dataset structure… particularly well-thought-out” (J2nE); “Careful consideration about complexity and available training information” (rJsa).
- **Insightful experiments**: “Early promise… with forward-looking implications” (J2nE); “Extensive experiments… derive new insights” (igHa); “Actionable insights for future research” (A8iu).
- **Clarity and reproducibility**: “Very well-written and organized… code is clear and professional” (J2nE); “Ensure researchers can readily use, validate, and extend the work” (A8iu).

**(2) Addressing Concerns**

 As indicated by the reviewers, main concerns—interpretation of verification score and scope of experiments—have been effectively resolved during rebuttal. We clarified verification function as a task-specific quantum metric with semantic depth, and introduced a revised definition separating syntax and semantic evaluation. Additional experiments conducted include:
- Benchmarking on revised verification function and new efficiency metric.
- Fine-tuning on quantum algorithm design tasks.
- Entropy analysis before/after fine-tuning.
- Open-book setting with web search.
- Integration with Cirq.

These enriched the experimental scope and demonstrated QCircuitBench’s generalizability.

**(3) Final Version Plans**

We will add:
- Full experimental tables for updated verification scores.
- Human performance baseline.
- Integration with other quantum programming frameworks.
- Implementation of more VQAs.
- Full visualization of reference/model-generated circuits.
- Highlights of conclusions, experimental takeaways, ethical considerations.
- GitHub repository link.

---

### Decision · Program_Chairs · 2025-09-18

**Decision:**

Accept (poster)

**Comment:**

The paper introduces QCircuitBench, a large-scale benchmarking dataset for AI-based quantum algorithm design with LLMs. It provides a systematic collection of 3 tasks (oracle construction, quantum algorithms design, random circuit synthesis), covering 23 algorithms, and 128,573 data points (i.e., specific instances of quantum algorithms tailored to a particular setting), across diverse problem classes of different complexities and difficulties. The dataset is modular and extensible with automatic verifications functions. The dataset is accompanied by baseline studies, detailed analyses, and open-source resources, making it a timely and valuable contribution to both the quantum computing and machine learning community.

During the rebuttal and discussion, the authors addressed key reviewer concerns. They clarified the methodology (e.g. data points and verification score). They also expanded the scope of the experiments by adding fine-tuning results on algorithm design tasks, efficiency metrics, entropy analysis, open book evaluation, and an integration with Cirq. Reviewers confirmed that these additions strengthened the work and satisfactorily addressed their earlier reservations. The authors also promise to add further material for the final submission of the paper.

While some limitations remain (e.g., (1) the evaluation metrics, while clarified, remain sensitive and strongly affect how performance is interpreted; (2) Fine-tuning experiments still show limited semantic correctness, indicating that the task is far from solved), I recommend accepting the paper. QCircuitBench is a timely and foundational resource at the intersection of AI and quantum computing. It fills an important gap by providing a rigorous, large-scale benchmark that can reveal the strengths and weaknesses of current LLMs, and it has the potential to become a widely adopted testbed that advances both AI for science and quantum algorithm design.

In further encourage the authors to include the promised additional content into the final camera-ready version of the paper, especially the visualizations, the human baselines, and the additional variational quantum algorithms (VQAs).